

# Impacts of the emergency operation of the South-to-North Water Diversion Project's eastern route on flooding and drainage in the water-receiving area: An empirical case from China

Kun Wang[1,2], Zongzhi Wang[2], Kelin Liu[2], Liang Cheng[2], Lihui Wang[3], Ailing Ye[2,3]

[1]College of water Conservancy and Hydropower Engineering, Hohai University, Nanjing, 210098, China
[2]State Key Laboratory of Hydrology-Water Resources and Hydraulic Engineering, Nanjing Hydraulic Research Institute, Nanjing 210029, China
[3]College of Civil Engineering, Fuzhou University, Fuzhou 350002, China

*Correspondence to*: Zongzhi.Wang (wangzz77@163.com)

**Abstract.** The water levels of lakes along the eastern route of the South-to-North Water Diversion Project (ER-SNWDP) are expected to rise significantly and subsequently affect the process of flood control and drainage in those lake basins. However, few studies have focused on the impacts of interbasin water diversion on the flood control and drainage of water-receiving areas at the lake basin scale. Using MIKE software, this paper builds a coupled hydrodynamic model to address the existing literature gap on the impacts of interbasin water diversion on the process of flood control and drainage in a water-receiving

lake basin, and it considers the many types of hydraulic structures in the model. First, a flood and waterlogging simulation model was constructed to simulate the interactions among the transferred water, waterlogging of the lakeside area surrounding Nansi Lake (NL), and water in NL and its tributaries. The ER-SNWDP was also considered in the model. Second, the model was calibrated and verified with measured data, and the results showed that the model is efficient and presents a Nash-Sutcliffe efficiency coefficient (NSE) between 0.65 and 0.99. Third, the process of flood and drainage in the

lakeside area of NL was simulated under different water diversion and precipitation values. Finally, the impacts of emergency operations of the ER-SNWDP on flood control and waterlogging drainage in the lakeside area of NL were analyzed based on the results from the proposed model, and selected implications are presented for the integrated management of the interbasin water diversion and the affected lakes.

## 1 Introduction

Interbasin water diversion is a useful approach to solving the spatial unevenness of water resources; however, this method changes the hydrologic characteristics of the water-receiving area. The eastern route of the South-to-North Water Diversion Project (ER-SNWDP) links Gaoyou Lake, Hongze Lake, Luoma Lake, Nansi Lake and Dongping Lake with 13 pump stations that transfer water from the downstream Yangtze River to the Huang-Huai-Hai Plain and Shandong Peninsula. These densely populated lake basins have contributed to the agricultural and economic development. Many rivers flow into

the lake. The terrain of the area around the lake (known as the lakeside area) is flat and low lying, and once heavy rainfall occurs, drainage is difficult due to blocking of the rising lake level, which results in waterlogging disasters in the lakeside area. According to the comprehensive plan of the SNWDP, (1) the first phase of the eastern route is designed to transfer 8.8 billion m³ of water annually, and approximately 7 billion m³ is expected to be consumed in the lakes and basins along the route (Webber et al., 2017); and (2) the water diversion period covers the non-flood season (October to May), and the water

level of the lakes is expected to obviously increase during project operations. However, the Shandong Peninsula has suffered from severe drought conditions for four consecutive years since the eastern route operated in 2013, and emergency water diversion has been performed many times to supply water for this area during the flood season. Furthermore, considering the rigid demand for water resources caused by rapid economic and social development, extreme hydrological events caused by environment changes have increased along the ER-SNWDP. Thus, frequent water transfers are expected in the non-flood




season to alleviate water shortages in the water-receiving areas, which extends beyond the planned design of the ER-SNWDP.

Therefore, simulations of the flood and waterlogging process in the lakeside areas under the condition of emergency water diversion by the SNWDP and analyses of the impacts of emergency water diversions on flood and waterlogging drainage characteristics and the scheduling of flood control and drainage projects must be performed to strengthen the scientific scheduling of water diversion projects and flood control projects in water-receiving regions. This situation represents a major scientific problem that must be resolved. However, to the best of the authors' knowledge, few studies have focused on the impacts of interbasin water transfer on flooding and drainage in lake basins along the water diversion route. Therefore, an important gap exists in the literature. Apart from a few papers, the effects of water-level increases caused by water diversion on the flooding and drainage process and inundated areas in lake basins have not been estimated via relevant models. Using the MIKE software package, this paper proposes a model that integrates MIKE 11, MIKE 21 and MIKE FLOOD to contribute to the existing literature and address the gap mentioned above.

Based on data availability and regional distribution, Nansi Lake Basin (NLB) is chosen as the research area in this study because this lake basin is an important storage node of the ER-SNWDP. According to the overall plan of the SNWDP, the water level of the upper lake is expected to rise by 50 cm and the lower lake is expected to rise by 70 cm during the project operation period. This basin is located in the north-south climate transition zone of China, and the phenomenon of abrupt alternation of droughts and floods frequently occurs. Because the lake is shallow and the flood drainage efficiency in the lakeside area has a sensitive response to the lake level, the NLB has a history of frequent flooding and waterlogging disasters. This paper attempts to address three issues:

(1) Identify a method of building a flood and waterlogging simulation model for a water-receiving region with multiple hydraulic structures under water diversion;

(2) Determine how emergency water diversion affects the flooding and drainage process in the NLB and analyze the waterlogging situation in the lakeside area of NL;

(3) Evaluate how to balance the risk of water shortage and flooding caused by interbasin water diversion in the water-receiving regions.

## 2 Literature review

### 2.1 Impacts of interbasin water diversion on the water-receiving regions

Due to the uneven spatial distribution of water resources and regional socio-economic development, the demand for water in certain regions far exceeds the available water amount, thereby resulting in an increasingly serious imbalance between water demand and supply (Cai and Ringler, 2007; Hu et al., 2010; Matete and Hassan, 2006; Webber et al., 2017). As the most effective and direct method of resolving the problem of water resource shortages, interbasin water diversion projects have been widely applied in water-deficient areas around the world (de Andrade et al., 2011; Wang et al., 2014; Zhang et al., 2015). According to data released by ICID (2005), the total annual amount of water transferred by water diversion projects around the world is 540 billion m$^3$, which accounts for approximately 14% of the world's annual water intake. By 2025, the annual water diversion is expected to reach 940 billion m$^3$. In the water supply and receiving area, interbasin water transfer projects significantly affect hydrological elements, such as the water quantity, water quality, water environment and flood disasters. A full understanding of these impacts is key to the scientific management and long-term operation of interbasin water diversion and represents the most popular global topic in water resource planning and management research (de Andrade et al., 2011; Aron et al., 1977; Davies et al., 1992; Khan et al., 1999; Liu and Zheng, 2002). Zhang et al. (2015) summarized relevant studies on interbasin water transfer from 1991 to 2014 and noted that the effects on the hydrological




environment caused by China's SNWDP and the corresponding long-term monitoring and protection policy for this project represent the most important current issues.

Current research on the hydrological effects caused by interbasin water transfer mainly focuses on the following aspects. (1) For groundwater, Kundell (1988) argues that a large amount of imported water significantly increases the available water and directly participates in the water cycle of the water-receiving regions, which has a positive effect on the water environment, groundwater exploitation and wetland restoration. Relevant studies indicate that a large amount of imported water can effectively alleviate the problem of decreased groundwater levels and ground subsidence caused by the perennial over-extraction of groundwater in selected areas (Larson et al., 2001; Liu and Zheng, 2002; Wang et al., 2014). Based on a large distributed hydrological model, Ye et al. (2014) evaluated the effect of the middle route of the SNWDP on the groundwater level of Haihe basin. The results showed that although the imported water cannot change the decreasing trend of the groundwater level in the water-receiving area, it can significantly reduce the rate of decrease. (2) Water quality is one of the most important factors underlying the success of interbasin water transfer projects. Scholars have simulated and evaluated the effects of interbasin water transfer projects on water quality in water supply and receiving areas. The imported water dilutes the concentration of nutrients, improves the ratio of runoff and pollution in the receiving area and subsequently improves the water quality. However, interbasin water transfer projects might also transfer pollutants from the water supply area or the river basin along the water diversion line into the water-receiving regions, thus worsening the water quality (Hu et al., 2008; Karamouz et al., 2010; Tang et al., 2014; Welch et al., 1992; Zhai et al., 2010). The Chicago interbasin water diversion project, which uses the Lake Michigan basin as its source, has come under criticism due to its chronic exposure risk of organic pollutants (Rasmussen et al., 2014). (3) Interbasin water transfer brings water from the water-supplying area to the water-receiving area through the water transmission channel, which is not conducive to flood control in the water-receiving area and the water transmission channel. Wang et al. (2013) studied the influence of an inter-basin water transfer project on hydraulic parameters during the flood season in a water-supplying area. Based on a two-dimensional mathematic model, Sun et al. (2008) took the Anyang River Basin as an example to study the influence of the middle line of the SNWTP on flood disasters in the river basin through which the project passes. As the storage node of the water diversion project, the large amount of water transferred to the lake significantly changes the interaction law between the water body of the lake basin and the water in the lake tributaries and subsequently affects the flood control and drainage of the lake basin. However, few quantitative research studies have focused on this issue.

## 2.2 Simulation of flooding and waterlogging disasters in a basin

Simulating the flood and waterlogging process based on a mathematical model is an important method for analyzing flood and waterlogging characteristics and assessing the disaster risk of a basin, and it is also an effective tool for planning the layout of flood and waterlogging control engineering (Dutta et al., 2015; Liu et al., 2015; Wang et al., 2018). The early flood and waterlogging simulations of a basin are mainly based on hydrological models, including SWMM (Lee and Heaney, 2003), MUSIC (Dotto et al., 2011; Hamel and Fletcher, 2014), SWAT (Dixon and Earls, 2012), and MIKE SHE (Vrebos et al., 2014), among others. However, hydrological models only simulate the flood-routing process according to the water balance equation and are unable to display the spatial distribution of flood movement. In addition, these models cannot accurately simulate the drainage process of the sluice, dam, pumping station and pipeline hydraulic structures. A hydrodynamic model simulates the water routing by solving the Saint-Venant equations, which can accurately reflect the movement of water in the plane and various hydraulic structures. With improvements in computer processing speed and the development of spatial digital elevation information, hydrodynamic models have gradually become an important tool for flood simulations (Moel et al., 2015). Hsu et al. (2000) built a waterlogging simulation model by coupling the SWMM model and a two-dimensional hydrodynamic model and simulated the rainstorm waterlogging of Taipei City. Bisht et al. 2016 combined SWMM with the MIKE URBAN and MIKE 21 models to simulate waterlogging in West Bengal, India. Li



et al. (2016) established a 1D and 2D coupling hydrodynamic model of Taining county in China based on MIKEFLOOD model, and simulated and analyzed the flood and waterlogging risk in the region. The MIKE 11 and MIKE 21 hydrodynamic models can simulate the influence of a variety of hydraulic structures on the water flow movement process, and the one-dimensional model and two-dimensional model can describe the coupling in different ways. Therefore, this model has been

applied to simulate flow movement of a variety of water bodies, including rivers, lakes, land and estuaries (Karim et al., 2016; Quan, 2014; Zolghadr et al., 2010).

### 3 Research area and data sources

### 3.1 Research area

NL (34°27'N–35°20'N, 116°34'E–117°21'E) is composed of four consecutive lakes, namely, Nanyang Lake, Dushan Lake,

Zhaoyang Lake and Weishan Lake (Fig. 1), and it is a typical large and shallow lake with an area of 1266 km$^2$ and an average depth of only 1.5 m (An and Li, 2009). To manage flooding in this basin, a pivotal project composed of a dam and sluices (Erji dam) was constructed at the middle of NL, and it divides the lake into an upper lake and lower lake. The sluices of the Erji dam control the flood discharge of the upper lake, and the Hanzhuang sluice and Linjia sluice control the flood discharge of the lower lake. The NLB is located in Yi-Shu-Si river system of the Huaihe River Basin with an area of 31700

km$^2$. The lakeside area, with a ground elevation of 36.79 m or less around the lake, is approximately 3969 km$^2$ and has a slope between 1/5000 and 1/20000 (Tian et al., 2013; Wang et al., 2010). A total of 53 rivers flow into NL, and 11 of them have a drainage area greater than 1000 km$^2$. Due to the low height and gentle slope of the riverbed, the rivers in the lakeside area have strong interactions with NL. Flood control embankments have been built on both sides of the main inflow channels and around NL to prevent flooding from entering the lakeside area. Due to the low-lying terrain and the construction of flood

control embankments, waterlogging in lakeside areas can no longer directly drain into rivers and lakes. The waterlogging water is primarily carried into rivers and NL by pumping stations in the lakeside area. However, because the drainage ability of the pumping stations is limited and the rainfall is concentrated, the NLB belongs to the historical frequent flooding area.



**Figure 1: Sketch map of the eastern route of the South-North Water Transfer Project and Nansi Lake BasinThe logo of Copernicus Publications.**

The ER-SNWDP transfers water from Yangzhou to NL and is divided into two sections in the NLB: water entering the lower lake at the Hanzhuang sluice and water entering the Linjia dam sluice. A pumping station has been built at the secondary dam to lift water from the lower lake to the upper lake, and the Changgou pumping station will be built at the higher lake 24.6 km from the Liangji River mouth to send water to the Shandong Peninsula. According to the first phase of the SNWDP, the discharge that enters the lower lake occurs at 200 m³/s, and 5/8 of this amount will be pumped into the upper lake. During project operation, the water level of lower lake will reach 32.8 m, which is 0.70 m

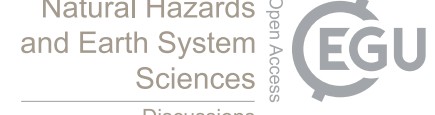

higher than the mean annual water level. The project also transfers water at 100 m³/s from the upper lake to the north. The water level of the upper lake maintains a normal storage level of 34 m, which is 0.48 m higher than the mean annual water level.

### 3.2 Data sources

The data sources for this study include terrain, hydrology, meteorology and hydraulic engineering data from the NLB and engineering layout and operation data of the eastern route of the SNWDP. (1) The digital elevation model (DEM) and river channel bathymetry were supplied by the Planning and Design Institute of the Huaihe Basin Hydraulic Management Bureau in Shandong Province, China. The DEM of the lakeside and NL in 2013 was derived from 1:7000 topographic maps, and the river channel bathymetry used in the 1-D model was reflected by 550 cross-sections separated by distances between 500 and

1000 m. (2) The hydrological data originated from the Shandong Provincial Hydrology Bureau. These data mainly include the discharge processes of typical floods in the upper boundaries of the rivers and the daily rainfall records at seven rainfall stations: Huayu, Liangshanzha, Wanglu, Wanggudui, Wangzhong and Xuecheng. Each station has daily precipitation records covering approximately 30 to 50 years. The daily water level records of four stations, Nanyang, Makou, Erji Lake, and Weishan (shown in Fig. 1), were also supplied. (3) The meteorological data were downloaded from the National

Meteorological Scientific Data Sharing Service Platform (http://data.cma.cn/), and the data include daily records of numerous meteorological parameters, including the wind field and evapotranspiration information. (4) The hydraulic engineering data were supplied by the Planning and Design Institute of the Huaihe Basin Hydraulic Management Bureau in Shandong Province, China. These data include the location and drainage capabilities of the pump stations, the locations and sizes of flood control embankments, and the hydraulic parameters of sluices. (5) The engineering data of the ER-SNWDP

were supplied by the Planning and Design Institute of the Huaihe Basin Hydraulic Management Bureau in Shandong Province, China.

### 4 Methodology

### 4.1 Research framework

To quantitatively study the impact of the water diversion project on flood control and drainage in the NLB, a hydrodynamic

model of waterlogging in lakeside areas of NL was constructed based on MIKE software with consideration of the ER-SNWDP. This model includes a one-dimensional model to simulate flood routing in the river that flows into NL (MIKE 11) and a two-dimensional model to simulate the evolution of plane flow in the lakeside area of NL (MIKE 21). Different hydraulic structures are set up in the model to simulate the flood control and drainage process of sluices, dams, pumps and other hydraulic structures in the research area and the water lifting process of the ER-SNWDP pumping station. Coupling of

the one-dimensional and two-dimensional models is supplied by reasonable links to reflect the interaction of water in the NL, tributary rivers and lakeside area. The established model is used to simulate the waterlogging process in the lakeside area under different scenarios. According to the results of the calculation, the influence law of the SNWDP on waterlogging in the lakeside area of NL is analyzed. Finally, related suggestions for balancing the water diversion and waterlogging risk are proposed. Fig. 2 illustrates the research framework of this paper.





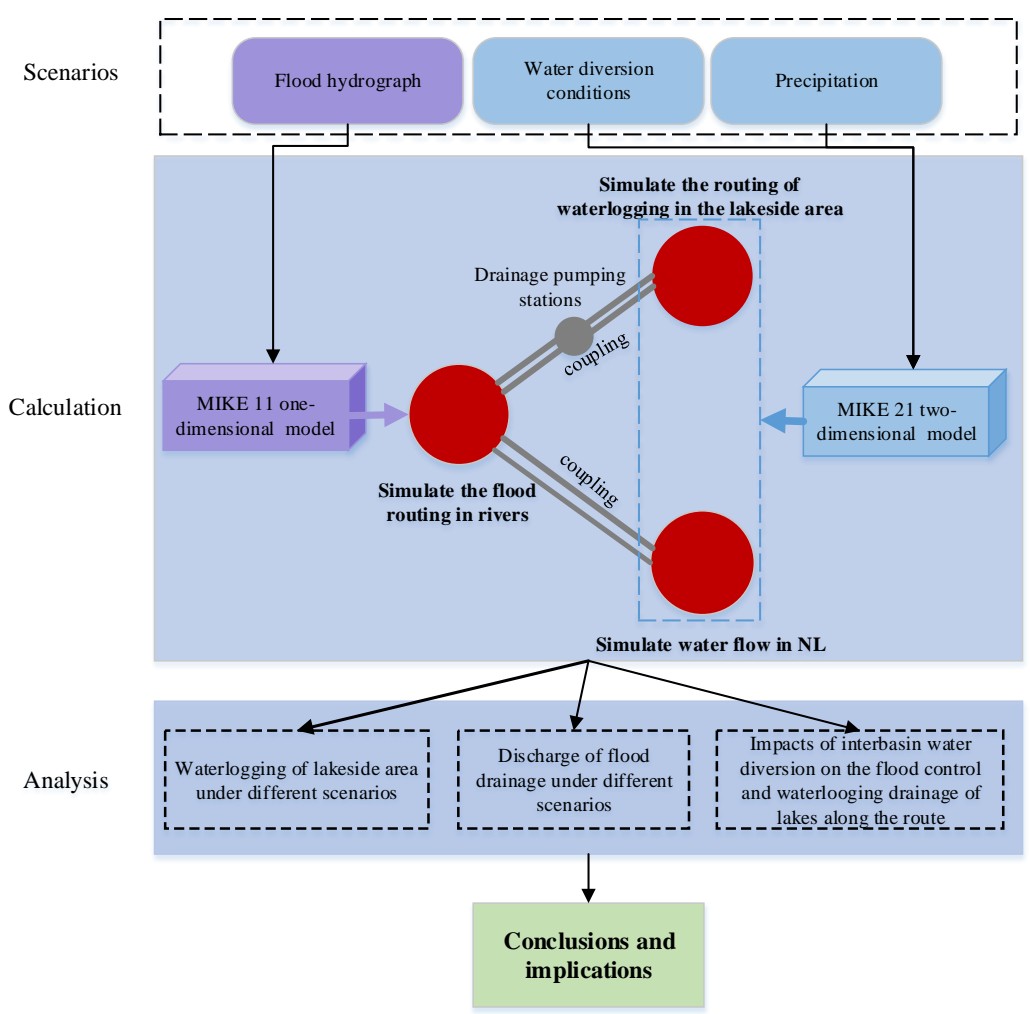

**Figure 2: Research framework of the interbasin water diversion influence on flood control and waterlogging drainage of lakes along the route.**

**4.2 One-dimensional hydrodynamic model of river net**

5   A total of 53 tributary rivers are located around NL, and they represent the key to studying the influence of water diversion on waterlogging in the lakeside area for accurately simulating the flood evolution and the interaction between flood evolution and the high water level of NL. Therefore, a 1-D mathematical model (MIKE 11) was used to simulate the flood routing. The control equation of the model is the Saint-Venant equations, which are composed of the continuity equation and momentum equation (DHI,2007):

10   continuity equation:

$$B = \frac{\partial z}{\partial T} + \frac{\partial Q}{\partial x} = q \ , \tag{1}$$

momentum equation:

$$\frac{\partial Q}{\partial t} + \frac{\partial}{\partial x}\left(\frac{\alpha Q^2}{A}\right) + gA\frac{\partial z}{\partial x} + \frac{gQ|Q|}{C^2 AR} = 0, \tag{2}$$

where x and t denote the spatial and temporal coordinates, respectively; A is the cross-sectional area; Q and z denote the

15   discharge and water level of cross-section, respectively; q is the lateral inflow; R is the hydraulic radius; C is the Chezy





coefficient; a is the momentum correction factor; and g denotes the gravitational acceleration. The Abbott-Ionescu 6 implicit difference method is used to solve the equation.

First, we generalized the river net based on a consideration of the data and computational efficiency. This river net primarily contains 11 rivers with drainage areas greater than 1000 km². A total of 550 cross sections were input into the river network
model to reflect the changes of river topography with adjacent sections spaced approximately 1000 m apart. We generalized the drainage pump stations in the lakeside area of NL. A total of 1000 pump stations are present in the lakeside area of NL, and the model generalized the pump station according to each pump station distribution on both sides of the rivers. The basic principle is to ensure that the total drainage discharge remains the same, with the generalized pumping stations along the river on both sides evenly distributed for a total of generalized 41 pumping stations. Finally, the boundary conditions of the
model were set. The upper boundary of the model inputs the discharge hydrograph of the upstream hydrological station of each river. As the lower boundary, the water level of the estuary is based on NL, which is simulated by the two-dimensional model.

**4.3 Two-dimensional hydrodynamic model of NL and the lakeside area**

The MIKE 21 hydrodynamic model was used to simulate the water movement in NL and the waterlogging evolution in the
lakeside area. The model covers the area below the 36.79 m contour line around NL and the lake surface. Because the one-dimensional model is adopted to simulate the rivers around the lake, the area of the rivers is removed from the two-dimensional model. The two-dimensional model is based on the Reynolds average stress equation of a three-dimensional incompressible fluid, which is subject to the Boussinesq hypothesis and the hydrostatic pressure hypothesis, and the control equations used in this model are given as follows (DHI, 2007):

continuity equation:

$$\frac{\partial h}{\partial t} + \frac{\partial h\bar{u}}{\partial x} + \frac{\partial h\bar{v}}{\partial y} = hS \ , \tag{3}$$

momentum equation of x direction:

$$\frac{\partial h\bar{u}}{\partial t} + \frac{\partial h\bar{u}^2}{\partial x} + \frac{\partial h\bar{v}\bar{u}}{\partial y} = f\bar{v}h - gh\frac{\partial \eta}{\partial x} - \frac{h}{\rho_0}\frac{\partial p_a}{\partial x} - \frac{gh^2}{2\rho_0}\frac{\partial \rho}{\partial x} + \frac{\tau_{sx}}{\rho_0} - \frac{\tau_{bx}}{\rho_0} - \frac{1}{\rho}\left(\frac{\partial s_{xx}}{\partial x} + \frac{\partial s_{xy}}{\partial x}\right) + \frac{\partial}{\partial x}\left(hT_{xx}\right) + \frac{\partial}{\partial x}\left(hT_{xy}\right) + hu_sS, \tag{4}$$

momentum equation of y direction:

$$\frac{\partial h\bar{v}}{\partial t} + \frac{\partial h\bar{v}^2}{\partial y} + \frac{\partial h\bar{u}\bar{v}}{\partial x} = f\bar{u}h - gh\frac{\partial \eta}{\partial y} - \frac{h}{\rho_0}\frac{\partial p_a}{\partial y} - \frac{gh^2}{2\rho_0}\frac{\partial \rho}{\partial y} + \frac{\tau_{sy}}{\rho_0} - \frac{\tau_{by}}{\rho_0} - \frac{1}{\rho_0}\left(\frac{\partial s_{yx}}{\partial y} + \frac{\partial s_{yy}}{\partial x}\right) + \frac{\partial}{\partial x}\left(hT_{xy}\right) + \frac{\partial}{\partial y}\left(hT_{yy}\right) + hv_sS, \tag{5}$$

where *x*, *y* and *z* are Cartesian coordinates; *t* denotes temporal coordinates; $\eta$ is the bottom elevation of the river; *d* is the depth of the water, $h=d+\eta$ is the total head of the water; *u*, *v* and *w* are the velocity components in the *x*, *y* and *z* directions, respectively; $p_a$ is the local atmospheric pressure; $\rho$ is the density of water; $\rho_0$ is the reference water density; $f=2\Omega sin\phi$ is a Coriolis parameter; $f\bar{v}$ and $f\bar{u}$ are the acceleration caused by the earth's rotation; $S_{xx}$, $S_{xy}$, $S_{yx}$ and $S_{yy}$ are the components of
the radiation stress tensor; $T_{xx}$, $T_{xy}$, $T_{yx}$ and $T_{yy}$ are the horizontal viscous stresses; *S* is the magnitude of the discharge due to point sources; and $(u_s,v_s)$ is the velocity by which the water is discharged into the ambient water.

The total area of the 2-D model is approximately 4750 km², with both sides of the river embankment and the 36.79 m contour line as the outer boundaries of the model. Dikes were set up on both sides of river around NL and at the Erji dam to simulate the flood control effect of levees. Sources were added to simulate the water transfer process of the ER-SNWDP.
When the model is running, the rainfall process of the XueCheng (XC), ZhongXingJi (ZXJ), WangZhong (WZ), JuYe (JY), and HouYing (HY) rainfall stations in the research area are input.





### 4.4 Waterlogging simulation model of NL and lakeside area with consideration of the SNWDP

The MIKE FLOOD model is used to couple the 1-D and 2-D model, and the specific process is described as follows: (1) A lateral link is applied to connect the into-lake rivers and lakeside area of NL to simulate the flood overtopping of dikes; and (2) a standard link is applied to connect the into-lake rivers and NL to reflect the influence of the height of the lake level in

blocking the drainage of the into-lake rivers. A total of 22 lateral connections and 52 standard connections are present in the coupling model.

### 4.5 Calibration and validation of the coupling model

The model is calibrated and validated using two actual floods that occurred in July 2007 and July 2008 in the NL basin. Fig. 3 shows the simulated and measured water levels of four stations: Nanyang (NY), Makou (MK), Erji Dam (ED) and

WeiShan (WS) in NL. Overall, the measured water level process shows good agreement with the simulated water level process, and the arrival of the simulated flood peak is consistent with the measured data. The Nash-Sutcliffe efficiency coefficient (NSE), which was proposed by Nash and Sutcliffe (1970), is used to evaluate the coupling model. The NSE for the daily flow varied from 0.67 (Erji Lake) to 0.82 (Weishan) during calibration and from 0.65 (Nanyang) to 0.99 (Weishan) during verification (Table 1), thus showing good agreement between the observed and simulated water levels. As a result,

the calibrated roughness coefficients n were 0.055 for agricultural fields, 0.08 for residential areas in the lakeside area, and 0.028 for NL.

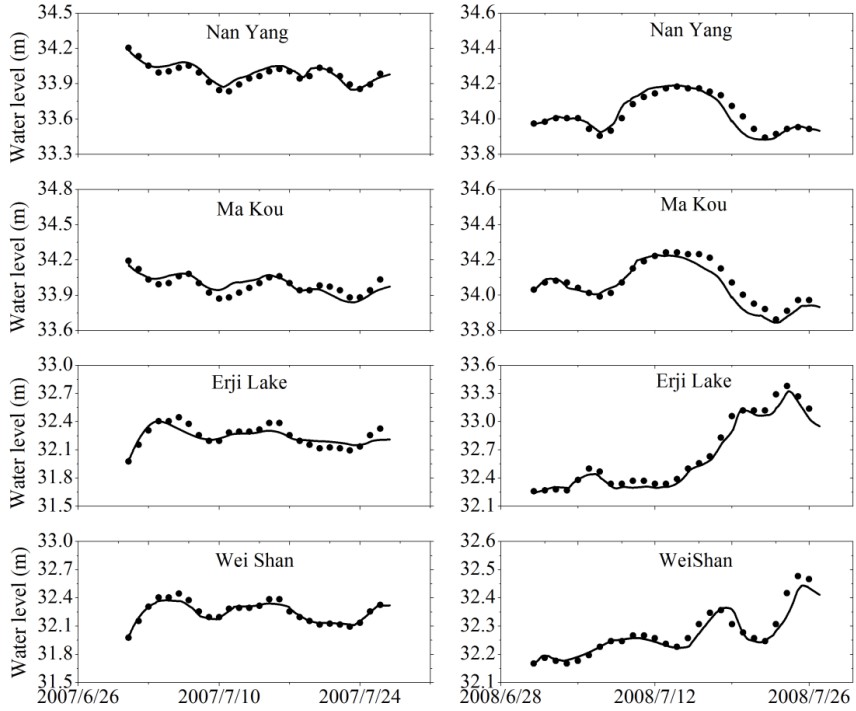

**Figure 3: Comparison of the observed and simulated water levels at selected locations for 2007 flood events under calibration conditions (left) and 2008 events under validation conditions (right). Black dots represent observed data, and black lines represent**
**model simulation results.**

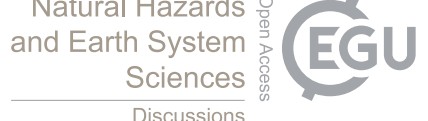



## 5 Results and discussion

### 5.1 Scenario design

NL is located in the China's north-south climate transition zone, and the time distribution of rainfall is severely uneven, with rainfall in the flood season accounting for 72% of the annual precipitation. The phenomenon of abrupt drought-flood

alternations frequently occurs, with this region experiencing drought disasters prior to the arrival of the flood season and waterlogging disasters during the flood season. For example, in 2003, the rainfall of this basin was nearly 0 mm in the first half year, and the upper lake of NL even had a dry period from April to June. However, heavy rains occurred after August 22 and caused a steep increase in the water level in NL, and 2003 represented a typical year of abrupt drought-flood alternations. The statistical data show that the waterlogged area of the NLB is 2360 km$^2$, and waterlogging disasters resulted in notably

large losses in the lakeside area. Fig. 4 shows the process of rainfall at Wanglu station and the upper and lower lake levels at Nanyang and Weishan station in 2003. After the ER-SNWDP was completed in 2013, it became the first choice for alleviating local drought via this project. In recent years, the situation of water resource shortages has become increasingly tense in Shandong Peninsula, China. Even in the flood season, local water resources still encounter difficulty in meeting the demand; thus, the supply of water to the region that relies on the ER-SNWDP is expected to be more frequent.

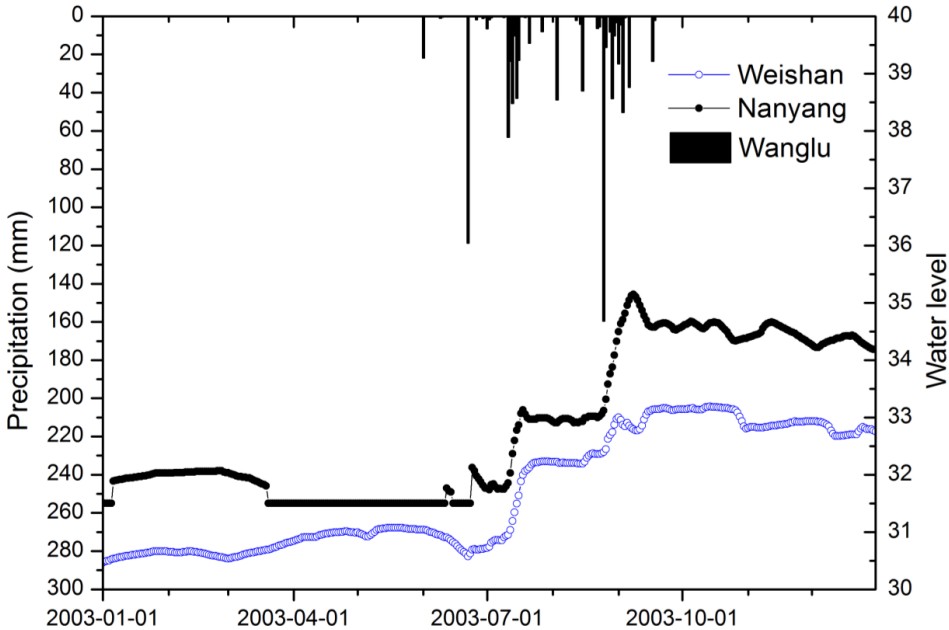

**Figure 4: Rainfall at Wanglu station in NLB and water level variations of the upper lake and lower lake in 2003. The upper lake is dry when the water level of Nanyang is 31.5 m.**

To analyze the influence of water diversion on waterlogging disasters in the lake basin along the ER, this paper set two conditions in which the ER-SNWDP supplies an emergency transfer of ecological water to NL and an emergency transfer of

water to Shandong Peninsula.

(1) An ER-SNWDP emergency requires a supply of ecological water for NL.

We assume that the emergency water transfer ended on August 22, 2003 when rainfall began, which means that water diversion only affected the initial water level of NL under this condition. The rainfall was measured from August 22, 2003 to September 2, 2003. During the water diversion period, the water level of NL maintained a normal storage level, with water

levels in the upper lake and lower lake of 34 m and 32.8 m, respectively. However, we consider that emergency water diversion occurs during the flood season. The initial water levels of the upper lake and lower lake are 34 m and 32.3 m when



rainfall begins in the 2-D model, respectively. This situation is recorded as scenario ②, and the situation in which the waterlogging process occurs without water diversion is recorded as scenario ① (Table 2).

(2) An ER-SNWDP emergency requires a supply of water for Shandong Peninsula, China.

In this condition, the influence of water diversion on waterlogging disasters in the lakeside area of NL under different rainstorm intensities is analyzed. The processes of a 3-day designed rainfall with return periods of 5 years, 10 years and 20 years at six precipitation stations were calculated. During rain events, the ER-SNWDP continues to supply water for Shandong Peninsula, China.

In summary, a total of 8 simulation scenarios were set up as shown in Table 2.

**5.2 Impacts of the ER-SNWDP emergency supply of ecological water for NL on the waterlogging of the NLB**

In the calculation results of scenario ① and scenario ②, the areas with a submerged depth larger than 0.1 m and 0.5 m, respectively, were counted. Table 3 shows that the rainfall from August 22 to September 2, 2003 under the condition of no water diversion caused the flooded area in the lakeside area to reach 1126.59 km$^2$ and the area with a submerged depth over 0.5 m reached 383.68 km$^2$. Statistical data show that the total waterlogging area under the 36.79 m contour line was 1284.21 km$^2$ in 2003. The simulation result is slightly smaller than the survey result because the simulation did not cover the entire year and rain remained in the basin after September 2, 2003. In general, the simulation results can be considered reasonable.

Table 4 shows the increase in the waterlogging area in the lakeside area of NL under the condition of water diversion in the ER-SNWDP compared with that without water diversion. When the phenomenon of abrupt drought-flood alternation occurs, the lake level increase via emergency operation of the ER-SNWDP during the flood season increases the waterlogging intensity in the lakeside area of NL. In the study area, the region with a submerged depth above 10 cm increased by 34.26 km$^2$, which represented an increase of 0.99% compared with that without water diversion. The heavy disaster area with a water depth of more than 50 cm increased by 51.09 km$^2$, which was 13.32% higher than that without water diversion.

Fig. 5 shows the waterlogging distribution in the lakeside area of NL under two scenarios. The comparison in Fig. 5(a) and (b) shows that the water diversion primarily increased the waterlogging area between the Dongyu River and Wanfu River in the western region of NL and had a relatively small impact on the eastern area of NL. The primary reason for this observation is that the western portion of NL is mountainous, whereas the western portion of NL is a low-lying plain. The interaction between NL and the western rivers is stronger than that between NL and the eastern rivers. Therefore, water-receiving lake-basins with plain areas should consider the effects of interbasin water diversion on the waterlogging disaster in. Fig. 6 presents the stage hydrograph for the Makou hydrographic station and the flood discharge of Erji dam for two scenarios. Fig. 6 indicates that emergency water diversion has an obvious influence on the water level of NL during the initial period of rain and the regulation of the Erji dam junction leads to a decrease in the water level difference between the two scenarios. Emergency water diversion increases the initial water level of NL at the beginning of the rainfall event, and the water level of the upper lake in scenario ② first reaches the water level at which the Erji dam begins flood drainage. With the increase in the water level, the discharge of the Erji dam also increases, which satisfactorily adjusts the water level of the upper lake. Affected by the higher water level of the lake in the early stage of water diversion, the time at which the Erji dam junction begins to drain the flood water in scenario ② occurs 4 days earlier than that in scenario ①, and the total amount of flood discharge increased by approximately 249 million m$^3$.





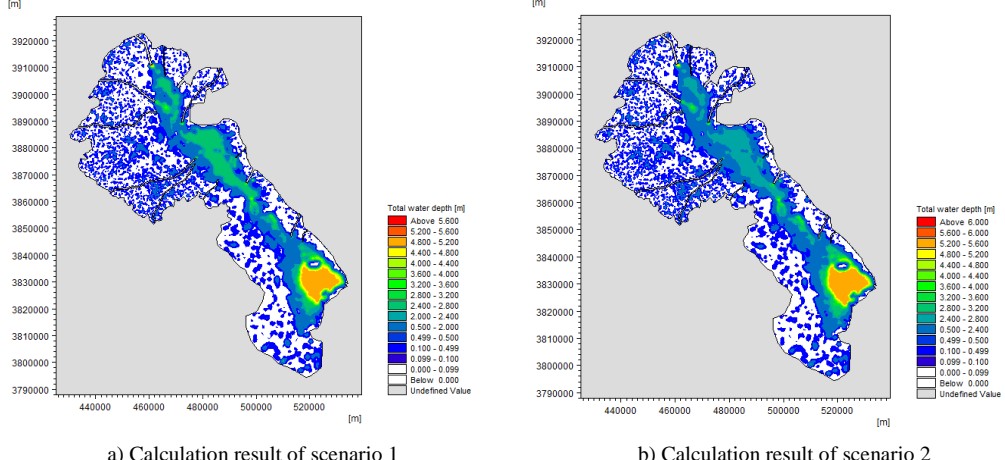

| a) Calculation result of scenario 1 | b) Calculation result of scenario 2 |
|---|---|

**Figure 5: Distribution of waterlogging in the lakeside area of NL.**

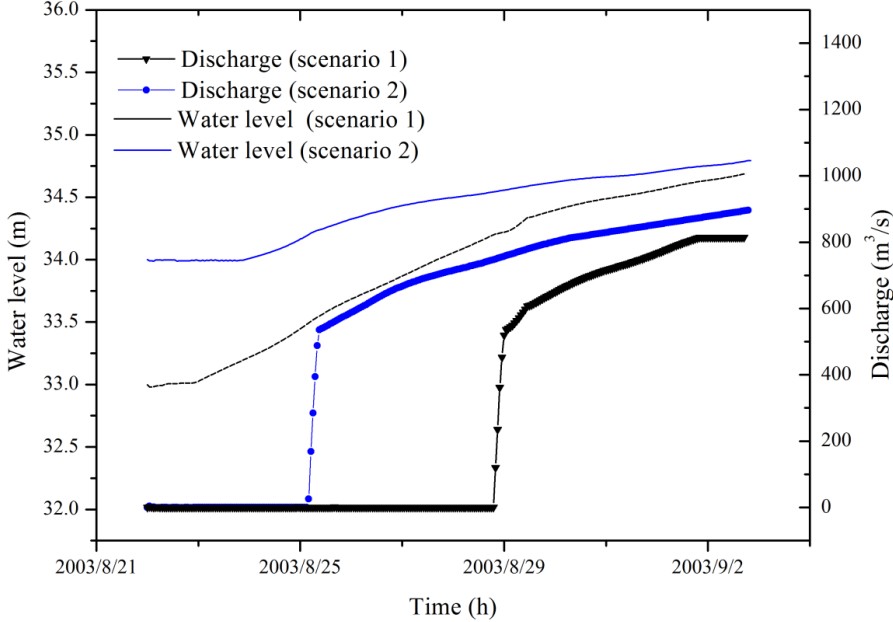

**Figure 6: Water level of Makou station in the upper lake and discharge of flood drainage of the Erji dam from August 22 to September 2, 2003 under scenario 1 (without water diversion) and scenario 2 (with water diversion).**

5 **5.3 Impacts caused by ER-SNWDP emergency water diversion for Shandong Peninsula during waterlogging of NLB.**

Table 5 shows the simulation results of waterlogging in the lakeside area in the case of water diversion and non-diversion by the ER-SNWDP to the Shandong Peninsula when a designed rainfall with 5-, 10- and 20-year return periods occurs in the basin. Emergency water diversion has certain effects on the waterlogged area in the lakeside area of NL. According to the comparison of scenario ③ and scenario ⑥, when the NLB encounters rainfall with a 5-year return period, the areas with a

10 submerged depth over 0.1 m and 0.5 m increase by 22.27 km² and 26.14 km², respectively, under the condition of emergency water diversion.




Fig. 7 illustrates the relative increase in the waterlogging area and its change trend in the lakeside area of NL under three designed rainfall conditions with water diversion and non-water diversion. Graph (a) shows the contrast of the waterlogged area with an inundated water depth above 0.1 m under different rainfalls, and graph (b) shows the waterlogged area with an inundated water depth above 0.5 m. The influence of emergency water diversion on waterlogging in the lakeside area

decreases with increasing rainfall. Affected by the emergency water diversion, the relative increase in the waterlogging area with an inundated water depth above 0.1 m is between 1.5% and 2.8% (Fig. 7(a)), whereas the relative increase in the waterlogging area with an inundated water depth above 0.5 m is between 8.4% and 43.1% (Fig. 7(b)) when a storm with 5-year, 10-year and 20-year return periods occurs in the NLB. The calculated results indicate that emergency water diversion has more obvious effects on the waterlogging area with deeper submerged water.

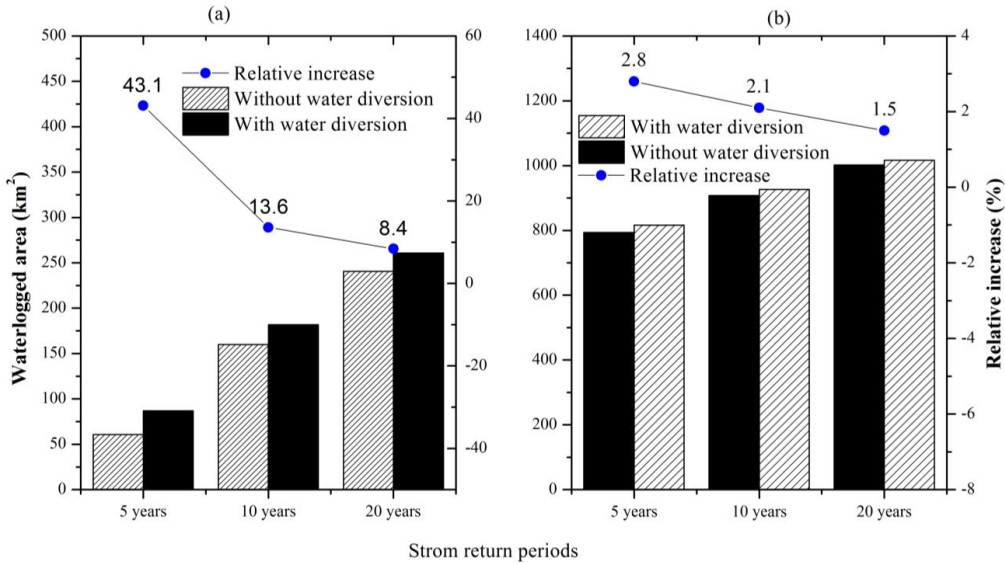

**Figure 7: Changes in the area of waterlogging in the lakeside area affected by water diversion.**

The emergency water diversion of the SNWDP alleviated the drought situation in Shandong Peninsula area and also increased the degree of waterlogging in the NLB under the situation of abrupt drought-flood alternations. The sluices of the Erji dam are the only flood-discharge channels of the upper lake, and the increased water in the upper lake due to water

diversion also enlarged the task of flood discharge in those sluices. Fig. 8 shows the flood discharge process of the Erji dam project under different rainfall levels. When the NLB encountered a storm with 5-year return periods, the upper lake level did not reach the water level necessary to begin drainage without the influence of water diversion. In the case of water diversion, the sluices began to drain the flood after 36 hours of rain, with a total discharge volume of 85 million $m^3$ (Fig. 8(a)). When the NLB encountered a storm with a 10-year return period, under the condition of water diversion, the time at

which the sluices began to drain the flood was the 30th hour, which was e36 hours ahead of the situation of no water diversion (at the 66th hour), and the total discharge volume was 104 million $m^3$ higher than that with no water diversion (Fig. 8(b)). When the NLB encountered a storm with a 20-year return period, under the condition of water diversion, the time at which the Erji dam project began to discharge the flood was the 26th hour, which was 32 hours ahead of the situation with no water diversion (at the 58th hour), and the total discharge volume was 129 million $m^3$ higher than that of no water

diversion (Fig. 8(b)).

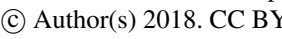



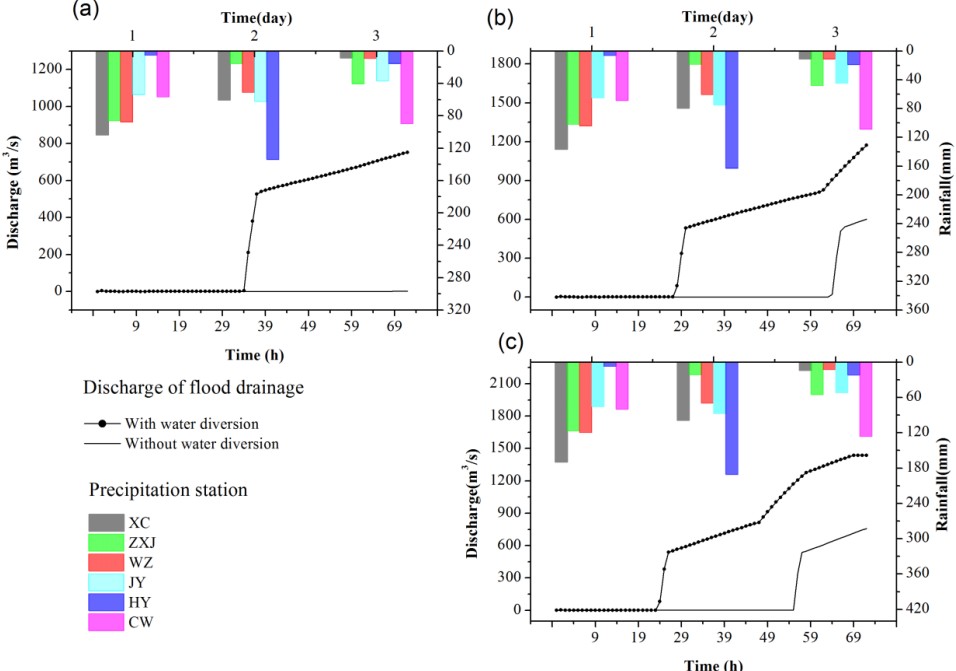

**Figure 8: Changes in the flood discharge of the Erji dam under water diversion with different storm levels.**

## 6 Conclusions and policy implications

The following selected conclusions are presented.

(1) One- and two-dimensional coupled floods and a waterlogging simulation model of the NLB were established to simulate the water diversion of the ER-SNWDP. The MIKE 11 model was applied to simulate the flow movement of water in the water diversion channel and the tributaries of NL, and the MIKE 21 model was applied to simulate the waterlogging in the lakeside area and the water flow in NL.

(2) The ER-SNWDP emergency water transfer to NL increases the risk of waterlogging damage in the lakeside area if it

occurs simultaneously with abrupt drought-flood alternations. The increased water level caused by water diversion decreases the efficiency of waterlogging drainage, and as a result, the waterlogged area with an inundated water depth above 0.1 m increased by 0.99% and an inundated water depth above 0.5 m increased by 13.32%. The flood-discharge time of the Erji dam increased by 4 days, and the total discharge volume increased by 249 million $m^3$ during the simulation.

(3) The ER-SNWDP emergency water transfer to Shandong Peninsula raised the water level of NL, which acted as a

regulation and storage lake. Compared with the no water transfer situation, the waterlogging areas in the lakeside area increased when NL encountered a storm with 5-year, 10-year and 20-year return periods under water diversion. The calculation results show that water diversion has a more obvious effect on the waterlogging area with an inundated water depth above 0.5 m, with the area increasing by 8.4-43.1%. The total volume of flooding discharged by the Erji dam also increased. In addition, we found that with the increase in rainfall intensity, the influence of water diversion on the lakeside

area in the NL inundated area gradually decreased and the water transfer had more serious effects in a rainstorm with a lower return period.

Certain implications for the management of interbasin water diversion and the lake basin along the route of water diversion project are presented below.





(1) For a complicated flood control and drainage system that contains a number of hydraulic structures, such as pumping stations, sluices, and embankments, flood movement behavior regulations must be implemented and flood disaster losses must be reduced by establishing a one- and two-dimensional coupled hydrodynamic model to accurately simulate the flow process and a clear movement direction of flooding.

(2) For a long-distance interbasin water transfer project, due to the large difference between high and low precipitation in the water-supply and water-receiving areas combined with global climate change and hydrological uncertainty, strengthening the analysis of emergency water diversion influence on flood control and drainage is not only necessary for scientific management of the inter-basin water transfer project but also conducive to realizing the expected benefits and reducing the negative effects of the project.

(3) To reduce the interbasin water transfer project effect on waterlogging in the water receiving area, we can take steps based on the following factors. First, additional emphasis should be placed on planning projects, increasing the waterlogging drainage pumping stations and enlarging the capacity of the flood discharge buildings in the water-receiving basin. Second, the hydrological forecasting and early warning ability should be improved and the accuracy and forecast period of the rainfall event should be increased and so to stop the water diversion or lower the water level of the lake before the rainstorm.

With respect to the directions for future research, we can expand on the following several aspects: (1) considering the vulnerability of hazard-affected bodies, populations as well as GDP and other information should be considered to more accurately reflect waterlogging disasters in the research area; and (2) a case study analysis of the balance between water transfer risk and water resource benefits should be conducted.

*Data availability.* All data except for the DEM of the lakeside and Nansihu Lake in 2013 were acquired by the authors. Data
except for the DEM can be requested by email from the authors wangzz77@163.com.

*Competing interests.* The authors declare that they have no conflict of interest.

*Author contribution.* Kun Wang prepared the manuscript with contributions from all co-authors. Kun Wang and Zongzhi Wang developed the model and Zongzhi Wang designed the scenario. Kelin Liu, Liang Cheng and Lihui Wang guided and participated in the modelling, Kelin Liu and Liang Cheng dealt with the boundary conditions of the model. Ailing Ye made
the electric artworks and words processing.

*Acknowledgments.* This study was financially supported by the National Key Research and Development Program of China (2017YFC0403504) and the National Science Foundation of China under grant No. 51479119 and 51579064.

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



**Appendix**

**Table 1 Statistical evaluation of model performance for water level simulations at selected gauging stations for 2007 and 2008 flood**
5   **events.**

| Gauging station | NSE | |
| --- | --- | --- |
| | 2007 | 2008 |
| Nanyang | 0.72 | 0.65 |
| Makou | 0.69 | 0.76 |
| Erji Lake (downstream) | 0.67 | 0.98 |
| Weishan | 0.82 | 0.99 |

**Table 2 Computational scenario setting**

| Sr. no. | Initial water level of NL | Rainfall | Whether the ER-SNWDP works during the simulation |
| --- | --- | --- | --- |
| ① | Actual water level on August 22, 2003 | Actual daily rainfall from August 22 to September 2, 2003 | NO |
| ② | 34.0 m in upper lake, 32.3 m in lower lake | | |
| ③ | Actual water level on August 22, 2003 | Designed storm with return periods of 5 years | NO |
| ④ | | 10 years | |
| ⑤ | | 20 years | |
| ⑥ | 34.0 m in upper lake, 32.3 m in lower lake | 5 years | YES |
| ⑦ | | 10 years | |
| ⑧ | | 20years | |





**Table 3 Results of waterlogging in the lakeside area of NL**

| Sr. no. | Water depth of NL | | Area with an inundated depth above 0.1 m | | Area with an inundated depth above 0.5 m | |
|---|---|---|---|---|---|---|
| | Average (m) | Max (m) | Total area (km$^2$) | Area ratio (%) | Total area (km$^2$) | Area ratio (%) |
| ① | 2.47 | 5.96 | 1126.59 | 32.51 | 383.68 | 11.07 |
| ② | 2.80 | 6.14 | 1160.85 | 33.50 | 434.77 | 12.55 |

**Table 4 Increment of the waterlogging area in the lakeside area of NL**

| Contrastive analysis | Area with an inundated depth above 0.1 m | | | Area with an inundated depth above 0.5 m | | |
|---|---|---|---|---|---|---|
| | Increment (km$^2$) | Relative increase (%) | Area ratio increase | Increment (km$^2$) | Relative increase (%) | Area ratio increase |
| Variation | 34.26 | 3.04 | 0.99 | 51.09 | 13.32 | 1.47 |

**Table 5 Results of waterlogging simulations of the lakeside area under different scenarios.**

| Sr. no. | Water depth of NL | | Area with an inundated depth above 0.1 m | | Area with an inundated depth above 0.5 m | |
|---|---|---|---|---|---|---|
| | Average (m) | Max (m) | Total area (km$^2$) | Area ratio (%) | Total area (km$^2$) | Area ratio (%) |
| ③ | 1.86 | 5.23 | 793.75 | 22.91 | 60.63 | 1.75 |
| ④ | 2.03 | 5.87 | 907.85 | 26.20 | 159.98 | 4.62 |
| ⑤ | 2.41 | 6.31 | 1002.05 | 28.92 | 240.54 | 6.94 |
| ⑥ | 2.10 | 5.87 | 816.02 | 23.55 | 86.77 | 2.50 |
| ⑦ | 2.26 | 6.10 | 926.40 | 26.73 | 181.73 | 5.24 |
| ⑧ | 2.17 | 6.08 | 1016.68 | 29.34 | 260.76 | 7.53 |