# Peer review of "Impacts of the emergency operation of the South-to-North Water Diversion Project's eastern route on flooding and drainage in the water-receiving area: An empirical case from China"

_Natural Hazards and Earth System Sciences, 2018_

## Referee Comment (RC1) · Anonymous Referee #1 · 17 Oct 2018

The effect of interbasin water diversion and flood management in a large lake basin is a very important and hot topic. In this study, authors constructed a flooding and waterlogging simulation model for assessing the effects of the South-to-North Water Diversion Project's eastern route on flooding and drainage Nansi Lake, a water-receiving area of the water diversion project in China. The issue discussed in this paper is meaningful. I would like to recommend its publication after solved the following problems. 1. Please kindly polish the language. some sentence too long, for example "Therefore, simulations of the flood and waterlogging process in the lakeside areas under the condition of

emergency water diversion by the SNWDP and analyses of the impacts of emergency water diversions on flood and waterlogging drainage characteristics and the scheduling of flood control and drainage projects must be performed to strengthen the scientific scheduling of water diversion projects and flood control projects in water-receiving regions. " and there are grammatically mistakes. I would like to recommend that authors used short sentences to replace too long sentence like this one.

2. The abstract can be more informative by highlighting the significance and novel contribution of this research. Moreover, key findings of this paper must be included in the abstract.

3. Please check the reference style according to journal requirements. In addition, please avoid multiple references. It is not recommended to cite over three references in one sentence.

4. The figures 3 and 5 are very beautiful. But it is a little confusing. Authors need to introduce the directions more clearly.

5. In the introduction, authors need to make the innovations (i.e., the research gap this paper fills) more clear.

6. In the lines 20-25, authors demonstrate three key issues that they aim to solve. Authors need to give answers to these questions in the conclusions.

7. Line 41: Wrong format of the citation to Bisht et al. 2016.

8. Line 24-25, page 10 "Considering emergency water diversion occurs during the flood season is scenario 2", while the ER-SNWDP doesn't work during the simulation in scenario 2 according to the table 2? Please modify table 2.

9. Line 25, page 11 the eastern portion of NL is mountainous?

10. Line 6-7, page 13. Text descriptions of both Figure a and Figure b are reversed. Please correction.

[Figure]

11. Line 25, page13. Should it be figure 8(c) instead of figure 8(b)?

---

## Referee Comment (RC2) · Siebert (Referee) · 28 Nov 2018

The authors present an interesting study on the effects of a giant water diversion system, installed in China to transfer water from Yangtze River to provinces/regions further in the north, e.g., water scarce Shandong Peninsula. Due to the continuous water shortage in the receiving areas, the channel is operating continuously, though there are considerable amounts of rainfall along the way. The study explains, how local weather phenomenon interacts with an almost transcontinental water diversion scheme. Since it is not my primary field of research, it is difficult to evaluate whether the presented

study reveals some new concepts, tools or methods. However, the scientific methodology and input data are valid, the scenario selection is relevant and fits the requirements of the study. And finally, the authors reach substantial conclusions and clearly show the hazard of a water transfer project like the SNWDP.

However, the manuscript is generally and particularly in the introduction part weak in English. Not only, but seriously influenced by that comes the second, much larger weakness. The presentation of the general setting and conditions of the study, the problem and why certain technical/hydraulic activities are done to manage/control water flow to, within and out of the Nansi Basin are not clearly described. The introduction is immature and also the presentation of the scenarios must be seriously improved. Particularly the wording is difficult, but also the description of figures lacks sufficient details. A lot of (at least to me) unknown technical phrases are used instead of international terminology. The figures have to be improved to meet journal requirements (e.g. include numbering of each figure, descriptions must be improved to allow understanding of each figure by itself). Finally, I strongly suggest to involve a native speaker. Doing so, the manuscript will surely meet the journal's requirements to be published in NHESS.

In addition to my general comments some specific in the following:

Introduction. The introduction is written in a way, that one can follow the author's arguments only, if one has already considerable knowledge about the ER-SNWDP. Too many things are just mentioned and terms are not described in a sufficient way. In particular, the following issues are difficult and not understandable:

Page 1 L15: What is a waterlogging simulation and wouldn't be flood simulation the better term? Which interactions are meant? L28: a map showing the most important geographical places, including contour lines and information about elevations would be required. Furthermore, some climatic characterisations (annual rainfall as colour-code) would help understanding the general conditions. L29: development of what?

L30: which lake and what does the lakeside area mean? It is somehow indicated in Fig. 1, but what are the borders, how are they defined, etc. L31: what does "blocking of the rising lake level" means and what are subsequent "waterlogging disasters"? The entire process chain is not clear to me. L34: There is no reason to distrust the publication of Webber et al., however, aren't there better information about intended water volumes from official authorities/reports, etc.? L34: what is the "water diversion period"? L35: does the project stops and runs in intervals? And why is it obvious, that water tables rise? If water is consumed the same amounts as brought into the basins, nothing happens. The operational scheme does not become clear. L37: what is "emergency water diversion"? I can imagine the possible meaning, but it has to be clarified. Why did water supply occurred sung flood period? L38: The sentence: "Furthermore, considering the rigid demand for water resources caused by rapid economic and social development, extreme hydrological events caused by environment changes have increased along the ER-SNWDP." is unclear.

Page 2 L3-6: Please rephrase the sentence the way you split it at least into 2-3. In the current version and with the amounts of questions, which rose at the passage before, the reason to do these simulations is still not clear. L6: which situation? L7-8: if there are studies, why is there a gap in the literature? L12: again, which gap? L14: why is it an important storage node? This is important to state. L15: again, the authors base on preliminary knowledge: it is not introduced, that the lake is separated into 2 halves by a dam. How can one know about the upper and the lower lake since the lake is just mentioned as the Nansi Lake. L17: what is the phenomenon, where explained, etc. L17-L19: I have difficulties to follow the argumentation, too many things are written in a row without making it clear. L20: as far as I understood, NLB is a flow-through basin, where water tables are fluctuating due to natural floods and droughts and additionally due to water pumped into the basin, which will be pumped out as it comes in. Why is it a water receiving area? L21: what are "structures under water diversion"? L34: what is "the world's annual water intake"? The world's water budget is closed.

Page 3 L24-26: please explain that corresponding effects by using the correct hydrological terminology. L30: what is the "disaster risk", which exact disaster(s) is/are meant?

Page 4: L5: why "land"? L15: the elevation is in respect to what: main sea level? L16: please explain slope in degrees. L20: please explain what you mean by: "waterlogging in lakeside areas can no longer directly drain into rivers and lakes" and how does it work: "The waterlogging water is primarily carried into rivers and NL by pumping stations in the lakeside area." L22: What does it mean the rainfall is concentrated? How high is the rate?

Page 5 Fig. 1: Please add (a), (b) and (c) to the single figures, as it is usual. In the upper right map: why is China's SE not continuously bordered? What does the inset map express and why is the international border southwards dashed? In the upper left map: please name the rivers In the lower map: please indicate the location of the dam, Description of Fig. 5: delete "The logo of Copernicus publications"

L4: what is Yangzhou? L5-8: please indicate the location of sluices and dams in Fig. 1, otherwise it is impossible to follow L8: what is the 1st phase, how is it defined, when does it end, etc. L8: what is the Liangji River mouth, 24.6 km up- or downstream? L9: "project operation": during which phase? Water table in respect to mean sea level?

Page 6: L1-3: "project operation": during which phase? Water table in respect to mean sea level? L9: river channel bathymetry of which river? L11: please indicate the location of rainfall stations in map of Fig. 1 L13: please indicate the location of water level recording stations in Fig. 1 L17: what are "hydraulic engineering data"?

Page7 Fig. 2: where is MIKE FLOOD integrated into the workflow? L8-9: Equations are usually cited in the text.

Page 8 L5: again, elevation reference is missing L6 1000 pump stations are situated in the lakeside area, are they taking water out of the lake or from the rivers? L18-19:

Equations should be cited in the text. L32: where is the model area and how was it chosen? L32: If I understand it right, lakeside area is outlined by the 36.79 m contour line. How was that contour line derived? Does the DEM resolve elevation in 1cm steps? Above, it is given; the area given by that contour line has an area of 3969 km2, now it is 4750 km2 large, which is almost $\frac{1}{4}$ larger. L35-36: why only 5 out of the 7 stations?

Page9 L1-6: The entire passage is not clear, particularly due to the use of unusual terms like "into-lake river". L16: I do not understand, why the Lake receives a roughness coefficient?

Page 10 L10: loss of what? L10: The information concerning the locations of the stations is missing to follow argumentation L18: not the paper, but the study, please change. L19: please explain "ecological water" and what is the "emergency transfer" L22: I assume the "emergency transfer" follows some kind of a pre-set protocoll, which means, it must be possible to request its starting and ending time from the SNWDP operating authorities. L22: rainfall where and why does the diversion now only affects NL? L25: in line 22 it is stated emergency diversion ended, here you state it continues... L26: what is the flood season, is it somehow restricted?

Page 11: L1-2: what is "the situation"? L4: what is "this condition"?

Unfortunately, the entire paragraph is not very clear. To me it stays unclear, when wis which scenario calculated and why (due to which conditions). It is a general difficulty in the entire paper. Due to unclear terminology, even headlines like 5.2 are not clear and it is impossible to follow the argumentation.

Page 12 L1-2 I don't understand the difference between scenarios 1 and 2, different initial water tables? If yes, what was the water table of Scenario 1? Table 2 is not clear, please re-organize it and give exact numbers L4-7: Here, for the first time, the reason for the study is clearly described and one can follow the intention of the authors. This should be integrated into the introduction, of course in a different way, but that's the reason for the study, I guess. L4-7: indicate the scenario number behind each

pre-condition and refer to table 2 directly, not at the next paragraph. L8: prevent one-sentence paragraphs.

L10: why are areas of 0.1 and 0.5 considered? L20: 0.99% of what? L25: EASTERN portion of the lake is mountainous... L26: the interaction is either given or not. There is no option to interact larger or less. Please describe what you want to say differently: e.g. The influence of...results in increased... L36: compared to what, scenario 1? Please describe sharply and refer to fig. 6.

Page 13 L6-7: please check, which one is correct: Figure 7 a and b or the reference in the text, in the moment it is switched. Fig. 7: Please explain a and b in Figure description L9-11: again, please be very clear in describing the effects of with/without active diversion. L18: in Fig. 8a it doe not look like 36 hours, more like 31 hours. Wrong figure? L20-25: please use some different descriptions for time: 30th hour is very uncommon and "e36 hours" is unclear to me. L25 reference is wrong, it is fig. 8c

Page 14: Fig. 8: please explain a, b, c in the figure description.

---

## Author Comment (AC1) · 3 Jan 2019

Responses to Referee' comments Referee #1: This is a very interesting paper. The effect of interbasin water diversion and flood management in a large lake basin is very important and hot topic. In this study, authors constructed a flood and waterlogging simulation model for assessing the effect of the South-to-North Water Diversion Project's eastern route on flooding and drainage Nansi Lake, a water-receiving area of the water diversion project in China. The model and the problem discussed in this paper were very useful and meaningful. I would like to recommend its publication. The

following issues need to be solved well before its publication. 1. Please kindly polish the language. For example, "Therefore, simulations of the flood and waterlogging process in the lakeside areas under the condition of emergency water diversion by the SNWDP and analyses of the impacts of emergency water diversions on flood and waterlogging drainage characteristics and the scheduling of flood control and drainage projects must be performed to strengthen the scientific scheduling of water diversion projects and flood control projects in water-receiving regions. "There are grammatically mistakes in this sentence. I would like to recommend that authors used short sentences to replace too long sentence like this one. Answer: The language in this paper has been improved. Thanks a lot. 2. The abstract can be more informative by highlighting the significance and novel contribution of this research. Moreover, key findings of this paper must be included in the abstract. The abstract is revised as follow: "The water levels of lakes along the eastern route of the South-to-North Water Diversion Project (ER-SNWDP) are expected to rise significantly and subsequently affect the process of flood control and drainage in those lake basins. However, few studies have focused on the impacts of the interbasin water diversion on the flood control and drainage of water-receiving areas at a lake basin scale. Using MIKE software, this paper builds a coupled hydrodynamic model to address the existing literature gap on the impacts of the interbasin water diversion on the process of flood control and drainage in a water-receiving lake basin, and it considers many types of hydraulic structures in the model. First, a flood and waterlogging simulation model was constructed to simulate the complex movement of the transferred water, waterlogging water in the lakeside area surrounding Nansi Lake (NL), and water in NL and its tributaries. The ER-SNWDP was also considered in the model. Second, the model was calibrated and verified with measured data, and the results showed that the model is efficient and presents a Nash-Sutcliffe efficiency coefficient (NSE) between 0.65 and 0.99. Third, the process of flood and drainage in the lakeside area of NL was simulated under different water diversion and precipitation values. Finally, the impacts of emergency operations of the ER-SNWDP on flood control and waterlogging

drainage in the lakeside area of NL were analyzed based on the results from the proposed model, and some implications are presented for the integrated management of the interbasin water diversion and the affected lakes." 3. Please check the reference style according to journal requirements. In addition, please avoid multiple references. It is not recommended to cite over three references in one sentence. Answer: We checked carefully and deleted some unimportant references in the manuscript. Thanks a lot. 4. The figures 3 and 5 are very beautiful. But it is a little confusing. Authors need to introduce the directions more clearly. Answer: We added a detailed description for all figures in the revised manuscript. Thanks a lot. 5. In the introduction, authors need to make the innovations (i.e., the research gap this paper fills) more clear. Answer: The introduction has been rewritten as recommended by the two experts, to the innovations more clear. 6. In the lines 20-25, authors demonstrate three key issues they aim to solve. Authors need to give answers to these questions in the conclusions. Answer: We answered the proposed issues in the conclusions one by one as: (1) In order to clarify the impacts that interbasin water transfer on the water-receiving area, this study using MIKE model to simulate the flood and waterlogging in NLB. One- and two-dimensional coupled floods and a waterlogging simulation model of the NLB were established to simulate the water diversion of the ER-SNWDP. The MIKE 11 model was applied to simulate the flow movement of the water in the water diversion channel and the tributaries of NL. The MIKE 21 model was applied to simulate the waterlogging in the lakeside area and the water flow in NL. The verification results show that the presented method can effectively simulate the flood and waterlogging process in NLB under the effect of the ER-SNWDP. (2) The ER-SNWDP emergency water transfer to NL increases the risk of waterlogging damage in the lakeside area if it occurs simultaneously with DFAA. The increased water level caused by water diversion decreases the efficiency of waterlogging drainage. As a result, the waterlogged area with an inundated water depth above 0.1 m increased by 0.99% and an inundated water depth above 0.5 m increased by 13.32%. The flood-discharge time of the Erji dam increased by 4 days and the total discharge volume increased by 249 million m3

during the simulation. (3) The ER-SNWDP emergency water transfer to Shandong Peninsula raised the water level of NL which acted as a regulation and storage lake. Compared with the no water transfer situation, the waterlogging areas in the lakeside area increased when NL encountered a storm with 5-year, 10-year and 20-year return periods under water diversion. The calculation results show that water diversion has a more obvious effect on the waterlogging area with an inundated water depth above 0.5 m. The total volume of flood discharged by the Erji dam also increased. 7. Line 41: Wrong format of the citation to Bisht et al. 2016. Answer: The citation is revised accordingly, thanks a lot. 8. Line 24-25, page 10 "Considering emergency water diversion occurs during the flood season is scenario 2", while the ER-SNWDP doesn't work during the simulation in scenario 2 according to the table 2? Please modify table 2. Answer: We carefully checked this part of content and revised the mistake as follow: "Considering the flood control safety of NLB, we assume that the emergency water transfer stopped at the beginning of rain, on August 22, 2003." Thanks a lot. 9. Line 25, page 11 the eastern portion of NL is mountainous? Answer: We revised the sentence as follow: "Comparing with the eastern part of NLB, the terrain of the western NLB is lower and flat, and the western NLB has a greater impediment to drainage water into NL when the water level of NL is high." Tanks a lot. 10. Line 6-7, page 13. Text descriptions of both figure a and figure b are reversed. Please correction. Answer: We corrected the mistakes in the manuscript. Thanks a lot. 11. Line 25, page13. Should it be figure 8(c) instead of figure 8(b)? Answer: We revised it in manuscript. Thanks a lot.

Please also note the supplement to this comment:
https://www.nat-hazards-earth-syst-sci-discuss.net/nhess-2018-216/nhess-2018-216-AC1-supplement.pdf

[Figure]

**N**

| VALUE | |
|---|---|
| 115°0'0"E | |
| 116°0'0"E | |
| 117°0'0"E | |

36°0'0"N

Liangji River

Guangfu River

Si River

Yunju River

Xiaoyi River

Zhuzhaoxin River

Dasha River

Zhushui River

Barna River

Wanfu River

Beisha River

ChengguoRiver

Dongyu River

Fuxing River

Shengli River

Dasha River

Shizi River

Taihangdi River

35°0'0"N

**Legeng**

- ⊗ Sluice
- ● Rainfall station
- ▲ Water level station
- ▬ Erji Dam
- ── ER-SNWDP
- ‒‒ Levee
- ── River
- ▆ Nansi Lake
- ☐ Lakeside area of NL

**DEM**

**\<VALUE\>**

| | |
|---|---|
| | 6.00 - 38.00 |
| | 38.00 - 70.00 |
| | 70.00 - 150.00 |
| | 150.00 - 300.00 |
| | 300.00 - 647.00 |

0  12.5  25      50      75      100
km

34°0'0"N

115°0'0"E        116°0'0"E        117°0'0"E

**Fig. 1.**

[Figure]

**Fig. 2.**

[Figure]

Fig. 3.

**Supplement:**

**Impacts of the emergency operation of the South-to-North Water Diversion Project's eastern route on flooding and drainage in the water-receiving area: An empirical case from China**

Kun Wang[1,2], Zongzhi Wang[2], Kelin Liu[2], Liang Cheng[2], Lihui Wang[3], Ailing Ye[2,3]

[1]College of water Conservancy and Hydropower Engineering, Hohai University, Nanjing, 210098, China
[2]State Key Laboratory of Hydrology-Water Resources and Hydraulic Engineering, Nanjing Hydraulic Research Institute, Nanjing 210029, China
[3]College of Civil Engineering, Fuzhou University, Fuzhou 350002, China

*Correspondence to*: Zongzhi.Wang (wangzz77@163.com)

**Abstract.** The water levels of lakes along the eastern route of the South-to-North Water Diversion Project (ER-SNWDP) are expected to rise significantly and subsequently affect the process of flood control and drainage in those lake basins. However, few studies have focused on the impacts of interbasin water diversion on the flood control and drainage of water-receiving areas at the lake basin scale. Using MIKE software, this paper builds a coupled hydrodynamic model to address the existing literature gap on the impacts of interbasin water diversion on the process of flood control and drainage in a water-receiving lake basin, and it considers the many types of hydraulic structures in the model. First, a flood and waterlogging simulation model was constructed to simulate the complex interactionsmovement betweenamong of the transferred water, waterlogging water ofin the lakeside area surrounding around Nansi Lake (NL), and the water in NL and its tributaries. The ER-SNWDP was also considered in the model. Second, the model was calibrated and verified with measured data, and the results showed that the model is efficient and presents a Nash-Sutcliffe efficiency coefficient (NSE) between 0.65 and 0.99. Third, the process of flood and drainage in the lakeside area of NL was simulated under different water diversion and precipitation values. Finally, the impacts of emergency operations of the ER-SNWDP on flood control and waterlogging drainage in the lakeside area of NL were analyzedanalysed based on the results from the proposed model, and some implications are presented for the integrated management of the interbasin water diversion and the affected lakes.

**1 Introduction**

Interbasin water diversion is a useful approach to solving the spatial unevenness of water resources, but at the same time ithowever, it makes the water cycle of the water-receiving area more complicated and brings a great challenge to integrated water management of the water-receiving area and the water diversion project (Matete and Hassan, 2006; Webber et al., 2017). The key to the long-term healthy operation of interbasin water transfer project is to clarify the influence of interbasin water transfer on the water-receiving area. In the last decades, several authors have revealed and discussed the impacts of the interbasin water transfer on water-receiving area from many perspectives, mainly including water quality, water resources and water ecosystem, and some implicationssuggestions of scientific management of the water-receiving area and water diversion project were proposed (Cole et al., 2011; Emanuel et al., 2015; Zhuang, 2016). A large quantity of transferred water will not only In addition to changingchange the water quality, the water environment and other hydrological characteristics of the water-receiving areas, a large quantity of transferred water will but also have an impact on flood control and waterlogging drainage in some water-receiving areas (Gupta and Zaag, 2008; Liang et al., 2012). Floods and waterlogging are one of the main natural disasters in terms of losses of human life and economic damages, and, flood control and drainage are the top priorities of watershed management(Arrighi et al., 2013; He et al., 2018; He et al., 2018b). However, to the best knowledge of the authors, there are very few studies on the impacts thatof interbasin water transfer on the flood control and drainage in water-receiving area from watershed perspective.

The eastern route of the South-to-North Water Diversion Project (ER-SNWDP) links Gaoyou Lake, Hongze Lake, Luoma Lake, Nansi Lake and Dongping Lake with 13 pump stations that transfer water from the downstream Yangtze River to the Huang-Huai-Hai Plain and Shandong Peninsula (Fig.1a and b). According to the comprehensive plan of the SNWDP, (1) the first phase planning (before 2030) of the eastern route is designed to transfer 8.9 billion $m^3$ water annually, and

5  approximately 7 billion $m^3$ is expected to be consumed in the above five lake basins and the route; (2) the water diversion period covers the non-flood season (October to next May), and the water diversion will be stopped in the rest of the time (Bureau of South to North Water Transfer of Planning, Designing and Management, Ministry of Water Resources, 2003). As the water-receiving areas and the transmitting channels of the ER-SNWDP, the five lakes are used to store and regulate water resources, and the water level of the five lakes are significantly raised when the project is operating.

10  The raised lake level will impede the flood control and waterlogging drainage during a rainstorm, and especially during the flood in the low-lying lake basin.

Otherwise, the Shandong Peninsula has suffered from severe drought and the water supply cannot meet the water demand even in the flood season  (June to October) for four consecutive years since the eastern route operated in 2013. Emergency water diversion, that is, the water  transfer through the water diversion

15  project in flood seasons to alleviate water shortages in the water-receiving areas, has been performed many times to supply water for Shandong Peninsula. Furthermore, considering the rigid demand for water resources caused by the rapid socioeconomic development , more frequent water transfers are expected in the flood season (Guo et al., 2018). Meanwhile, extreme  rainfall events caused by  climate changes have increased in eastern China (Liu et al., 2015). Thus, the

20  probability of rainstorms during water diversion period in these lake basins will increase. In order to strengthen the scientific scheduling of flood control projects in water-receiving areas and water diversion projects, it is necessary to clarify the influence law of water diversion on flood control and drainage in these lake basins.

Based on data availability and regional distribution, the Nansi Lake Basin (NLB), a flood prone area, is chosen as the research area in this study (Fig.2). The reason why we choose the NLB is that the NL  is an important

25  storage node of the ER-SNWDP for NL is the largest freshwater lake in northern China and has the largest water storage capacity among the lakes along the ER-SNWDP (Zhang, 2009). Otherwise, the NLB has a history of frequent flooding and waterlogging disasters due to the low  drainage capacity  in the geomorphic low-lying area around NL (lakeside area) . NL is a storage pond of the ER-SNWDP

30  , and the flow of water transferred into Nansi Lake is 200 $m^3$/s and the outflow is 100 $m^3$/s. A large quantity of water assumed in the Nansi Lake Basin, so it is a water receiving area. According to the overall plan of the SNWDP, the water level of the upper lake is expected to rise by 50 cm and the lower lake is expected to rise by 70 cm during the project operation period. The lake level raised by the ER-SNWDP will decrease the drainage efficiency of pump stations and hinders flood discharge of rivers in lakeside area, and then influence the flood control and waterlogging

35  drainage of NLB.

Therefore, this paper proposes a model that integrates MIKE 11, MIKE 21 and MIKE FLOOD to simulate the flood and waterlogging process in NLB under the condition of emergency water diversion by the ER-SNWDP. And then, simulate the flood and waterlogging under different rainstorm and water diversion condition. Finally, according the simulation results to quantify the impacts of the interbain water transfer on the flood

[revised manuscript text omitted]
 refers to the area with a ground elevation (above sea level)  below 36.79 m  around the lake,

20   and  the ground slope in this area  is between 0.0029 ° and 0.0057 ° (Tian et al., 2013; Wang et al., 2010). A total of 53 rivers flow into NL, and 11 of them have a drainage area greater than 1000 km$^2$. Due to the low height and gentle slope of the riverbed, the rivers in the lakeside area have strong interactions with NL. Flood control embankments have been built on both sides of the main inflow channels and around NL to prevent flooding from entering the lakeside area. Due to the low-lying terrain and the construction of flood control embankments, waterlogging in

25  lakeside areas cannot  drain itself into rivers and NL. The waterlogging water in the lakeside area  mainly pumped into rivers and NL  through pumping stations . However, the existing pumping stations in this region cannot resist the rainstorm waterlogging occurred every five years, some pumping stations even cannot resist rainstorm waterlogging occurred every three years. When encountering heavy rainstorm, the water cannot be discharged in time, therefore the NLB is a flood prone area in the history

30  .

[Figure]

**Figure 1̶2̶: Location of the study area, with the positions of rainfall and water level gauges, rivers and hydraulic projects considered in the study River system and topographic map of Nansi Lake Basin** ̶S̶k̶e̶t̶c̶h̶ ̶m̶a̶p̶ ̶o̶f̶ ̶t̶h̶e̶ ̶e̶a̶s̶t̶e̶r̶n̶ ̶r̶o̶u̶t̶e̶ ̶o̶f̶ ̶t̶h̶e̶ ̶S̶o̶u̶t̶h̶-̶N̶o̶r̶t̶h̶ ̶W̶a̶t̶e̶r̶ ̶T̶r̶a̶n̶s̶f̶e̶r̶ ̶P̶r̶o̶j̶e̶c̶t̶ ̶a̶n̶d̶ ̶N̶a̶n̶s̶i̶ ̶L̶a̶k̶e̶ ̶B̶a̶s̶i̶n̶T̶h̶e̶ ̶l̶o̶g̶o̶ ̶o̶f̶ ̶C̶o̶p̶e̶r̶n̶i̶c̶u̶s̶ ̶P̶u̶b̶l̶i̶c̶a̶t̶i̶o̶n̶s̶.

5   The ER-SNWDP transfers water from Yangzhou City to NL and is divided into two sections in the NLB: water entering the lower lake at the Hanzhuang sluice and ̶w̶a̶t̶e̶r̶ ̶e̶n̶t̶e̶r̶i̶n̶g̶ the Linjia dam sluice. A pumping station has been built at the secondary dam to lift water from the lower lake to the upper lake,̶.̶ ̶a̶n̶d̶ ̶tThe Changgou pumping station is built at 24m south of Liangji River Estuary in the upper lake, ̶w̶i̶l̶l̶ ̶b̶e̶ ̶b̶u̶i̶l̶t̶ ̶a̶t̶ ̶t̶h̶e̶ ̶h̶i̶g̶h̶e̶r̶ ̶l̶a̶k̶e̶ ̶2̶4̶.̶6̶ ̶k̶m̶ ̶t̶h̶e̶ ̶L̶i̶a̶n̶g̶j̶i̶ ̶R̶i̶v̶e̶r̶ ̶m̶o̶u̶t̶h̶ to ̶s̶e̶n̶d̶ transfer water to the Shandong Peninsula. According to the first phase planing of the ER-̶-SNWDP, the discharge that enters the

10   lower lake ̶o̶c̶c̶u̶r̶s̶ ̶a̶tis 200 m$^3$/s, and 5/8 of this amount will be pumped into the upper lake. When the water diversion project is running̶D̶u̶r̶i̶n̶g̶ ̶p̶r̶o̶j̶e̶c̶t̶ ̶o̶p̶e̶r̶a̶t̶i̶o̶n̶, the water level (in respect to mean sea level) of the lower lake will reach 32.8 m, which is 0.70 m higher than the mean annual water level. The project ̶a̶l̶s̶o̶ transfers water from the upper lake to the north at the flow rate of 100 m$^3$/s. The water level of the upper lake maintains a normal storage level of 34 m, which is 0.48 m higher than the mean annual water level.

15   **3.2 Data sources**

[revised manuscript text omitted]

**5 Results and discussion**

**5.1 Scenario design**

NL is located in the China's north-south climate transition zone, and the time distribution of rainfall is severely uneven, with rainfall in the flood season accounting for 72% of the annual precipitation. This basin is located in the north-south climate transition zone of China, and the phenomenon of drought−flood abrupt alternation (DFAA) frequently occurs. DFAA refers to a rainstorm after a long period of drought and can result in severe flood damage.  Fig. 5 shows the process of rainfall at Wanglu station and the water levels of upper and lower lake at Nanyang and Weishan station in 2003. Since there was no precipitation in the first half of the year, NL dried up from April to June 2003, which caused a devastating impaction to the ecosystem of Nansi Lake. In the future, if NLB encounters another drought like 2003, emergency water diversion by the ER-SNWDP will be the first choose to provide ecological water for NL. However, heavy rains occurred after August 22 and caused a steep increase in the water level in NL, and 2003 represented a typical year of DFAA. The statistical data show that the waterlogged area of the NLB is 2360 km$^2$, and waterlogging disasters resulted in a great economic loss in the lakeside area.  So we take 2003 as an example to research the impacts of the emergency ecological water diversion by the ER-SNWDP on the waterlogging of the NLB.

 In recent years, the situation of water resource shortages has become increasingly tense in Shandong Peninsula, China. Even in the flood season, local water resources still encounter difficulty in meeting the demand; thus, the supply of water to the region that relies on the ER-SNWDP is expected to be more frequent.

[Figure]

**Figure 5: Rainfall at Wanglu station in NLB and water level variations of the upper lake and lower lake in 2003. The upper lake is dry when the water level of Nanyang is 31.5 m. Before July 2003, the Nansi Lake Basin suffered severe drought, but heavy rains in August caused serious floods.**

To analyze the influence of water diversion on waterlogging disasters in the lake basin along the ER -SNWDP, this paper set two conditions in which the ER-SNWDP supplies an emergency transfer of ecological water to NL and an emergency transfer of water to Shandong Peninsula. The ecological water refers to the transferred water to maintain the normal development and relative stability of all kinds of ecological systems in NL during drought period, and to prevent the recurrence of the dry lake situation that occurred in 2003.

(1) An ER-SNWDP emergency requires a supply of ecological water for NL.

Take a waterlogging occurred in NLB after August 22, 2003 as example, the impact of water diversion on the waterlogging process in NLB was studied. Considering flood control safety of NLB, Wwe assume that the emergency water transfer ended stopped on August 22, 2003 when rainfall beganat the beginning of rain, on August 22, 2003 (Fig.5). Because the water diversion project is no longer operational during the rainfall, the effect of water diversion on the flood process of NL is mainly to lift the water level of NL. which means that water diversion only affected the initial water level of NL under this condition. Therefore, two scenarios of waterlogging of NLB in August 2003 were simulated: The waterlogging process of NLB under the influence of the emergency water transfer in August 2003 is recorded as scenario ② , and the waterlogging process without water diversion is recorded as scenario ① , see table 2 for scenario Settings. The difference between scenario ① and scenario ② is the initial water level of the NL. In scenario ① , the water level of the upper lake and lower lake are 33.01 m and 32.20 m, which measured on August 22, 2003. In scenario ② , the water level of the upper lake and lower lake have been raised to 34 m and 32.3 m by the emergency water diversion. The rainfall process of the two scenarios was measured from August 22, 2003 to September 2, 2003. During the water diversion period, the water level of NL maintained a normal storage level, with water levels in the upper lake and lower lake of 34 m and 32.8 m, respectively. However, we consider that emergency water diversion occurs during the flood season. The initial water levels of the upper lake and lower lake are 34 m and 32.3 m when rainfall begins in the 2-D model, respectively. This situationis recorded as scenario ②, and the situation in which the waterlogging process occurs without water diversion is recorded as scenario ① (Table 2).

(2) An ER-SNWDP emergency requires a supply of water for Shandong Peninsula, China.

In this condition, the influence of water diversion on waterlogging disasters in the lakeside area of NL under different rainstorm intensities is analyzed. The processes of a 3-day designed rainfall with return periods of 5 years, 10 years and 20 years at six precipitation stations were calculated. Affected by the emergency water diversion, the initial water level of the upper lake and lower lake are 34.00 m and 32.30 m in scenario ⑥, ⑦ and ⑧. Because the water was transferred to alleviate the water shortage of Shandong Peninsula, China, the ER-SNWDP continued to operate during the rainfall. As a contrast, the initial water level of the upper lake and lower lake are 33.01 m and 32.20 m, which measured on August 22, 2003, in scenario ③, ④ and ⑤.In summary, a total of 8 simulation scenarios were set up as shown in Table 2.

**5.2 Impacts of the emergency water diversion by the ER-SNWDP emergency supply of ecological water for NL on the waterlogging of the NLB.**

Rice, cotton, corn and soybean are the main crops in, and the waterlogging tolerance depths of them are 0.5m, 0.1m, 01m and 0.1m, respectively (Wang, 2015). Therefore, the area with inundated deep above 0.1m and 0.5 is counted in the simulation results. In the calculation results of scenario ① and scenario ②, the areas with a submerged depth larger than 0.1 m and 0.5 m were counted respectively. Table 3 shows that the rainfall from August 22 to September 2, 2003 under the condition of no water diversion caused the inundated area in the lakeside area to reach 1126.59 km$^2$ and the area with a submerged depth over 0.5 m reached 383.68 km$^2$. Statistical data show that the total waterlogging area under the 36.79 m contour line was 1284.21 km$^2$ in 2003. The simulation result is slightly smaller than the survey result because the simulation did not cover the entire year and rain remained in the basin after September 2, 2003. In general, the simulation results can be considered reasonable.

Table 4 shows the increase in the waterlogging area in the lakeside area of NL under the condition of water diversion in the ER-SNWDP compared with that without water diversion. When the phenomenon of DFAA occurs, the lake level increase via emergency operation of the ER-SNWDP during the flood season increases the waterlogging intensity in the lakeside area of NL. Compared with the situation without water diversion, the area with a submerged depth over 10 cm increased by 34.26 km$^2$. Emergency water diversion resulted in a relative increase of 0.99% in submerged area.  The heavy disaster area with a water depth of more than 50 cm increased by 51.09 km$^2$, which was 13.32% higher than that without water diversion.

[revised manuscript text omitted]

The emergency water diversion of the SNWDP alleviated the drought situation in Shandong Peninsula area and also increased the degree of waterlogging in the NLB under the situation of abrupt drought flood alternations. The sluices of the Erji dam are the only flood-discharge channels of the upper lake, and the increased water in the upper lake due to water diversion also enlarged the task of flood discharge in those sluices. Fig. 9 shows the flood discharge process of the Erji dam project under different design rainfall . When the NLB encountered a storm with 5-year return periods, the upper lake level did not reach the flood discharge conditions without the influence of water diversion (Fig.8 a).  Affected by water diversion, the sluices began to drain the flood after 30 hours of rain, with a total discharge volume of 85 million m³ (Fig. 9a). When the NLB encountered a rainstorm with a 10-year return period, under the condition of water diversion,  the sluices began to drain the flood after 28 hour of rain, which was 36 hours ahead of the situation of no water diversion (at the 66th hour)  The total flood volume that discharge   was 104 million m³ higher than that without the affected of water diversion (Fig. 9b). When the NLB encountered a storm with a 20-year return period, under the condition of water diversion, the time  that the Erji dam project began to discharge the flood was  26 hours after the rain, which was 32 hours ahead of the situation with no water diversion (at the 58th hour) Comparing with the scenario without water diversion,  the total discharge volume  has increased 129 million m³  (Fig. 9c).

[Figure]

Figure 89: **Changes in the flood discharge of the Erji dam under water diversion with different storm levels. Flood discharge under rainfalls for different return periods: 5 year (a), 10 year (b), 20 year (c). The bar chart with different colors show the three-day rainfall process at each rainfall station. Black lines show the flood discharge process of the Erji dam with and without the effect of water diversion.**

**6 Conclusions and policy implications**

The following selected conclusions are presented.

(1) In order to clarify the impacts that interbasin water transfer on the water-receiving area, this study using MIKE model to simulate the flood and waterlogging in NLB. One- and two-dimensional coupled floods and a waterlogging simulation model of the NLB were established to simulate the water diversion of the ER-SNWDP. The MIKE 11 model was applied to simulate the flow movement of water in the water diversion channel and the tributaries of NL, and the MIKE 21 model was applied to simulate the waterlogging in the lakeside area and the water flow in NL. The verification results show that the presented method can effectively simulate the flood and waterlogging process in NLB under the effect of the ER-SNWDP.

[revised manuscript text omitted]

Bureau of South to North Water Transfer of Planning, Designing and Management, Ministry of Water Resources. Brief Introduction of General Planning for South-to-North Water Transfer Project, China Water Resources, 2003(2):11-13+18. (In Chinese).

Cai, X. eta Ringler, C.: Balancing agricultural and environmental water needs in China: Alternative scenarios and policy options, Water Policy, 9(SUPPL. 1), 95–108, doi:10.2166/wp.2007.047, 2007.

Cole, D. S., Carver, W. B., Hall, B. S. and Slover, P. C.: Interbasin Transfers of Water, Proc. 2011 Georg. Water Resour. Conf., 2011.

Davies, B. R., Thoms, M. eta Meador, M.: An assessment of the ecological impacts of inter‐basin water transfers, and their threats to river basin integrity and conservation, Aquat. Conserv. Mar. Freshw. Ecosyst., 2(4), 325–349, doi:10.1002/aqc.3270020404, 1992.

Dixon, B. eta Earls, J.: Effects of urbanization on streamflow using SWAT with real and simulated meteorological data, Appl. Geogr., 35(1–2), 174–190, 2012.

Dotto, C. B. S., Kleidorfer, M., Deletic, A., Rauch, W., Mccarthy, D. T. eta Fletcher, T. D.: Performance and sensitivity analysis of stormwater models using a Bayesian approach and long-term high resolution data, Environ. Model. Softw., 26(10), 1225–1239, 2011.

Dutta, D., Herath, S. eta Musiake, K.: Flood inundation simulation in a river basin using a physically based distributed hydrologic model, Hydrol. Process., 14(3), 497–519, 2015.

Emanuel, R. E., Buckley, J. J., Caldwell, P. V., McNulty, S. G. and Sun, G.: Influence of basin characteristics on the effectiveness and downstream reach of interbasin water transfers: Displacing a problem, Environ. Res. Lett., 10(12), doi:10.1038/s41598-017-06225-9, 2015.

Guo, Y., Liu S., Liang X Y. Study on Emergency Water Supply Scheme for North Extension of Eastern Route of South-to-North Water Transfer Project, Haihe Water Resources, 2018(3). (In Chinese).

Gupta, J. and Zaag, P. Van Der: Interbasin water transfers and integrated water resources management : Where engineering, science and politics interlock, Physics and Chemistry of the Earth, 33, 28–40, doi:10.1016/j.pce.2007.04.003, 2008.

He, B., Huang, X., Ma, M., Chang, Q., Tu, Y., Li, Q., Zhang, K. and Hong, Y.: Analysis of flash flood disaster characteristics in China from 2011 to 2015, Nat. Hazards, doi: 10.1007/s11069-017-3052-7, 2018a.

[revised manuscript text omitted]

**Appendix**

**Table 1 Statistical evaluation of model performance for water level simulations at selected gauging stations for 2007 and 2008 flood events.**

| Gauging station | NSE | |
|---|---|---|
| | 2007 | 2008 |
| Nanyang | 0.72 | 0.65 |
| Makou | 0.69 | 0.76 |
| Erji Lake (downstream) | 0.67 | 0.98 |
| Weishan | 0.82 | 0.99 |

**Table 2 Computational scenario setting**

| Sr. no. | Initial water level of NL | Rainfall | Whether the ER-SNWDP works during the simulation |
|---|---|---|---|
| ① | 33.01 m in upper lake, 32.20 m in lower lake | Actual daily rainfall from August 22 to September 2, 2003 | NO |
| ② | 34.0 m in upper lake, 32.3 m in lower lake | | |
| ③ | | Designed storm with return periods of 5 years | |
| ④ | Actual water level on August 22, 2003 | 10 years | NO |
| ⑤ | | 20 years | |
| ⑥ | | 5 years | |
| ⑦ | 34.0 m in upper lake, 32.3 m in lower lake | 10 years | YES |
| ⑧ | | 20years | |

**Table 3 Results of waterlogging in the lakeside area of NL**

| Sr. no. | Water depth of NL | | Area with an inundated depth above 0.1 m | | Area with an inundated depth above 0.5 m | |
|---|---|---|---|---|---|---|
| | Average (m) | Max (m) | Total area (km$^2$) | Area ratio (%) | Total area (km$^2$) | Area ratio (%) |
| ① | 2.47 | 5.96 | 1126.59 | 32.51 | 383.68 | 11.07 |
| ② | 2.80 | 6.14 | 1160.85 | 33.50 | 434.77 | 12.55 |

**Table 4 Increment of the waterlogging area in the lakeside area of NL**

| Contrastive analysis | Area with an inundated depth above 0.1 m | | | Area with an inundated depth above 0.5 m | | |
|---|---|---|---|---|---|---|
| | Increment (km$^2$) | Relative increase (%) | Area ratio increase | Increment (km$^2$) | Relative increase (%) | Area ratio increase |
| Variation | 34.26 | 3.04 | 0.99 | 51.09 | 13.32 | 1.47 |

**Table 5 Results of waterlogging simulations of the lakeside area under different scenarios.**

| Sr. no. | Water depth of NL | | Area with an inundated depth above 0.1 m | | Area with an inundated depth above 0.5 m | |
|---|---|---|---|---|---|---|
| | Average (m) | Max (m) | Total area (km$^2$) | Area ratio (%) | Total area (km$^2$) | Area ratio (%) |
| ③ | 1.86 | 5.23 | 793.75 | 22.91 | 60.63 | 1.75 |
| ④ | 2.03 | 5.87 | 907.85 | 26.20 | 159.98 | 4.62 |
| ⑤ | 2.41 | 6.31 | 1002.05 | 28.92 | 240.54 | 6.94 |
| ⑥ | 2.10 | 5.87 | 816.02 | 23.55 | 86.77 | 2.50 |
| ⑦ | 2.26 | 6.10 | 926.40 | 26.73 | 181.73 | 5.24 |
| ⑧ | 2.17 | 6.08 | 1016.68 | 29.34 | 260.76 | 7.53 |

---

## Author Comment (AC2) · 3 Jan 2019

Responses to Referee #2' comments

Referee #2:  The authors present an interesting study on the effects of a giant water diversion system, installed in China to transfer water from Yangtze River to provinces/regions further in the north, e.g., water scarce Shandong Peninsula. Due to the continuous water shortage in the receiving areas, the channel is operating contin- uously, though there are considerable amounts of rainfall along the way. The study ex-

plains, how local weather phenomenon interacts with an almost transcontinental water diversion scheme. Since it is not my primary field of research, it is difficult to evaluate whether the presented. study reveals some new concepts, tools or methods. However, the scientific methodology and input data are valid, the scenario selection is relevant and fits the requirements of the study. And finally, the authors reach substantial conclusions and clearly show the hazard of a water transfer project like the SNWDP. However, the manuscript is generally and particularly in the introduction part weak in English. Not only, but seriously influenced by that comes the second, much larger weakness. The presentation of the general setting and conditions of the study, the problem and why certain technical/hydraulic activities are done to manage/control water flow to, within and out of the Nansi Basin are not clearly described. The introduction is immature and also the presentation of the scenarios must be seriously improved. Particularly the wording is difficult, but also the description of figures lacks sufficient details. A lot of (at least to me) unknown technical phrases are used instead of international terminology. The figures have to be improved to meet journal requirements (e.g. include numbering of each figure, descriptions must be improved to allow understanding of each figure by itself). Finally, I strongly suggest to involve a native speaker. Doing so, the manuscript will surely meet the journal's requirements to be published in NHESS.

In addition to my general comments some specific in the following: Page 1 L15: What is a waterlogging simulation and wouldn't be flood simulation the better term? Which interactions are meant?

Answer: The sentence has been is revised as follow. "First, a flood simulation model was constructed to simulate the complex movement of the transferred water, waterlogging water in the lakeside area around Nansi Lake (NL) and the water in NL and its tributaries.

L28: a map showing the most important geographical places, including contour lines and information about elevations would be required. Furthermore, some climatic characterisations (annual rainfall as colourcode) would help understanding the general conditions.

Answer: Fig.1 has been divided to Fig.1 (in section1) and Fig.2 (in section 2).The contour lines of annual rainfall has been added in Fig.1 b, and Surface elevation information of Nansi Lake basin has bend added in Fig.2.

L29: development of what? L30: which lake and what does the lakeside area mean? It is somehow indicated in Fig. 1, but what are the borders, how are they defined, etc.

Answer: The economic development in lake basins along the ER-SNWDP. The lakeside area refers to the area with ground elevation below 36.79 m around Nansi Lake (Fig.2). We explained it in the manuscript, and Fig.2 is modified. Tanks a lot.

L31: what does "blocking of the rising lake level" means and what are subsequent "waterlogging disasters"? The entire process chain is not clear to me.

Answer: The lake level raised by the ER-SNWDP will decrease the drainage efficiency of pump stations and hinder flood discharge of rivers in the lakeside area and then influence the flood control and waterlogging drainage of the NLB.

L34: There is no reason to distrust the publication of Webber et al., however, aren't there better information about intended water volumes from official authorities/reports, etc.?

Answer: We added the information from official authorities/reports (Bureau of South to North Water Transfer of Planning, Designing and Management, Ministry of Water Resources, 2003). Thanks a lot.

L34: what is the "water diversion period"? L35: does the project stops and runs in intervals? And why is it obvious, that water tables rise? If water is consumed the same amounts as brought into the basins, nothing happens. The operational scheme does not become clear.

Answer: The water diversion period is from October to next May. Because Nansi Lake

is the storage pond, the water level of NL will be raised during the operation of the project (Bureau of South to North Water Transfer of Planning, Designing and Management, Ministry of Water Resources, 2003). We explain those in the manuscript as follow: According to the comprehensive plan of the SNWDP, (1) the first phase planning (before 2030) of the eastern route is designed to transfer 8.9 billion m3 water annually, and approximately 7 billion m3 is expected to be consumed in the above five lake basins and the route; and (2) the water diversion period covers the non-flood season (October to next May), and the water diversion will be stopped during the rest of the time (Bureau of South to North Water Transfer of Planning, Designing and Management, Ministry of Water Resources, 2003). As the water-receiving areas and the transmitting channels of the ER-SNWDP, the five lakes are used to store and regulate water resources, and the water level of the five lakes are significantly raised when the project is operating.

L37: what is "emergency water diversion"? I can imagine the possible meaning, but it has to be clarified. Why did water supply occurred sung flood period?

Answer: We revised the introduction and explained why did water supply occurred during flood period and what is "emergency water diversion" as follow: Otherwise, the Shandong Peninsula has suffered from severe drought and the water supply cannot meet the water demand even in the flood season for four consecutive years since the eastern route operated in 2013. Emergency water diversion, that is, the water transfer through the water diversion project in flood seasons to alleviate water shortages in the water-receiving areas, has been performed many times to supply water for Shandong Peninsula. Furthermore, considering the rigid demand for water resources caused by rapid economic and social development, frequent water transfers are expected in the flood season to alleviate water shortages in the water-receiving areas, which extends beyond the planned design of the ER-SNWDP (Guo et al., 2018).

L38: The sentence: "Furthermore, considering the rigid demand for water resources caused by rapid economic and social development, extreme hydrological events

caused by environment changes have increased along the ER-SNWDP." is unclear. And social development, extreme hydrological events caused by environment changes have increased along the ER-SNWDP." is unclear.

Answer: Those sentence have been revised as follow: Furthermore, considering the rigid demand for water resources caused by the rapid socioeconomic development, more frequent water transfers are expected to alleviate water shortages in the water-receiving areas in flood seasons (Guo et al., 2018). Meanwhile, extreme hydrological rainfall events caused by climate changes have increased in eastern China (Liu et al., 2015). Thus, the probability of rainstorm during the water diversion in these lake basins will increase.

Page 2 L3-6: Please rephrase the sentence the way you split it at least into 2-3. In the current version and with the amounts of questions, which rose at the passage before, the reason to do these simulations is still not clear.

Answer: the sentence is revised as follow: Therefore, simulate the flood and water-logging process in the water-receiving lake basin under the condition of emergency water diversion by the ER-SNWDP. And then, based on the simulation results to quantify the interbain water transfer on the flood control and waterlogging drainage in the water-receiving lake basin has great significance for the scientific management of the ER-SNWDP and the flood control and waterlogging drainage in lake basins along the route.

L6: which situation? L7-8: if there are studies, why is there a gap in the literature? L12: again, which gap?

Answer: We reorganized this part of the language in the introduction.

L14: why is it an important storage node?

Answer: We explained in the introduction as follow: The reason why we choose the NLB is that the NL basin is an important storage node of the ER-SNWDP. Because the

NL is the largest freshwater lake in northern China and has the largest water storage capacity among the lakes along the ER-SNWDP (Zhang, 2009).

This is important to state. L15: again, the authors base on preliminary knowledge: it is not introduced, that the lake is separated into 2 halves by a dam. How can one know about the upper and the lower lake since the lake is just mentioned as the Nansi Lake.

Answer: We explained perfected Fig.2 and cited it in introduction. Thanks a lot.

L17: what is the phenomenon, where explained, etc.

Answer: We added the explanation of "drought-flood abrupt alternation" phenomenon as follow: DFAA refers to a rainstorm after a long period of drought and can result in severe flood damage.

L17-L19: I have difficulties to follow the argumentation, too many things are written in a row without making it clear. L20: as far as I understood, NLB is a flow-through basin, where water tables are fluctuating due to natural floods and droughts and additionally due to water pumped into the basin, which will be pumped out as it comes in. Why is it a water receiving area?

Answer: Because Nansi Lake is a storage pond of the east route of the south to north water diversion project, and the flow of water transferred into Nansi Lake is 200 m3/s and the outflow is 100 m3/s. A large quantity of water assumed in the Nansi Lake Basin, so it a water receiving area.

L21: what are "structures under water diversion"?

Answer: The sentence has been deleted.

L34: what is "the world's annual water intake"? The world's water budget is closed.

Answer: This should be "the world's annual water withdrawals", and has been corrected. Thanks a lot.

Page 3 L24-26: please explain that corresponding effects by using the correct hydrological terminology.

Answer: We has revised the sentence as follow: Sun et al. (2008) took the Anyang River Basin which is crossed by the middle route of the SNWTP as an example to study the influence of the water diversion project on flooding in the river basin that the project passes through.

L30: what is the "disaster risk", which exact disaster(s) is/are meant?

Answer: It is the flood and waterlogging disaster. We have revised it in the manuscript.

Page 4: L5: why "land"?

Answer: It should be flood movement on the land in the lakeside area. We have corrected it. L15: the elevation is in respect to what: main sea level? L16: please explain slope in degrees. Answer: It is the height above sea level. Tanks a lot. We changed the content as follow: The lakeside area, refers to the area with a ground elevation (above sea level) below 36.79 m around the lake, and has the ground gradients in this area slope is between 0.0029° and 0.0057° (Tian et al., 2013; Wang et al., 2010).

L20: please explain what you mean by:" waterlogging in lakeside areas can no longer directly drain into rivers and lakes "and how does it work: "The waterlogging water is primarily carried into rivers and NL by pumping stations in the lakeside area."

Answer: We explained this part as follow: Flood control embankments have been built on both sides of the main inflow channels and around NL to prevent flooding from entering the lakeside area. Due to the low-lying terrain and the construction of flood control embankments, waterlogging in lakeside areas cannot directly drain itself into rivers and NL. The waterlogging water in the lakeside area is mainly pumped into rivers and NL through pumping stations.

L22: What does it mean the rainfall is concentrated? How high is the rate?

Answer: We want to express that the drainage capacity in the Nansi Lake Basin is low, and waterlogging caused by rainfall storm frequently occurs. The sentence has modified as follow: The existing pumping stations in this region cannot resist the rainstorm waterlogging occurred every five years, some pumping stations even cannot resist rainstorm waterlogging occurred every three years. Encountering heavy rainstorm, the water cannot be discharged in time, the NLB belongs to the historical frequent flooding area.

Page 5 Fig. 1: Please add (a), (b) and (c) to the single figures, as it is usual. In the upper right map: why is China's SE not continuously bordered? What does the inset map express and why is the international border southwards dashed? In the upper left map: please name the rivers. In the lower map: please indicate the location of the dam, Description of Fig. 5: delete "The logo of Copernicus publications" L4: what is Yangzhou? L5-8: please indicate the location of sluices and dams in Fig.1, otherwise it is impossible to follow.

Answer: Those figures have been modified as these suggestions. We made a map of China according to the convention of Chinese map. Thank you for your suggestions.

L8: what is the 1st phase, how is it defined, when does it end, etc.

Answer: "1st phase" refers to the first phase planning (before 2030) of the east route of the south to north water diversion project. We revised the sentence as follow: According to the first phase planing of the ER- SNWDP, the discharge that enters the lower lake occurs at 200 m3/s, and 5/8 of this amount will be pumped into the upper lake. Furthermore, this has been explained in introduction. Tanks a lot.

L8: what is the Liangji River mouth, 24.6 km up- or downstream? L9: "project operation": during which phase? Water table in respect to mean sea level?

Answer: This part has been revised as follow: The Changgou pumping station is built at 24m south of Liangji River Estuary in the upper lake, to transfer water to the Shandong

Peninsula. Water table has been explained in manuscript.

Page 6: L1-3: "project operation": during which phase? Water table in respect to mean sea level?

Answer: Here we want to express when the water diversion project is running, the water level of NL will be raised. This section has been modified as follow: When the water diversion project is running, the water level (in respect to mean sea level) of the lower lake will reach 32.8 m, which is 0.70 m higher than the mean annual water level.

L9: river channel bathymetry of which river? L11: please indicate the location of rainfall stations in map of Fig. 1. L13: please indicate the location of water level recording stations in Fig. 1

Answer: The channel bathymetry of all rivers simulated in the 1-D model. The location of rainfall stations have been added in Fig.2. Thanks a lot.

L17: what are "hydraulic engineering data"?

Answer: We explained as follow: (4) The hydraulic engineering data including the technique parameters of sluices, pumping stations and levees in the NLB were supplied by the Planning and Design Institute of the Huaihe Basin Hydraulic Management Bureau in Shandong Province, China.

Page7 Fig. 2: where is MIKE FLOOD integrated into the workflow? L8-9: Equations are usually cited in the text.

Answer: The corresponding modification of Fig. 3 is made. And we revised the reference.

Page 8 L5: again, elevation reference is missing

Answer: The elevation reference has been added in the manuscript.

L6 1000 pump stations are situated in the lakeside area, are they taking water out of

the lake or from the rivers?

Answer: They taking flood water out of lakeside area to rivers and Nansi Lake. We explained this as follow: A total of 1000 draining pump stations are used to drainage waterlogging water from the lakeside area to rivers and NL.

L18-19: Equations should be cited in the text.

Answer: We revised it in the manuscript. Thanks a lot.

L32: where is the model area and how was it chosen? L32: If I understand it right, lakeside area is outlined by the 36.79 m contour line. How was that contour line derived? Does the DEM resolve elevation in 1cm steps?

Answer: The model area includes NL and the lakeside area around the lake (Fig.2). This area was chosen according the advice of the Planning and Design Institute of the Huaihe Basin Hydraulic Management Bureau in Shandong Province, China. We explained it in section 3.2 Data sources.

Above, it is given; the area given by that contour line has an area of 3969 km2, now it is 4750 km2 large, which is almost 1 4 larger.

Answer: The model area includes the lake surface (1266 km2) and the lakeside area (3969 km2) and deducts the area of the rivers that simulated by 1-D model. The results (4750) was calculated by Arc GIS. We explained it in the manuscript.

L35-36: why only 5 out of the 7 stations?

Answer: We checked the manuscript and modified the stations. Tanks a lot.

Page9 L1-6: The entire passage is not clear, particularly due to the use of unusual terms like "into-lake river".

Answer: We want to express that the model uses the lateral link method to link the lakeside area and tributaries of NL to simulate the flood exchange between the lakeside

area and tributaries. We revised it as follow: A lateral link is applied to connect the lakeside area and tributaries of NL to simulate the flood exchange between the lakeside area and tributaries.

L16: I do not understand, why the Lake receives a roughness

Answer: Limited by data and technical means, it is a pity that we cannot obtain the roughness data of NL. Considering that we concerned more about the inundation process in the lakeside area. We only set a roughness for the lake in the model and the verified results show that the model error is acceptable. At home and abroad, the spatial distribution of roughness is one of the unsolved problems in large watershed flood simulation based on hydrodynamic model. In future research we will try to solve this problem. Tanks a lot.

Page 10 L10: loss of what?

Answer: It refers economic loss. We revised it as follow: The statistical data show that the waterlogged area of the NLB is 2360 km2, and waterlogging disasters resulted in a great economic loss in the lakeside area.

L10: The information concerning the locations of the stations is missing to follow argumentation

Answer: The locations of the stations have been added to Fig.2. Tanks a lot.

L18: not the paper, but the study, please change.

Answer: We checked carefully and modified the manuscript. Thanks a lot.

L19: please explain "ecological water" and what is the "emergency transfer" L22: I assume the "emergency transfer" follows some kind of a pre-set protocoll, which means, it must be possible to request its starting and ending time from the SNWDP operating authorities.

Answer: We explained these as follow: The ecological water refers to the transferred

water to maintain the normal development and relative stability of all kinds of ecological systems in the NLB during drought period, and to prevent the recurrence of dry lake situation that occurred in 2003. Emergency water diversion, that is, transfer water through water diversion project in flood season to alleviate water shortages in the water-receiving areas. It has been explained in introduction.

L22: rainfall where and why does the diversion now only affects NL? L25: in line 22 it is stated emergency diversion ended, here you state it continues... L26: what is the flood season, is it somehow restricted? Page 11: L1-2: what is "the situation"? L4: what is "this condition"? Unfortunately, the entire paragraph is not very clear. To me it stays unclear, when wis which scenario calculated and why (due to which conditions). It is a general difficulty in the entire paper. Due to unclear terminology, even headlines like 5.2 are not clear and it is impossible to follow the argumentation

Answer: This paragraph in section 5.1 scenario design was amended as follow: (1) An ER-SNWDP emergency requires a supply of ecological water for NL. Take the waterlogging occurred in the NLB after August 22, 2003 as example, the impact of water diversion on the waterlogging process in the NLB was studied. Considering flood control safety of the NLB, we assume that the emergency water transfer stopped at the beginning of rain on August 22, 2003 (Fig.5). Because the water diversion project is no longer operational during the rainfall, the effect of water diversion on the flood process of NL is mainly to lift the water level of NL. Therefore, two scenarios of waterlogging of the NLB in August 2003 were simulated: The waterlogging process of the NLB under the influence of the emergency water transfer in August 2003 is recorded as scenario âŚą , and the waterlogging process without water diversion is recorded as scenario âŚă , see table 2 for scenario settings. The difference between scenario âŚă and scenario âŚą is the initial water level of NL. In scenario âŚă , the water level of the upper lake and lower lake are measured on August 22, 2003. In scenario âŚą , the water level of the upper lake and lower lake have been raised to 34 m and 32.3 m by the emergency water diversion. The rainfall process of the two scenarios was measured

from August 22, 2003 to September 2, 2003. The headlines has been revised as fol-low: Impacts of the emergency water diversion by the ER-SNWDP on the waterlogging of the NLB. Tanks a lot.

Page 12 L1-2 I don't understand the difference between scenarios 1 and 2, different initial water tables? If yes, what was the water table of Scenario 1? Table 2 is not clear, please re-organize it and give exact numbers.

Answer: The difference between scenarios 1 and 2 has been explained in answer Page 10 L22: - Page 11: L1-2. And the Table 2 has been revised. Thanks a lot.

L4-7: Here, for the first time, the reason for the study is clearly described and one can follow the intention of the authors. This should be integrated into the introduction, of course in a different way, but that's the reason for the study, I guess.

Answer: We have revised the introduction according to your suggestion.

L4-7: indicate the scenario number behind each pre-condition and refer to table 2 directly, not at the next paragraph. L8: prevent one sentence paragraphs.

Answer: We have revised this paragraph as follow: In this condition, the influence of water diversion on waterlogging disasters in the lakeside area of NL under different rainstorm intensities is analyzed. The processes of a 3-day designed rainfall with return periods of 5 years, 10 years and 20 years at six precipitation stations were calculated. Affected by the emergency water diversion, the initial water level of the upper lake and lower lake are 34.00 m and 32.30 m in scenario âŚě, âŚę and âŚğ. Because the water was transferred to alleviate the water shortage of Shandong Peninsula, China, the ER-SNWDP continued to operate during the rainfall. As a contrast, the initial water level of the upper lake and lower lake are 33.01 m and 32.20 m, which measured on August 22, 2003, in scenario âŚć, âŚč and âŚď. In summary, a total of 8 simulation scenarios were set up as shown in Table 2.

L10: why are areas of 0.1 and 0.5 considered?

Answer: We explained as follow: Rice, cotton, corn and soybean are the main crops in NLB, and the waterlogging tolerance depths of them are 0.5m, 0.1m, 01m and 0.1m, respectively (Lin et al.,2007).

L20: 0.99% of what?

Answer: 0.99% of relative increase in submerged area.

L25: EASTERN portion of the lake is mountainous... L26: the interaction is either given or not. There is no option to interact larger or less. Please describe what you want to say differently: e.g. The influence of...results in increased...

Answer: We revised this sentence as follow: Comparing with the eastern part of NLB, the terrain of the western NLB is lower and flat, and the western NLB has a greater impediment to drainage water into NL when the water level of NL is high.

L36: compared to what, scenario 1? Please describe sharply and refer to fig. 6

Answer: We revised this part as follow: Affected by the higher initial water level raised by the water diversion, the flood discharge start time of the Erji dam junction in scenario âŚą is 4 days earlier than that in scenario âŚă. Furthermore, the total amount of flood discharge in scenario âŚą increased by approximately 249 million m3 compared to scenario âŚă (Fig.7). Under Figure 7, we added a detailed description. Thanks a lot.

Page 13 L6-7: please check, which one is correct: Figure 7 a and b or the reference in the text, in the moment it is switched. Fig. 7: Please explain a and b in Figure description.

Answer: We checked Fig.8, and modified it in manuscript. A detailed description has been added. Thanks a lot.

L9-11: again, please be very clear in describing the effects of with/without active diversion.

Answer: We revised the effects of water diversion as follow: When the NLB encountered a design rainfall with 5-year return periods, the water of level upper lake did not reach the flood discharge condition without the influence of water diversion (Fig.9 a). Affected by water diversion, the sluices of the Erji dam began to drain the flood 30 hours after the start of the rain, and the total discharge volume is 85 million m3 (Fig. 9a). When the NLB encountered a rainstorm with 10-year return periods under the affection of water diversion, the sluices began to drain the flood 28 hours after rainfall. Compared with the situation without water diversion, the flood discharge time of the sluices was 36 hours earlier (at the 66th hour). The total flood volume discharged by the Erji dam project was 104 million m3 higher than that without the affected of water diversion (Fig. 9b). When the NLB encountered a storm with a 20-year return periods under the condition of water diversion, the time that the sluices began to discharge the flood was 26 hours after the start of the rain, which was 32 hours ahead of the situation with no water diversion (at the 58th hour). Affected by water diversion, the total discharge volume of flood has increased 129 million m3 (Fig. 9c).

L18: in Fig. 8a it do not look like 36 hours, more like 31 hours. Wrong figure? L20-25: please use some different descriptions for time: 30th hour is very uncommon and "e36 hours" is unclear to me. L25 reference is wrong, it is fig. 8c

Answer: We checked carefully and revised it in text and figure. Thanks a lot.

Page 14: Fig. 8: please explain a, b, c in the figure description.

Answer: We and descriptions under Fig. 9. The flood discharge under rainfalls for different return periods: 5 year (a), 10 year (b), 20 year (c). The bar chart with different colors show the three-day rainfall process at each rainfall station. Black lines show the flood discharge process of the Erji dam with and without the effect of water diversion.

Please also note the supplement to this comment:
https://www.nat-hazards-earth-syst-sci-discuss.net/nhess-2018-216/nhess-2018-216-AC2-supplement.pdf

Legend

—— River

—— ER-SNWDP

—— International boundary

Lake

Shandong Province

Yellow River

Yangtze River

Yangzhou City

**Fig. 1.**

The following figure shows the study area map.

Legend
- Lake
- ER-SNWDP
- annual_rainfall
- Shandong Province

**Fig. 2.**

**Fig. 3.**

Legeng
⊗ Sluice
● Rainfall station
▲ Water level station
━━ Erji Dam
── ER-SNWDP
‧‧‧ Levee
── River
▉ Nansi Lake
▢ Lakeside area of NL

DEM
<VALUE>
6.00 - 38.00
38.00 - 70.00
70.00 - 150.00
150.00 - 300.00
300.00 - 647.00

[Figure]

**Fig. 4.**

[Figure]

(a)

Discharge of flood drainage

With water diversion
Without water diversion

Precipitation station

XC
LSZ
WL
WGD
WZ
WC

(b)

(c)

**Fig. 5.**

---

## Author Response (AR1)

**Responses to Referees' comments**

**Referee #1: This is a very interesting paper. The effect of interbasin water diversion and flood management in a large lake basin is very important and hot topic. In this study, authors constructed a flood and waterlogging simulation model for assessing the effect of the South-to-North Water Diversion Project's eastern route on flooding and drainage Nansi Lake, a water-receiving area of the water diversion project in China. The model and the problem discussed in this paper were very useful and meaningful. I would like to recommend its publication. The following issues need to be solved well before its publication.**

**Answer:** This manuscript was edited for English language usage, grammar, spelling and punctuation by one or more native English-speaking editors at Nature Research Editing Service.

**1. Please kindly polish the language. For example, "Therefore, simulations of the flood and waterlogging process in the lakeside areas under the condition of emergency water diversion by the SNWDP and analyses of the impacts of emergency water diversions on flood and waterlogging drainage characteristics and the scheduling of flood control and drainage projects must be performed to strengthen the scientific scheduling of water diversion projects and flood control projects in water-receiving regions." There are grammatically mistakes in this sentence. I would like to recommend that authors used short sentences to replace too long sentence like this one.**

**Answer:** The language in this paper has been improved. This sentence has been revised as follow: "To strengthen the scientific scheduling of flood control projects in water-receiving areas and water diversion projects, it is necessary to clarify the influence law of water diversion on flood control and drainage in these lake basins."

Thanks a lot.

**2. The abstract can be more informative by highlighting the significance and novel contribution of this research. Moreover, key findings of this paper must be included in the abstract.**

The abstract is revised as follow: The water levels of lakes along the eastern route of the South-to-North Water Diversion Project (ER-SNWDP) are expected to rise significantly and subsequently affect the processes of flood control and drainage in corresponding lake basins. However, few studies have focused on the impacts of inter-basin water diversion on the flood control and drainage of water-receiving areas at the lake basin scale. Using MIKE software, this paper builds a coupled hydrodynamic model to address the existing literature gap regarding the impacts of inter-basin water diversion on the processes of flood control and drainage in a water-receiving lake basin, and it considers the many types of hydraulic structures in the model. First, a flood simulation model was constructed to simulate the complex movement of water transferred by the ER-SNWDP, by waterlogging in the lakeside area around Nansi Lake (NL) and water in the NL and its tributaries. The ER-SNWDP was also considered in the model. Second, the model was calibrated and verified with measurement data, and the results showed that the model was efficient and presented a

Nash-Sutcliffe efficiency coefficient (NSE) between 0.65 and 0.99. Third, the processes of flooding and draining in the lakeside area of NL were simulated under different water diversion and precipitation values. Finally, the impacts of the emergency operations of the ER-SNWDP on flood control and waterlogging drainage in the lakeside area of NL were analysed based on the results from the proposed model, and some implications are presented for the integrated management of inter-basin water diversion and affected lakes.

**3. Please check the reference style according to journal requirements. In addition, please avoid multiple references. It is not recommended to cite over three references in one sentence.**

**Answer:** We checked carefully and deleted some unimportant references in the manuscript. Thanks a lot.

**4. The figures 3 and 5 are very beautiful. But it is a little confusing. Authors need to introduce the directions more clearly.**

**Answer:** We added a detailed description for all figures in the revised manuscript.
Thanks a lot.

**5. In the introduction, authors need to make the innovations (i.e., the research gap this paper fills) more clear.**

**Answer:** The introduction has been rewritten as recommended by the two experts, to the innovations more clear.

**6. In the lines 20-25, authors demonstrate three key issues they aim to solve. Authors need to give answers to these questions in the conclusions.**

**Answer:** We answered the proposed issues in the conclusions one by one as:

(1) To clarify the impacts that inter-basin water transfer has on the water-receiving areas, this study used the MIKE model to simulate flooding and waterlogging in the NLB. One- and two-dimensional coupled flooding and waterlogging simulation models of the NLB were established to simulate the water diversion of the ER-SNWDP. The MIKE 11 model was applied to simulate the flow movement of water in the water diversion channel and tributaries of NL, and the MIKE 21 model was applied to simulate waterlogging in the lakeside area and water flow in NL. The verification results show that the presented method can effectively simulate the flooding and waterlogging processes in the NLB under the effect of the ER-SNWDP.

(2) The ER-SNWDP emergency water transfer to NL increases the risk of waterlogging damage in the lakeside area if it occurs simultaneously with the DFAA. The increased water level caused by water diversion decreases the efficiency of waterlogging drainage, and as a result, the waterlogged area with an inundated water depth above 0.1 m increased by 0.99%, and that with an inundated water depth above 0.5 m increased by 13.32%. The flood-discharge time of the ED increased by 4 days, and the total discharge volume increased by 249 million m3 during the simulation.

(3) The ER-SNWDP emergency water transfer to Shandong Peninsula raised the water level of NL,

which acted as a regulation and storage lake. Compared with the no water transfer situation, the waterlogging areas in the lakeside area increased when NL encountered storms with 5-year, 10-year and 20-year return periods under water diversion. The calculation results show that water diversion has a more obvious effect on waterlogging areas, with an inundated water depth above 0.5 m, and the area increasing by 8.4-43.1%. The total volume of flooding discharged by the ED also increased. In addition, we found that with the increase in rainfall intensity, the influence of water diversion on the lakeside area in the NL inundated area gradually decreased and the water transfer had more serious effects during rainstorms with lower return period.

**7. Line 41: Wrong format of the citation to Bisht et al. 2016.**

**Answer:** The citation is revised accordingly, thanks a lot.

**8. Line 24-25, page 10 "Considering emergency water diversion occurs during the flood season is scenario 2", while the ER-SNWDP doesn't work during the simulation in scenario 2 according to the table 2? Please modify table 2.**

**Answer:** We carefully checked this part of content and revised the mistake as follow: Considering the flood control safety of the NLB, we assume that the emergency water transfer stopped at the beginning of the rainfall event on August 22, 2003 (Fig.5). Because the water diversion project is no longer operational during rainfall, the effect of water diversion on the flood process of NL mainly increases the water level of NL. Therefore, two scenarios of waterlogging of the NLB in August

2003 were simulated: the waterlogging process of the NLB under the influence of emergency water transfer in August 2003, which is recorded as scenario ②, and the waterlogging process without water diversion is recorded as scenario ① (see table 2 for scenario settings).

Thanks a lot.

**9. Line 25, page 11 the eastern portion of NL is mountainous?**

**Answer:** We revised the sentence as follow: "Compared with the eastern NL, due to the lower and flatter terrain in the western NL, the raised water level of NL has a greater impediment to drainage in the western NL."

Tanks a lot.

**10. Line 6-7, page 13. Text descriptions of both figure a and figure b are reversed. Please correction.**

**Answer:** We corrected the mistakes in the manuscript. Thanks a lot.

**11. Line 25, page13. Should it be figure 8(c) instead of figure 8(b)?**

**Answer:** We revised it in manuscript. Thanks a lot.

**Referee #2**: The authors present an interesting study on the effects of a giant water diversion system, installed in China to transfer water from Yangtze River to provinces/regions further in the north, e.g., water scarce Shandong Peninsula. Due to the continuous water shortage in the receiving areas, the channel is operating continuously, though there are considerable amounts of rainfall along the way. The study explains, how local weather phenomenon interacts with an almost transcontinental water diversion scheme. Since it is not my primary field of research, it is difficult to evaluate whether the presented. study reveals some new concepts, tools or methods. However, the scientific methodology and input data are valid, the scenario selection is relevant and fits the requirements of the study. And finally, the authors reach substantial conclusions and clearly show the hazard of a water transfer project like the SNWDP.

However, the manuscript is generally and particularly in the introduction part weak in English. Not only, but seriously influenced by that comes the second, much larger weakness. The presentation of the general setting and conditions of the study, the problem and why certain technical/hydraulic activities are done to manage/control water flow to, within and out of the Nansi Basin are not clearly described. The introduction is immature and also the presentation of the scenarios must be seriously improved. Particularly the wording is difficult, but also the description of figures lacks sufficient details. A lot of (at least to me) unknown technical phrases are used instead of international terminology. The figures have to be improved to meet journal requirements (e.g. include numbering of each figure, descriptions must be improved to allow understanding of each figure by itself). Finally, I strongly suggest to involve a native speaker.

Doing so, the manuscript will surely meet the journal's requirements to be published in NHESS.

**Answer:** This manuscript was edited for English language usage, grammar, spelling and punctuation by one or more native English-speaking editors at Nature Research Editing Service.

**In addition to my general comments some specific in the following:**

**Page 1 L15: What is a waterlogging simulation and wouldn't be flood simulation the better term? Which interactions are meant?**

**Answer:** The sentence has been is revised as follow: First, a flood simulation model was constructed to simulate the complex movement of water transferred by the ER-SNWDP, by waterlogging in the lakeside area around Nansi Lake (NL) and water in the NL and its tributaries.

**L28: a map showing the most important geographical places, including contour lines and information about elevations would be required. Furthermore, some climatic characterisations (annual rainfall as colourcode) would help understanding the general conditions.**

**Answer:** Fig. 1 has been divided to Fig. 1 (in section1) and Fig. 2 (in section 2).The contour lines of annual rainfall has been added in Fig.1 b, and Surface elevation information of Nansi Lake basin has bend added in Fig. 2.

**L29: development of what? L30: which lake and what does the lakeside area mean? It is somehow indicated in Fig. 1, but what are the borders, how are they defined, etc.**

**Answer:** The economic development in lake basins along the ER-SNWDP. The lakeside area refers to the area with ground elevation below 36.79 m around Nansi Lake (Fig. 2). We explained it in the manuscript, and Fig. 2 is modified.

[Figure]

**Figure 2: Location of the study area, with the positions of the rainfall and water level gauges, rivers and hydraulic projects considered in the river system and a topographic map of the NLB. The area surrounded by the black line is the lakeside area, which is defined according to the suggestion of the Planning and Design Institute of the Huaihe Basin Hydraulic Management Bureau in Shandong Province, China.**

Tanks a lot.

**L31: what does "blocking of the rising lake level" means and what are subsequent**

**"waterlogging disasters"? The entire process chain is not clear to me.**

**Answer:** This sentence has been revised as follow: As water-receiving areas and transmitting channels of the ER-SNWDP, these five lakes are used to store and regulate water resources, and the water levels of the five lakes are significantly increased when the project is operating. The increased lake level impedes flood control and waterlogging drainage in the water-receiving lake basin, especially in low-lying lake basins.

**L34: There is no reason to distrust the publication of Webber et al., however, aren't there better information about intended water volumes from official authorities/reports, etc.?**

**Answer:** We added the information from official authorities/reports (Bureau of South to North Water Transfer of Planning, Designing and Management, Ministry of Water Resources, 2003).
Thanks a lot.

**L34: what is the "water diversion period"? L35: does the project stops and runs in intervals? And why is it obvious, that water tables rise? If water is consumed the same amounts as brought into the basins, nothing happens. The operational scheme does not become clear.**

**Answer:** The water diversion period is from October to next May. Because Nansi Lake is the storage pond, the water level of NL will be raised during the operation of the project (Bureau of South to North Water Transfer of Planning, Designing and Management, Ministry of Water Resources, 2003).

We explain those in the manuscript as follow: According to the comprehensive plan of the SNWDP, (1) the first planning phase (before 2030) of the eastern route is designed to transfer 8.9 billion m3 of water annually, and approximately 7 billion m3 is expected to be consumed in the above five lake basins and route; (2) the water diversion period covers the non-flood season (October to the following May), and the water diversion ceases for the rest of the time (Bureau of South to North Water Transfer of Planning, Designing and Management, Ministry of Water Resources, 2003). Lake basins along the ER-SNWDP, which are also called water-receiving areas for large quantities of water are consumed in these basins during the water transfer period. As water-receiving areas and transmitting channels of the ER-SNWDP, these five lakes are used to store and regulate water resources, and the water levels of the five lakes are significantly increased when the project is operating.

**L37: what is "emergency water diversion"? I can imagine the possible meaning, but it has to be clarified. Why did water supply occurred sung flood period?**

**Answer**: We revised the introduction and explained why did water supply occurred during flood period and what is "emergency water diversion" as follow: In addition, the Shandong Peninsula has suffered from severe drought and the water supply cannot meet the water demand even in the flood season (June to October) for four consecutive years since the eastern route began operation in 2013. Emergency water diversion (i.e., water transfer through the water diversion project in flood seasons to alleviate water shortages in water-receiving areas) has been performed many times to supply water to the Shandong Peninsula. Furthermore, considering the rigorous demand for water resources

caused by rapid socio-economic development, more frequent water transfers are expected in the flood season (June to October) (Guo et al., 2018).

**L38: The sentence: "Furthermore, considering the rigid demand for water resources caused by rapid economic and social development, extreme hydrological events caused by environment changes have increased along the ER-SNWDP." is unclear. And social development, extreme hydrological events caused by environment changes have increased along the ER-SNWDP." is unclear.**

**Answer:** Those sentence have been revised as follow: Furthermore, considering the rigorous demand for water resources caused by rapid socio-economic development, more frequent water transfers are expected in the flood season (June to October) (Guo et al., 2018). Meanwhile, extreme rainfall events caused by climate changes have increased in eastern China (Liu et al., 2015). Thus, the probability of rainstorms during the water diversion period in these lake basins will increase.

**Page 2 L3-6: Please rephrase the sentence the way you split it at least into 2-3. In the current version and with the amounts of questions, which rose at the passage before, the reason to do these simulations is still not clear.**

**Answer:** The sentence is revised as follow: This study aims to bridge the knowledge gap on the impacts of water diversion by the ER-SNWDP on flood control and inundation in the NLB. For this purpose, the following were set as the two main objectives of this study: 1) to develop

one-dimensional and two-dimensional hydrodynamic models that can simulate both the ER-SNWDP operations and its impacts on the flooding and waterlogging processes within the NLB, and 2) to clarify the impacts of the ER-SNWDP on flood control and waterlogging drainage in the NLB under different rainstorm events.

**L6: which situation?**

**Answer:** We reorganized this part of the language in the introduction. This sentence has been deleted. Thanks a lot.

**L7-8: if there are studies, why is there a gap in the literature? L12: again, which gap?**

**Answer:** We reorganized this part of the language in the introduction. This sentence has been revised as follow: However, the impacts of water diversion on flood control and waterlogging drainage in the NLB are relatively unexplored. A gap in assessing the impacts of inter-basin water diversion on flood inundation in the NLB still exists. An integrated model that could explicitly simulate the impacts of water transfer project operation on the spatiotemporal aspects of inundation is one of the necessary tools that can help bridge this gap.

**L14: why is it an important storage node?**

**Answer:** We explained in the introduction as follow: The reason why we chose the NLB is that the

NL is an important storage node of the ER-SNWDP, as the NL is the largest freshwater lake in northern China and has the largest water storage capacity among the lakes along the ER-SNWDP (Zhang, 2009).

**This is important to state. L15: again, the authors base on preliminary knowledge: it is not introduced, that the lake is separated into 2 halves by a dam. How can one know about the upper and the lower lake since the lake is just mentioned as the Nansi Lake.**

**Answer:** We perfected Fig.2 and cited it in introduction. Thanks a lot.

**L17: what is the phenomenon, where explained, etc.**

**Answer:** We added the explanation of "drought-flood abrupt alternation" phenomenon in section **5.1 Scenario design** as follow: DFAA refers to a rainstorm after a long period of drought and can result in severe flood damage.

**L17-L19: I have difficulties to follow the argumentation, too many things are written in a row without making it clear.**

**Answer:** We have made a major revision to the introduction.

Thanks a lot.

**L20: as far as I understood, NLB is a flow-through basin, where water tables are fluctuating due to natural floods and droughts and additionally due to water pumped into the basin, which will be pumped out as it comes in. Why is it a water receiving area?**

**Answer:** Because Nansi Lake is a storage pond of the east route of the south to north water diversion project, and the flow of water transferred into Nansi Lake is 200 m$^3$/s and the outflow is 100 m$^3$/s. A large quantity of water assumed in the Nansi Lake Basin, so it a water-receiving area.

We have explained it in the introduction as follow: Lake basins along the ER-SNWDP, which are also called water-receiving areas for large quantities of water are consumed in these basins during the water transfer period.

**L21: what are "structures under water diversion"?**

**Answer:** The sentence has been deleted.

**L34: what is "the world's annual water intake"? The world's water budget is closed.**

**Answer:** This should be "the world's annual water withdrawals", and has been corrected. Thanks a lot.

**Page 3 L24-26: please explain that corresponding effects by using the correct hydrological terminology.**

**Answer:** We has revised the sentence as follow: Based on a two-dimensional mathematic model, Sun et al. (2008) used the Anyang River Basin, which intersects with the middle route of the SNWTP, as an example to study the influence of the water diversion project on flooding in the river basin that the project passes through.

**L30: what is the "disaster risk", which exact disaster(s) is/are meant?**

**Answer:** It is the flood and waterlogging disaster. We have revised it in the manuscript.

**Page 4: L5: why "land"?**

**Answer:** It should be flood movement on the land in the lakeside area. We have corrected it.

**L15: the elevation is in respect to what: main sea level? L16: please explain slope in degrees.**

**Answer:** It is the height above sea level. Tanks a lot. We changed the content as follow: The lakeside area refers to the area with a ground elevation (above sea level) below 36.79 m around the lake, and the ground slope in this area is between 0.0029 ° and 0.0057 ° (Tian et al., 2013; Wang et al., 2010).

**L20: please explain what you mean by:" waterlogging in lakeside areas can no longer directly drain into rivers and lakes "and how does it work: "The waterlogging water is primarily**

**carried into rivers and NL by pumping stations in the lakeside area.**"

**Answer:** We explained this part as follow: Flood control embankments have been built on both sides of the main inflow channels and around NL to prevent flooding from entering the lakeside area. Due to the low-lying terrain and the construction of flood control embankments, waterlogging in lakeside areas cannot drain into rivers and NL. The waterlogged water in the lakeside area is mainly pumped into rivers and NL through pumping stations.

**L22: What does it mean the rainfall is concentrated? How high is the rate?**

**Answer:** We want to express that the drainage capacity in the Nansi Lake Basin is low, and waterlogging caused by rainfall storm frequently occurs. The sentence has modified as follow: However, the existing pumping stations in this region cannot resist rainstorm waterlogging every five years, and some pumping stations cannot resist rainstorm waterlogging every three years. When encountering heavy rainstorms, the water cannot be discharged in time; therefore the NLB is historically known as a flood prone area.

**Page 5 Fig. 1: Please add (a), (b) and (c) to the single figures, as it is usual.**

**Answer:** Fig. 1 has been divided into Fig. 1 and Fig. 2, and these figures have been modified as these suggestions.

**In the upper right map: why is China's SE not continuously bordered? What does the inset map express and why is the international border southwards dashed? In the upper left map: please name the rivers.**

**Answer:** We made a map of China according to the convention of Chinese map.

**In the lower map: please indicate the location of the dam, Description of Fig. 5: delete "The logo of Copernicus publications"**

**Answer:** The location of the dam has been added in Fig. 2, and "The logo of Copernicus publications" of Fig. 5 has been deleted.

Thanks a lot.

**L4: what is Yangzhou?**

Answer: It refers to Yangzhou City, we have revised in the manuscript.

**L5-8: please indicate the location of sluices and dams in Fig.1, otherwise it is impossible to follow.**

**Answer:** These figures have been modified as these suggestions.

Thank you for your suggestions.

**L8: what is the 1st phase, how is it defined, when does it end, etc.**

**Answer:** "1st phase" refers to the first phase planning (before 2030) of the east route of the south to north water diversion project. We revised the sentence as follow: According to the first phase planning of the ER-SNWDP, the discharge that enters the lower lake is 200 $m^3$/s, and 5/8 of this amount is pumped into the upper lake. Furthermore, this has been explained in introduction.

Tanks a lot.

**L8: what is the Liangji River mouth, 24.6 km up- or downstream?**

**Answer:** This part has been revised as follow: The Changgou pumping station is built 24 m south of the Liangji River Estuary in the upper lake to transfer water to the Shandong Peninsula. The Changgou pumping station is built 24 m south of the Liangji River Estuary in the upper lake to transfer water to the Shandong Peninsula.

**L9: "project operation": during which phase? Water table in respect to mean sea level?**

**Answer:** Water table has been explained in manuscript as follow: When the water diversion project is running, the water level (with respect to mean sea level) of the lower lake reaches 32.8 m, which is 0.70 m higher than the mean annual water level.

**Page 6: L1-3: "project operation": during which phase? Water table in respect to mean sea level?**

**Answer:** Here we want to express when the water diversion project is running, the water level of NL will be raised. This section has been modified as follow: When the water diversion project is running, the water level (in respect to mean sea level) of the lower lake will reach 32.8 m, which is 0.70 m higher than the mean annual water level.

**L9: river channel bathymetry of which river?**

**Answer:** The channel bathymetry of all rivers simulated in the 1-D model.

**L11: please indicate the location of rainfall stations in map of Fig. 1. L13: please indicate the location of water level recording stations in Fig. 1**

**Answer:** The location of rainfall stations have been added in Fig.2.

Thanks a lot.

**L17: what are "hydraulic engineering data"?**

**Answer:** We explained as follow: (4) The hydraulic engineering data, including the technical parameters of sluices, pumping stations and levees in the NLB, were supplied by the Planning and

Design Institute of the Huaihe Basin Hydraulic Management Bureau in Shandong Province, China. These data include the location and drainage capabilities of the pump stations, the locations and sizes of the flood control embankments, and the hydraulic parameters of the sluices.

**Page7 Fig. 2: where is MIKE FLOOD integrated into the workflow?**

**Answer:** The corresponding modification of Fig. 3 is made. Thanks a lot.

**L8-9: Equations are usually cited in the text.**

**Answer:** We have revised the reference in the manuscript. Thanks a lot.

**Page 8 L5: again, elevation reference is missing**

**Answer:** The elevation reference has been added in the manuscript.

**L6 1000 pump stations are situated in the lakeside area, are they taking water out of the lake or from the rivers?**

**Answer:** They taking flood water out of lakeside area to rivers and Nansi Lake. We explained this as follow: A total of 1000 draining pump stations are used to drain waterlogged water from the lakeside area to rivers and NL, and the model generalized the pump stations according to the distribution of

each pump station on both sides of the rivers.

**L18-19: Equations should be cited in the text.**

**Answer:** We revised it in the manuscript.

Thanks a lot.

**L32: where is the model area and how was it chosen? L32: If I understand it right, lakeside area is outlined by the 36.79 m contour line. How was that contour line derived? Does the DEM resolve elevation in 1cm steps?**

**Answer:** The model area includes NL and the lakeside area around the lake (Fig.2). This area was chosen according the advice of the Planning and Design Institute of the Huaihe Basin Hydraulic Management Bureau in Shandong Province, China. We explained it in section 3.2 Data sources.

**Above, it is given; the area given by that contour line has an area of 3969 km2, now it is 4750 km2 large, which is almost 1 4 larger.**

**Answer:** The model area includes the lake surface (1266 km$^2$) and the lakeside area (3969 km$^2$) and deducts the area of the rivers that simulated by 1-D model. The results (4750) was calculated by Arc GIS. We explained it in the manuscript.

**L35-36: why only 5 out of the 7 stations?**

**Answer:** We checked the manuscript and modified the stations.

Tanks a lot.

**Page9 L1-6: The entire passage is not clear, particularly due to the use of unusual terms like "into-lake river".**

**Answer:** We want to express that the model uses the lateral link method to link the lakeside area and tributaries of NL to simulate the flood exchange between the lakeside area and tributaries. We revised it as follow: A lateral link is applied to connect the lakeside area and the tributaries of NL to simulate the flood exchange between the lakeside area and tributaries.

**L16: I do not understand, why the Lake receives a roughness**

**Answer:** Limited by data and technical means, it is a pity that we cannot obtain the roughness data of NL. Considering that we concerned more about the inundation process in the lakeside area. We only set a roughness for the lake in the model and the verified results show that the model error is acceptable. At home and abroad, the spatial distribution of roughness is one of the unsolved problems in large watershed flood simulation based on hydrodynamic model. In future research we will try to solve this problem.

Tanks a lot.

**Page 10 L10: loss of what?**

**Answer:** It refers economic loss. We revised it as follow: The statistical data show that the waterlogged area of the NLB is 2360 km$^2$, and waterlogging disasters resulted in a great economic loss in the lakeside area.

**L10: The information concerning the locations of the stations is missing to follow argumentation**

**Answer:** The locations of the stations have been added to Fig.2.

Tanks a lot.

**L18: not the paper, but the study, please change.**

**Answer:** We checked carefully and modified the manuscript. Thanks a lot.

**L19: please explain "ecological water" and what is the "emergency transfer" L22: I assume the "emergency transfer" follows some kind of a pre-set protocoll, which means, it must be possible to request its starting and ending time from the SNWDP operating authorities.**

**Answer:** We explained these as follow: Ecological water refers to the transferred water need to

maintain the normal development and relative stability of all types of ecological systems in NL during the drought period and prevent the recurrence of the dry lake situation that occurred in 2003. Emergency water diversion, that is, transfer water through water diversion project in flood season to alleviate water shortages in the water-receiving areas. It has been explained in introduction.

**L22: rainfall where and why does the diversion now only affects NL?**

**Answer:** It refers to rainfall in NLB. Because the ER-SNWDP is stopped when the rainfall beginning, so the diversion only increased the initial water level of the NL during the flood.

**L25: in line 22 it is stated emergency diversion ended, here you state it continues...**

**Answer:** When transfer water to the Shandong peninsula, the project need to operate even the rainfall occurs in NLB. So we state it continues

**L26: what is the flood season, is it somehow restricted?**

**Answer:** The flood season is from June to October, during which the rainstorm occurs frequently.

**Page 11: L1-2: what is "the situation"? L4: what is "this condition"? Unfortunately, the entire paragraph is not very clear. To me it stays unclear, when wis which scenario calculated and why (due to which conditions). It is a general difficulty in the entire paper. Due to unclear**

**terminology, even headlines like 5.2 are not clear and it is impossible to follow the argumentation**

**Answer:** This paragraph in section 5.1 scenario design was amended as follow:

(1) An ER-SNWDP emergency requires a supply of ecological water for NL.

Taking a waterlogging occurring in the NLB after August 22, 2003, as an example, the impact of water diversion on the waterlogging process in the NLB was studied. Considering the flood control safety of the NLB, we assume that the emergency water transfer stopped at the beginning of the rainfall event on August 22, 2003 (Fig.5). Because the water diversion project is no longer operational during rainfall, the effect of water diversion on the flood process of NL mainly increases the water level of NL. Therefore, two scenarios of waterlogging of the NLB in August 2003 were simulated: the waterlogging process of the NLB under the influence of emergency water transfer in August 2003, which is recorded as scenario ② , and the waterlogging process without water diversion is recorded as scenario ① (see table 2 for scenario settings). The difference between scenario ① and scenario ② is the initial water level of NL. In scenario ① , the water levels of the upper lake and lower lake are 33.01 m and 32.20 m, respectively, as measured on August 22, 2003. In scenario ② , the water level of the upper lake and lower lake have been raised to 34 m and 32.3 m, respectively by emergency water diversion. The rainfall processes of the two scenarios were measured from August 22, 2003, to September 2, 2003.

The headlines has been revised as follow: Impacts of emergency water diversion by the ER-SNWDP on waterlogging of the NLB.

Tanks a lot.

**Page 12 L1-2 I don't understand the difference between scenarios 1 and 2, different initial water tables? If yes, what was the water table of Scenario 1? Table 2 is not clear, please re-organize it and give exact numbers.**

**Answer:** The difference between scenarios 1 and 2 has been explained in answer Page 10 L22: - Page 11: L1-2. And the Table 2 has been revised.

Thanks a lot.

**L4-7: Here, for the first time, the reason for the study is clearly described and one can follow the intention of the authors. This should be integrated into the introduction, of course in a different way, but that's the reason for the study, I guess.**

**Answer:** We have revised the introduction according to your suggestion.

Thanks a lot.

**L4-7: indicate the scenario number behind each pre-condition and refer to table 2 directly, not at the next paragraph.**

**Answer:** We have revised this paragraph as follow: Under this condition, the influence of water diversion on waterlogging disasters in the lakeside area of NL under different rainstorm intensities is analysed. The processes of 3-day designed rainfall events with return periods of 5 years, 10 years and

20 years at six precipitation stations were calculated. Affected by the emergency water diversion, the initial water levels of the upper lake and lower lake were 34.00 m and 32.30 m, respectively, in scenarios ⑥, ⑦ and ⑧.Because the water was transferred to alleviate the water shortage of Shandong Peninsula, China, the ER-SNWDP continued to operate during rainfall events. In contrast, the initial water levels of the upper lake and lower lake were 33.01 m and 32.20 m, respectively, as measured on August 22, 2003, in scenario ③, ④ and ⑤.In summary, a total of 8 simulation scenarios were set up, as shown in Table 2.

**L8: prevent one sentence paragraphs.**

**Answer:** the sentence has been add to the previous paragraph.

**L10: why are areas of 0.1 and 0.5 considered?**

**Answer:** We explained as follow: Rice, cotton, corn and soybeans are the main crops in the study region, and their waterlogging tolerance depths are 0.5 m, 0.1 m, 01 m and 0.1 m, respectively (Wang, 2015).

**L20: 0.99% of what?**

**Answer:** 0.99% of relative increase in submerged area. The sentence has been revised as follow: Emergency water diversion resulted in a relative increase of 0.99% in the submerged area. The heavy

disaster area, with a water depth of more than 50 cm increased by 51.09 km$^2$, which was 13.32% higher than that without water diversion.

**L25: EASTERN portion of the lake is mountainous...**

**Answer:** We revised this sentence as follow: Compared with the eastern NL, due to the lower and flatter terrain in the western NL, the raised water level of NL has a greater impediment to drainage in the western NL.

**L26: the interaction is either given or not. There is no option to interact larger or less. Please describe what you want to say differently: e.g. The influence of...results in increased...**

**Answer:** We revised this sentence as follow: We revised this sentence as follow: Compared with the eastern NL, due to the lower and flatter terrain in the western NL, the raised water level of NL has a greater impediment to drainage in the western NL.

**L36: compared to what, scenario 1? Please describe sharply and refer to fig. 6**

**Answer:** We revised this part as follow: Affected by the higher initial water level raised by water diversion, the flood discharge start time of the ED junction in scenario ② is 4 days earlier than that in scenario ①. Furthermore, the total amount of flood discharge in scenario ② increases by approximately 249 million m$^3$ compared to that in scenario ① (Fig.7).

Under Figure 7, we added a detailed description.

Thanks a lot.

**Page 13 L6-7: please check, which one is correct: Figure 7 a and b or the reference in the text, in the moment it is switched. Fig. 7: Please explain a and b in Figure description.**

**Answer:** We checked Fig.8, and modified it in manuscript. A detailed description has been added. Thanks a lot.

**L9-11: again, please be very clear in describing the effects of with/without active diversion.**

**Answer:** We revised the effects of water diversion as follow:

When the NLB encountered a storm with a 5-year return period, the upper lake level did not reach the flood-discharge conditions without the influence of water diversion (Fig. 8 a). Affected by water diversion, the sluices began to drain the flood water after 30 hours of rain, with a total discharge volume of 85 million m$^3$ (Fig. 9a). When the NLB encountered a rainstorm with a 10-year return period, under the condition of water diversion, the sluices began to drain the flood water after 28 hours of rain, which was 36 hours ahead of the situation with no water diversion (at the 66th hour). The total flood volume that was discharged by the ED project was 104 million m$^3$ greater than that without the effect of water diversion (Fig. 9b). When the NLB encountered a storm with a 20-year return period, under the condition of water diversion, the time that the ED project began to discharge the flood occurred 26 hours after the rain started, which was 32 hours ahead of the situation with no water diversion (at the 58th hour). Compared with the scenario without water diversion, the total

discharge volume increased 129 million m$^3$ (Fig. 9c).

**L18: in Fig. 8a it do not look like 36 hours, more like 31 hours. Wrong figure?**

**Answer:** We checked carefully and revised it in text and figure.

Thanks a lot.

**L20-25: please use some different descriptions for time: 30th hour is very uncommon and "e36 hours" is unclear to me. L25 reference is wrong, it is fig. 8c**

**Answer:** We c revised these mistakes in manuscript.

Thanks a lot.

**Page 14: Fig. 8: please explain a, b, c in the figure description.**

**Answer:** We and descriptions under Fig. 9 as follow: Figure 9: Changes in the flood discharge of the ED under water diversion with different storm levels. Flood discharge under rainfall events for different return periods: 5 years (a), 10 years (b), 20 years (c). The bar chart with different colours shows the three-day rainfall processes at each rainfall station. Black lines represent the flood-discharge process of the ED with and without the effect of water diversion.

**Impacts of the  eastern route of the South-to-North Water Diversion Project emergency operation  on flooding and drainage in  water-receiving area: An empirical case  in China**

Kun Wang[1,2], Zongzhi Wang[2], Kelin Liu[2], Liang Cheng[2], Lihui Wang[3], Ailing Ye[2,3]

[1]College of water Conservancy and Hydropower Engineering, Hohai University, Nanjing, 210098, China
[2]State Key Laboratory of Hydrology-Water Resources and Hydraulic Engineering, Nanjing Hydraulic Research Institute, Nanjing 210029, China
[3]College of Civil Engineering, Fuzhou University, Fuzhou 350002, China

*Correspondence to*: Zongzhi.Wang (wangzz77@163.com)

**Abstract.** The water levels of lakes along the eastern route of the South-to-North Water Diversion Project (ER-SNWDP) are expected to rise significantly and subsequently affect the process of flood control and drainage in  corresponding lake basins. However, few studies have focused on the impacts of inter-basin water diversion on the flood control and drainage of water-receiving areas at the lake basin scale. Using MIKE software, this paper builds a coupled hydrodynamic model to address the existing literature gap regarding  the impacts of inter-basin water diversion on the process of flood control and drainage in a water-receiving lake basin, and it considers the many types of hydraulic structures in the model. First, a flood  simulation model was constructed to simulate the complex  movement     water transferred by the ER-SNWDP, by waterlogging  in the lakeside area  around Nansi Lake (NL) and water in the NL and its tributaries. The ER-SNWDP was also considered in the model. Second, the model was calibrated and verified with measure d data, and the results showed that the model  was efficient and presented a Nash-Sutcliffe efficiency coefficient (NSE) between 0.65 and 0.99. Third, the process of flood and drain age in the lakeside area of NL w as simulated under different water diversion and precipitation values. Finally, the impacts of the emergency operations of the ER-SNWDP on flood control and waterlogging drainage in the lakeside area of NL were  analysed based on the results from the proposed model, and some implications are presented for the integrated management of  inter-basin water diversion and  affected lakes.

**1 Introduction**

Inter-basin water diversion is a useful approach to solving the spatial unevenness of water resources,  however, it makes the water cycle of the water-receiving area more complicated and brings a great challenge to the integrated water management of the water-receiving area and water diversion project (Matete and Hassan, 2006; Webber et al., 2017). The key to the long-term healthy operation of the inter-basin water transfer project is to clarify the influence of inter-basin water transfer on the water-receiving area. In recent decades, several authors have revealed and discussed the impacts of inter-basin water transfer on water-receiving areas from many perspectives, which mainly include water quality, water resources and water ecosystems, and some  suggestions regarding the scientific management of water-receiving areas and the water diversion project were proposed (Cole et al., 2011; Emanuel et al., 2015; Zhuang, 2016). A large quantity of transferred water not only  changes the water quality,  water environment and other hydrological characteristics of water-receiving areas but also impacts flood control and waterlogging drainage in some water-receiving areas (Gupta and Zaag, 2008; Liang et al., 2012). Flood ing and waterlogging are  two of the main natural disasters in terms of losses of human life and economic damage, and flood control and drainage are the top priorities of watershed management (Arrighi et al., 2013; He et al., 2018; He et al., 2018b).

However, to the best of our knowledge, there are very few studies on the impacts of inter-basin water transfer on flood control and drainage in water-receiving areas from a watershed perspective.

The eastern route of the South-to-North Water Diversion Project (ER-SNWDP) links Gaoyou Lake, Hongze Lake, Luoma Lake, Nansi Lake (NL) and Dongping Lake with 13 pump stations that transfer water from  downstream of the Yangtze River to the Huang-Huai-Hai Plain and Shandong Peninsula (Fig.1a and 1b). According to the comprehensive plan of the SNWDP, (1) the first planning phase (before 2030) of the eastern route is designed to transfer 8.9 billion $m^3$ of water annually, and approximately 7 billion $m^3$ is expected to be consumed in the above five lake basins and route; (2) the water diversion period covers the non-flood season (October to the following May), and the water diversion ceases for the rest of the time (Bureau of South to North Water Transfer of Planning, Designing and Management, Ministry of Water Resources, 2003). Lake basins along the ER-SNWDP, which are also called water-receiving areas for large quantities of water are consumed in these basins during the water transfer period. As water-receiving areas and transmitting channels of the ER-SNWDP, these five lake are used to store and regulate water resources, and the water levels of the five lakes are significantly increased when the project is operating. The increased lake level impedes flood control and waterlogging drainage in the water-receiving lake basin  especially  in low-lying lake basins.

In addition, the Shandong Peninsula has suffered from severe drought and the water supply cannot meet the water demand even in the flood season (June to October) for four consecutive years since the eastern route began  operation in 2013. Emergency water diversion (i.e., water  transfer through the water diversion project in flood seasons to alleviate water shortages in water-receiving areas) has been performed many times to supply water to the  Shandong Peninsula. Furthermore, considering the  rigorous demand for water resources caused by rapid socio-economic development , more frequent water transfers are expected in the flood season (June to October) (Guo et al., 2018). Meanwhile, extreme  rainfall events caused by  climate changes have increased in eastern China (Liu et al., 2015). Thus, the probability of rainstorms during the water diversion period in these lake basins will increase. To strengthen the scientific scheduling of flood control projects in water-receiving areas and water diversion project, it is necessary to clarify the influence law of water diversion on flood control and drainage in these lake basins.

Based on data availability and regional distribution, the N Basin (NLB), which is a flood prone area, was chosen as the research area in this study (Fig. 2). The reason why we chose the NLB is that the NL  is an important storage node of the ER-SNWDP, as the NL is the largest freshwater lake in northern China and has the largest water storage capacity among the lakes along the ER-SNWDP (Zhang, 2009). In addition, the NLB has a history of frequent flooding and waterlogging disasters due to the low  drainage capacity  in the geomorphic low-lying area around NL (i.e., lakeside area) . NL is a storage pond of the ER-SNWDP; the flow of water transferred into NL is 200 $m^3/s$, and the outflow is 100 $m^3/s$. According to the overall plan of the SNWDP, the water level of the upper lake is expected to rise by 50 cm, and that of the lower lake is expected to rise by 70 cm during the project operation period. A large amount of water diversion has a significant impact on the hydrological situation of NL. The impacts have instigated several studies that aimed to understand the ER-SNWDP effect on the NLB mainly focusing on environmental and water resource management (Ma et al., 2006; Wu et al., 2011; Zhang, 2009; Zhao et al., 2017). The lake level increased by the ER-SNWDP will decrease the drainage efficiency of pump stations , hinder the flood discharge of rivers in lakeside area, and then influence the flood control and waterlogging drainage of the NLB. However, the impacts of water diversion on flood control and waterlogging drainage in the NLB are relatively unexplored. A gap in assessing the impacts of inter-basin water diversion on

flood inundation in the NLB still exists. An integrated model that could explicitly simulate the impacts of water transfer project operation on the spatiotemporal aspects of inundation is one of the necessary tools that can help bridge this gap. Therefore, this This paper study aims to bridge the knowledge gap on the impacts of water diversion by the ER-SNWDP on flood control and inundation in the NLB. For this purpose, the following were set as the two main objectives of this study: 1) to develop one-dimensional and two-dimensional hydrodynamic models that can simulate both the ER-SNWDP operations and its impacts on the flooding and waterlogging processes within the NLB, and 2) to clarify the impacts of the ER-SNWDP on flood control and waterlogging drainage in the NLB under different rainstorm events. More specifically, the following sub-objectives will be addressed in this paper: proposes(1) develop a model that integrates MIKE 11, MIKE 21 and MIKE FLOOD to simulate the flooding and waterlogging processes in the NLBakeside areas under the condition of emergency water diversion by the ER-SNWDP, (2) validate the adequacy of the model in simulating flooding and waterlogging processes in the NLB, (3) and simulate flooding and waterlogging processes with the effects of the water diversion project under rainstorms with different return periods, and (4) perform a detailed analysis analyses of the impacts of emergency water diversionsthe inter-basin water transfer on flood control and waterlogging drainage in NLB. drainage characteristics and the scheduling of flood control and drainage projects must be performed to strengthen the scientific scheduling of water diversion projects and flood control projects in water receiving regions. This situation represents a major scientific problem that must be resolved. However, to the best of the authors' knowledge, few studies have focused on the impacts of interbasin water transfer on flooding and drainage in lake basins along the water diversion route. Therefore, an important gap exists in the literature, Apart from a few papers, the effects of water level increases caused by water diversion on the flooding and drainage process and inundated areas in lake basins have not been estimated via relevant models. the existing literature and address the gap mentioned above.

This paper attempts to address three issues:

(1) Identify a method of building a flood and waterlogging simulation model for a water receiving region with multiple hydraulic structures under water diversion;

(2) Determine how emergency water diversion affects the flooding and drainage process in the NLB and analyze the waterlogging situation in the lakeside area of NL;

(3) Evaluate how to balance the risk of water shortage and flooding caused by interbasin water diversion in the water receiving regions.

(a)                                                                                          (b)

[Figure]

**Figure 1: Sketch of the eastern route of the South-North Water Transfer Project (a). Annual rainfall contours and lakes along the route (b).**

**2 Literature review**

**2.1 Impacts of inter-basin water diversion on the water-receiving regions**

5    Due to the uneven spatial distribution of water resources and regional socio-economic development, the demand for water in certain regions far exceeds the available water amount, thereby resulting in an increasingly serious imbalance between water demand and supply (Cai and Ringler, 2007; Hu et al., 2010; Matete and Hassan, 2006; Webber et al., 2017). As the most effective and direct method of resolving the problem of water resource shortages problem, inter-basin water diversion projects have been widely applied in water-deficient areas around the world (de Andrade et al., 2011; Wang et al., 2014;

10    Zhang et al., 2015). According to data released by the International Commission on Irrigation & Drainage (ICID) (2005), the total annual amount of water transferred by water diversion projects around the world is 540 billion m$^3$, which accounts for approximately 14% of the world's annual water withdrawals. By 2025, the annual water diversion is expected to reach 940 billion m$^3$. In the water supply and receiving areas, inter-basin water transfer projects significantly affect hydrological elements, such as the water quantity, water quality, the water environment and flood disasters. A full understanding of these

15    impacts is key to the scientific management and long-term operation of inter-basin water diversion and represents the most popular global topic in water resource planning and management research (Aron et al., 1977; Davies et al., 1992; Khan et al., 1999; Liu and Zheng, 2002). Zhang et al. (2015) summarized relevant studies on inter-basin water transfer from 1991 to 2014 and noted that the effects on the hydrological environment caused by China's SNWDP and the corresponding long-term monitoring and protection policy for this project represent the most important current issues.

20    Current research on the hydrological effects caused by inter-basin water transfer mainly focuses on the following aspects. (1) For groundwater, Kundell (1988) argues that a large amount of imported water significantly increases the amount of available water and directly participates in the water cycle of the in water-receiving regions, which has a positive effect on the water environment, groundwater exploitation and wetland restoration. Relevant studies have indicated that a large amount of imported water can effectively alleviate the problem of decreased groundwater levels and ground subsidence

25    caused by the perennial over-extraction of groundwater in selected areas (Larson et al., 2001; Liu and Zheng, 2002; Wang et

al., 2014). Based on a large hydrological distribution model, Ye et al. (2014) evaluated the effect of the middle route of the SNWDP on the groundwater level of the Haihe Basin. The results showed that although imported water cannot change the decreasing trend of the groundwater level in the water-receiving area, it can significantly reduce the rate of decrease. (2) Water quality is one of the most important factors underlying the success of inter-basin water transfer projects. Scholars have simulated and evaluated the effects of inter-basin water transfer projects on water quality in water supply and receiving areas. Imported water dilutes the concentration of nutrients, improves the ratio of runoff and pollution in the receiving area and subsequently improves the water quality. However, inter-basin water transfer projects might also transfer pollutants from the water supply area or river basin along the water diversion line into water-receiving regions, thus worsening the water quality (Hu et al., 2008; Karamouz et al., 2010; Tang et al., 2014; Welch et al., 1992; Zhai et al., 2010). The Chicago inter-basin water diversion project, which uses the Lake Michigan basin as its source, has received criticism due to its chronic exposure risk of organic pollutants (Rasmussen et al., 2014). (3) Inter-basin water transfer brings water from the water-supply area to the water-receiving area through the water transmission channel, which is not conducive to flood control in water-receiving areas and water transmission channels. Wang et al. (2013) studied the influence of an inter-basin water transfer project on hydraulic parameters during the flood season in a water-supply area. Based on a two-dimensional mathematic model, Sun et al. (2008) used the Anyang River Basin, which intersects with the middle route of the SNWTP, as an example to study the influence of the water diversion project on flooding in the river basin that the project passes through. As the storage node of the water diversion project, the large amount of water transferred into the lake significantly changes the interaction law between the water body of the lake basin and the water in the lake tributaries, which subsequently affects the flood control and drainage of the lake basin. However, few quantitative studies have focused on this issue.

**2.2 Simulation of flooding and waterlogging disasters in a basin**

Simulating the flooding and waterlogging processes based on a mathematical model is an important method for analysing flooding and waterlogging characteristics and assessing the flooding and waterlogging disaster risk of a basin; this simulation is also an effective tool for planning the engineering layout of flooding and waterlogging control engineering (Dutta et al., 2015; Liu et al., 2015; Wang et al., 2018). The early flooding and waterlogging simulations of a basin are mainly based on hydrological models, including the Storm Water Management Model (SWMM) (Lee and Heaney, 2003), Model for Urban Storm water Improvement Conceptualisation (MUSIC) (Dotto et al., 2011; Hamel and Fletcher, 2014), Soil and Water Assessment Tool (SWAT) (Dixon and Earls, 2012), and MIKE SHE (Vrebos et al., 2014), among others. However, hydrological models only simulate the flood-routing process according to the water balance equation and are unable to display the spatial distribution of flood movement. In addition, these models cannot accurately simulate the drainage process of sluice, dam, pumping station and pipeline hydraulic structures. A hydrodynamic model simulates water routing by solving the Saint-Venant equations, which can accurately reflect the movement of water on a plane in and various hydraulic structures. With improvements in computer processing speed and the development of spatial digital elevation information, hydrodynamic models have gradually become an important tool for flood simulations (Moel et al., 2015). Hsu et al. (2000) built a waterlogging simulation model by coupling the SWMM model with a two-dimensional hydrodynamic model and simulated the rainstorm waterlogging in Taipei. Bisht et al. (2016) combined the SWMM with the MIKE URBAN and MIKE 21 models to simulate waterlogging in West Bengal, India. Li et al. (2016) established 1-D and 2-D coupled hydrodynamic models of Taining County in China based on the MIKEFLOOD model and simulated and analysed the flooding and waterlogging risks in the region. The MIKE 11 and MIKE 21 hydrodynamic models can simulate the influence of a variety of hydraulic structures on the water flow movement process, and the one-dimensional and two-dimensional models can describe this coupling in different ways. Therefore, this model has

been applied to simulate the flow movement of a variety of water bodies, including rivers, lakes, flood water on the groundland and estuaries (Karim et al., 2016; Quan, 2014; Zolghadr et al., 2010).

**3 Research area and data sources**

**3.1 Research area**

5  NL (34°27'N–35°20'N, 116°34'E–117°21'E) is composed of four consecutive lakes, namely, (i.e., Nanyang Lake (NY), Dushan Lake, Zhaoyang Lake and Weishan Lake (WS) (Fig. 12)), and it is a typical large and shallow lake, with an area of 1266 km² and an average depth of only 1.5 m (An and Li, 2009). To manage flooding in this basin, a pivotal project composed of a dam and sluices (Erji dam (ED)) was constructed at in the middle of NL, and itwhich divides the lake into an upper lake and lower lake. The sluices of the EDErji dam control the flood discharge of the upper lake, and the Hanzhuang

10  sluice and Linjia sluice control the flood discharge of the lower lake. The NLB is located in the Yi-Shu-Si river system of the Huaihe River Basin with an area of 31700 km². The lakeside area, refers to the area with a ground elevation (above sea level) of below 36.79 m or less around the lake, is approximately 3969 km² and has the grounda slope in this area slope is between 1/50000.0029 ° and 1/200000.0057 ° (Tian et al., 2013; Wang et al., 2010). A total of 53 rivers flow into NL, and 11 of them have a drainage area greater than 1000 km². Due to the low height and gentle slope of the riverbed, the rivers in the lakeside

15  area have strong interactions with NL. Flood control embankments have been built on both sides of the main inflow channels and around NL to prevent flooding from entering the lakeside area. Due to the low-lying terrain and the construction of flood control embankments, waterlogging in lakeside areas can not longer directly drain into rivers and lakesNL. The waterlogging waterlogged water in the lakeside area is primarily carriedmainly pumped into rivers and NL by through pumping stations in the lakeside area. However, the existing pumping stations in this region cannot resist rainstorm waterlogging every five

20  years, and some pumping stations cannot resist rainstorm waterlogging every three years. When Eencountering heavy rainstorms, the water cannot be discharged in time; therefore the NLB is historically known as a flood prone areabelongs to the historical frequent flooding area.

[Figure]

**Figure 12: Location of the study area, with the positions of the rainfall and water level gauges, rivers and hydraulic projects considered in the river system and a topographic map of the NLB. Sketch map of the eastern route of the South-North Water Transfer Project and Nansi Lake BasinThe logo of Copernicus Publications. The area surrounded by the black line is the lakeside area, which is defined according to the suggestion of the Planning and Design Institute of the Huaihe Basin Hydraulic Management Bureau in Shandong Province, China.**

The ER-SNWDP transfers water from Yangzhou City to NL and is divided into two sections in the NLB: water entering the lower lake at the Hanzhuang sluice (HS) and water entering the Linjia dam sluice (LS). A pumping station has beenwas built at the secondary dam to lift water from the lower lake to the upper lake. and tThe Changgou pumping station is built 24 km south of the Liangji River Estuary in the upper lake will be built at the higher lake 24.6 km the Liangji River mouth to send transfer water to the Shandong Peninsula. According to the first phase planning of the ER-SNWDP, the discharge that enters the lower lake occurs atis 200 $m^3$/s, and 5/8 of this amount is will be pumped into the upper lake. When the water diversion project is runningDuring project operation, the water level (with respect to mean sea level) of the lower lake will reaches 32.8 m, which is 0.70 m higher than the mean annual water level. The project also transfers water from the upper lake to the north at the a flow rate of 100 $m^3$/s. The water level of the upper lake maintains a normal storage level of 34 m, which is 0.48 m higher than the mean annual water level.

**3.2 Data sources**

The data sources for this study include terrain, hydrologicaly, meteorologicaly and hydraulic engineering data from the NLB and the engineering layout and operation data of the eastern route of the ER-SNWDP. (1) The digital elevation model (DEM) and river channel bathymetry were supplied by the Planning and Design Institute of the Huaihe Basin Hydraulic

Management Bureau in Shandong Province, China. The lakeside area of NL is also provided by this Planning and Design Institute. The DEM of the lakeside area and NL in 2013 was derived from 1:7000 topographic maps, and the river channel bathymetry of all rivers simulated used in the 1-D model was reflected by 550 cross-sections separated by at distances between 500 and 1000 m. (2) The hydrological data originated from the Shandong Provincial Hydrology Bureau. These data

5    mainly include the discharge processes of typical floods in the upper boundaries of the rivers and the daily rainfall records at seven six rainfall stations: Huayu (HY), Liangshanzha (LSZ), Wanglu (WL), Wanggudui (WGD), Wangzhong (WZ) and Xuecheng (XC). Each station has daily precipitation records covering approximately 30 to 50 years. The daily water level records of four stations, (NanyangNY, Makou (MK), Erji LakeED, and Weishan WS (shown in Fig. 12)), were also supplied. (3) The meteorological data were downloaded from the National Meteorological Scientific Data Sharing Service Platform

10   (http://data.cma.cn/), and the data include the daily records of numerous meteorological parameters, including the wind field and evapotranspiration information. (4) The hydraulic engineering data, including the technical parameters of sluices, pumping stations and levees in the NLB, were supplied by the Planning and Design Institute of the Huaihe Basin Hydraulic Management Bureau in Shandong Province, China. These data include the location and drainage capabilities of the pump stations, the locations and sizes of the flood control embankments, and the hydraulic parameters of the sluices. (5) The

15   engineering data of the ER-SNWDP were supplied by the Planning and Design Institute of the Huaihe Basin Hydraulic Management Bureau in Shandong Province, China.

**4 Methodology**

**4.1 Research framework**

To quantitatively study the impact of the water diversion project on flood control and drainage in the NLB, a hydrodynamic

20   model of waterlogging in lakeside areas of NL was constructed based on MIKE software with consideration of the ER-SNWDP. This model includes a one-dimensional model to simulate flood routing in the river that flows into NL (MIKE 11) and a two-dimensional model to simulate the evolution of plane flow in the lakeside area of NL (MIKE 21). Different hydraulic structures are set up in the model to simulate the flood control and drainage processes of sluices, dams, pumps and other hydraulic structures in the research area and the water lifting process of the ER-SNWDP pumping station. Coupling of

25   the one-dimensional and two-dimensional models is performedsupplied by reasonable links to reflect the interaction of water in theamong NL, tributary rivers and the lakeside area. The established model is used to simulate the waterlogging process in the lakeside area under different scenarios. According to the results of the calculation, the influence law of the SNWDP on waterlogging in the lakeside area of NL is analyszed. Finally, related suggestions for balancing the water diversion and waterlogging risk are proposed. Fig. 2 3 illustrates the research framework of this paper.

[Figure]

**Figure  _3_: Research framework of the interbasin water diversion influence on _the_ flood control and waterlogging drainage of lakes along the route.**

**4.2 One-dimensional hydrodynamic model of _the_ river net**

5   A total of 53 tributary rivers are located around NL, and they represent the key to studying the influence of water diversion on waterlogging in the lakeside area for accurately simulating the flood evolution and  interaction between flood evolution and the high water level of NL. Therefore, a 1-D mathematical model (MIKE 11) was used to simulate  flood routing. The control equation of the model is the Saint-Venant equation _(Abbott et al., 1979)_, which  _is_ composed of  _a_ continuity equation and momentum equation :

10   continuity equation:

$$B = \frac{\partial z}{\partial T} + \frac{\partial Q}{\partial x} = q \ , \tag{1}$$

momentum equation:

$$\frac{\partial Q}{\partial t} + \frac{\partial}{\partial X}\left(\frac{\alpha Q^2}{A}\right) + gA\frac{\partial z}{\partial x} + \frac{gQ|Q|}{C^2 AR} = 0, \tag{2}$$

where $x$ and $t$ denote the spatial fland temporal coordinates, respectively; $A$ represents the cross-sectional area; $Q$ and $z$

15   denote the discharge and water level of _the_ cross-section, respectively; $q$ represents the lateral inflow; $R$ represents the

hydraulic radius; *C* representsis the Chezy coefficient; *a* representsis the momentum correction factor; and *g* denotes the gravitational acceleration. The Abbott-Ionescu 6 implicit difference method is used to solve the equation.

First, we generalized the river network by considering based on a consideration of the data and computational efficiency. This river network primarily contains 11 rivers, with drainage areas greater than 1000 km². A total of 550 cross--sections were input into the river network model to reflect the changes of in river topography, with adjacent sections spaced approximately 1000 m apart. We generalized the drainage pump stations in the lakeside area of NL. A total of 1000 draining pump stations are used to drain waterlogged water from present in the the lakeside area to of rivers and NL, and the model generalized the pump stations according to the distribution of each pump station distribution on both sides of the rivers. The basic principle is to ensure that the total drainage discharge remains the same, with the generalized pumping stations along both sides of the river are on both sides evenly distributed for a total of 41 generalized 41 pumping stations. Finally, the boundary conditions of the model were set. The upper boundary of the model inputs the a discharge hydrograph of the upstream hydrological station of each river. As the lower boundary, the water level of the estuary is based on NL, which is simulated by the two-dimensional model.

**4.3 Two-dimensional hydrodynamic model of NL and the lakeside area**

The MIKE 21 hydrodynamic model was used to simulate the water movement in NL and the waterlogging evolution in the lakeside area. The model covers the area below the 36.79 m contour line around NL and the lake surface. The model area includes NL and the lakeside area around the lake (Fig.2). Because the one-dimensional model is adopted to simulate the rivers around the lake, the area of the rivers is removed from the two-dimensional model. The two-dimensional model is based on the Reynolds average stress equation of a three-dimensional incompressible fluid, which is subject to the Boussinesq hypothesis and the hydrostatic pressure hypothesis (J V Boussinesq, 1872), and the control equations used in this model are given as follows (DHI, 2007):

continuity equation:

$$\frac{\partial h}{\partial t} + \frac{\partial h\bar{u}}{\partial x} + \frac{\partial h\bar{v}}{\partial y} = hS, \tag{3}$$

momentum equation of in the x direction:

$$\frac{\partial h\bar{u}}{\partial t} + \frac{\partial h\bar{u}^2}{\partial x} + \frac{\partial h\bar{v}\bar{u}}{\partial y} = f\bar{v}h - gh\frac{\partial\eta}{\partial x} - \frac{h}{\rho_0}\frac{\partial p_a}{\partial x} - \frac{gh^2}{2\rho_0}\frac{\partial\rho}{\partial x} + \frac{\tau_{sx}}{\rho_0} - \frac{\tau_{bx}}{\rho_0} - \frac{1}{\rho}\left(\frac{\partial s_{xx}}{\partial x} + \frac{\partial s_{xy}}{\partial x}\right) + \frac{\partial}{\partial x}(hT_{xx}) + \frac{\partial}{\partial x}\left(hT_{xy}\right) + hu_sS, \tag{4}$$

momentum equation of in the y direction:

$$\frac{\partial h\bar{v}}{\partial t} + \frac{\partial h\bar{v}^2}{\partial y} + \frac{\partial h\bar{u}\bar{v}}{\partial x} = f\bar{u}h - gh\frac{\partial\eta}{\partial y} - \frac{h}{\rho_0}\frac{\partial p_a}{\partial y} - \frac{gh^2}{2\rho_0}\frac{\partial\rho}{\partial y} + \frac{\tau_{sy}}{\rho_0} - \frac{\tau_{by}}{\rho_0} - \frac{1}{\rho_0}\left(\frac{\partial s_{yx}}{\partial y} + \frac{\partial s_{yy}}{\partial x}\right) + \frac{\partial}{\partial x}\left(hT_{xy}\right) + \frac{\partial}{\partial y}\left(hT_{yy}\right) + hv_sS, \tag{5}$$

where *x*, *y* and *z* are Cartesian coordinates; *t* denotes the temporal coordinates; $\eta$ represents is the bottom elevation of the river; *d* representsis the depth of the water, $h=d+\eta$ represents is the total head of the water; *u*, *v* and *w* representare the velocity components in the *x*, *y* and *z* directions, respectively; $p_a$ representsis the local atmospheric pressure; $\rho$ represents is the density of water; $\rho_0$ representsis the reference water density; $f=2\Omega sin\phi$ representsis a Coriolis parameter; $f\bar{v}$ and $f\bar{u}$ representare the acceleration caused by the earth's rotation; $S_{xx}$, $S_{xy}$, $S_{yx}$ and $S_{yy}$ representare the components of the radiation stress tensor; $T_{xx}$, $T_{xy}$, $T_{yx}$ and $T_{yy}$ representare the horizontal viscous stresses; *S* representsis the magnitude of the discharge due to point sources; and $(u_s, v_s)$ denotesis the velocity by which the water is discharged into the ambient water.

The total area of the 2-D model is approximately 4750 km², including NL (1266km²), the lakeside area (3696km²) shown in Fig.2, and removes the area of the rivers simulated by the 1-D model, with both sides of the river embankment and the 36.79 m contour line as the outer boundaries of the model. Dikes were set up on both sides of the river around NL and at the Erji damED to simulate the flood control effect of levees. Sources were added to simulate the water transfer process of the ER-

SNWDP. When the model is running, the rainfall process of the HY, LSZ, WL, WGD, XC, and WZ  rainfall stations in the research area are input.

**4.4 Waterlogging simulation model of NL and the lakeside area  when considering the SNWDP**

The MIKE FLOOD model is used to couple the 1-D and 2-D model, and the specific process is described as follows: (1) A lateral link is applied to connect the lakeside area  and the tributaries of NL to simulate the flood exchange between the lakeside area and tributaries, and (2) a standard link is applied to connect the into-lake rivers and NL to reflect the influence of the lake level height in blocking the drainage of into-lake rivers. A total of 22 lateral connections and 52 standard connections are present in the coupling model.

**4.5 Calibration and validation of the coupling model**

The model is calibrated and validated using two actual floods that occurred in July 2007 and July 2008 in the NLB. Fig. 4 shows the simulated and measured water levels at four stations: (NY, (MK, (ED and (WS in NL. Overall, the measured water level process shows good agreement with the simulated water level process, and the arrival of the simulated flood peak is consistent with that of the measured data. The Nash-Sutcliffe efficiency (NSE) coefficient, which was proposed by Nash and Sutcliffe (1970), is used to evaluate the coupling model. The NSE for the daily flow varied from 0.67 (ED) to 0.82 (WS) during calibration and from 0.65 (NY) to 0.99 (WS) during verification (Table 1), thus showing good agreement between the observed and simulated water levels. As a result, the calibrated roughness coefficients, n, were 0.055 for agricultural fields, 0.08 for residential areas in the lakeside area, and 0.028 for NL.

[Figure]

**Figure 4: Comparison of the observed and simulated water levels at the selected locations for 2007 flood events under calibration conditions (left) and 2008 events under validation conditions (right). Black dots represent observed data, and black lines represent model simulation results.**

**5 Results and discussion**

**5.1 Scenario design**

NL is located in  China's north-south climate transition zone, and the temporal distribution of rainfall is severely uneven, with rainfall in the flood season accounting for 72% of the annual precipitation. This basin is located in the north-south climate transition zone of China, and the phenomenon of drought-flood abrupt alternation (DFAA) frequently occurs. DFAA refers to a rainstorm after a long period of drought and can result in severe flood damage.  Fig. 4 shows the process of rainfall at Wanglu station and the water level of the upper and lower lakes at Nanyang and Weishan stations in 2003. Since there was no precipitation in the first half of the year, NL dried up from April to June 2003, which had a devastating impact on the ecosystem of NL. In the future, if the NLB encounters another drought, such as that in 2003, emergency water diversion by the ER-SNWDP will be the first choice to provide ecological water for NL. However, heavy rains occurred after August 22 and caused a steep increase in the water level in NL, and 2003 was a typical year of DFAA. The statistical data show that the waterlogged area of the NLB is 2360 km², and waterlogging disasters resulted in a great economic loss in the lakeside area.  Therefore, we take 2003 as an example to research the impacts of emergency ecological water diversion by the ER-SNWDP on waterlogging in the NLB.  In recent years,  water resource shortages have become increasingly intense in Shandong Peninsula, China. Even in the flood season, local water resources still encounter difficulty in meeting  demands; thus, the supply of water to the region that relies on the ER-SNWDP is expected to be more frequent.

[Figure]

**Figure 4: Rainfall at Wanglu station in the NLB and water level variations in the upper lake and lower lake in 2003. The upper lake is dry when the water level of  NY is 31.5 m. Before July 2003, the NLB suffered severe drought, but heavy rains in August caused serious floods.**

To analyse the influence of water diversion on waterlogging disasters in the lake basin along the ER -SNWDP, this paper set two conditions in which the ER-SNWDP supplies an emergency transfer of ecological water to NL and  Shandong Peninsula. Ecological water refers to the transferred water need to maintain the normal development and relative stability of all types of ecological systems in NL during the drought period and prevent the recurrence of the dry lake situation that occurred in 2003.

(1) An ER-SNWDP emergency requires a supply of ecological water for NL.

Taking a waterlogging occurring in the NLB after August 22, 2003, as an example, the impact of water diversion on the waterlogging process in the NLB was studied. Considering the flood control safety of the NLB, we assume that the emergency water transfer  stopped  at the beginning of the rainfall event on August 22, 2003 (Fig.5). Because the water diversion project is no longer operational during rainfall, the effect of water diversion on the flood process of NL mainly increases the water level of NL.  Therefore, two scenarios of waterlogging of the NLB in August 2003 were simulated:  the waterlogging process of the NLB under the influence of  emergency water transfer in August 2003, which is recorded as scenario ②, and the waterlogging process without water diversion is recorded as scenario ①  (see table 2 for scenario Settings). The difference between scenario ① and scenario ② is the initial water level of  NL. In scenario ①, the water level of the upper lake and lower lake are 33.01 m and 32.20 m, respectively, as measured on August 22, 2003. In scenario ②, the water level of the upper lake and lower lake have been raised to 34 m and 32.3 m, respectively by  emergency water diversion. The rainfall process of the two scenarios  were measured from August 22, 2003, to September 2, 2003. ~~During the water diversion period, the water level of NL maintained a normal storage level, with water levels in the upper lake and lower lake of 34 m and 32.8 m, respectively. However, we consider that emergency water diversion occurs during the flood season. The initial water levels of the upper lake and lower lake are 34 m and 32.3 m when rainfall begins in the 2 D model, respectively. This situation is recorded as scenario ②, and the situation in which the waterlogging process occurs without water diversion is recorded as scenario ① (Table 2).~~

(2) An ER-SNWDP emergency requires a supply of water for Shandong Peninsula, China.

 Under this condition, the influence of water diversion on waterlogging disasters in the lakeside area of NL under different rainstorm intensities is analysed. The processes of  3-day designed rainfall events with return periods of 5 years, 10 years and 20 years at six precipitation stations were calculated. Affected by the emergency water diversion, the initial water levels of the upper lake and lower lake were 34.00 m and 32.30 m, respectively, in scenarios ⑥, ⑦ and ⑧. Because the water was transferred to alleviate the water shortage of Shandong Peninsula, China, the ER-SNWDP continued to operate during rainfall events. In contrast, the initial water levels of the upper lake and lower lake were 33.01 m and 32.20 m, respectively, as measured on August 22, 2003, in scenario ③, ④ and ⑤. In summary, a total of 8 simulation scenarios were set up, as shown in Table 2.

**5.2 Impacts of  emergency water diversion by the ER-SNWDP  on  waterlogging of the NLB.**

Rice, cotton, corn and soybeans are the main crops in the study region, and their waterlogging tolerance depths are 0.5 m, 0.1 m, 01 m and 0.1 m, respectively (Wang, 2015). Therefore, the areas with inundated depths above 0.1 m and 0.5 m are counted in the simulation results. In the calculation results of scenario ① and scenario ②, the areas with  submerged depth larger than 0.1 m and 0.5 m are counted . Table 3 shows that the rainfall events from August 22 to September 2, 2003, under the condition of no water diversion, which caused the inundated area in the lakeside  region to reach 1126.59 km², and the area with a submerged depth over 0.5 m reached 383.68 km². Statistical data show that the total waterlogging area under the 36.79 m contour line was 1284.21 km² in 2003. The simulation result is slightly smaller than the

survey result because the simulation did not cover the entire year, and rain remained in the basin after September 2, 2003. In general, the simulation results can be considered reasonable.

Table 4 shows the increase in the waterlogging area in the lakeside area of NL under the condition of water diversion in the ER-SNWDP compared with that without water diversion. When the phenomenon of DFAA

5    occurs, the lake level increase via the emergency operation of the ER-SNWDP during the flood season increases the waterlogging intensity in the lakeside area of NL. Compared with the situation without water diversion, the area with a submerged depth over 10 cm increased by 34.26 km². Emergency water diversion resulted in a relative increase of 0.99% in the submerged area.  The heavy disaster area, with a water depth of

10    more than 50 cm increased by 51.09 km², which was 13.32% higher than that without water diversion.

Fig. 6 shows the waterlogging distribution in the lakeside area of NL under two scenarios. The comparison in Fig. 6  and  shows that the water diversion primarily increased the waterlogging area between the Dongyu River and Wanfu River in the western region of NL and had a relatively small impact on the eastern area of NL. Compared with the eastern NL, due to the lower and flatter terrain in the western NL, the raised water level of NL has a greater impediment to drainage

15    in the western NL. ~~Because of the lower and flat terrain in the western part of NL, the high water level of NL has a greater impediment to the drainage in the western part of the lakeThe primary reason for this observation is that the western portion of NL is mountainous, whereas the western portion of NL is a low-lying plain. The interaction between NL and the western rivers is stronger than that between NL and the eastern rivers.thes in6~~7 presents the stage hydrograph for

20    the M hydrographic station and the flood discharge of the E for two scenarios. Fig. 7 indicates that emergency water diversion has an obvious influence on the water level of NL during the initial period of rain, and the regulation of the E junction leads to a decrease in the water level difference between the two scenarios. Emergency water diversion increases the initial water level of NL at the beginning of the rainfall event, and the water level of the upper lake in scenario ② first reaches the water level at which the E begins flood drainage. With the increase in the water

25    level, the discharge of the E also increases, which satisfactorily adjusts the water level of the upper lake. Affected by the higher initial water level raised by water diversion, the flood discharge start time of the E junction  in scenario ② is 4 days earlier than that in scenario ①. Furthermore,  the total amount of flood discharge in scenario ② increase by approximately 249 million m³ compared to that in scenario ① (Fig.7).

[Figure]

a) Calculation result of scenario 1                    b) Calculation result of scenario 2

30    **Figure 6: Distribution of waterlogging in the lakeside area of NL.**

[Figure]

**Figure 67**: Water level at the of MKakou station in the upper lake and the discharge of flood drainage of the EDrji dam from August 22 to September 2, 2003, under scenario 1 (without water diversion) and scenario 2 (with water diversion).

**5.3 Impacts caused by ER-SNWDP emergency water diversion for Shandong Peninsula during waterlogging of the NLB.**

Table 5 shows the simulation results of waterlogging in the lakeside area in the case of  the diversion and non-diversion of water by the ER-SNWDP to the Shandong Peninsula when  designed rainfall events with 5- year, 10- year and 20-year return periods occurs in the basin. Emergency water diversion has certain effects on the waterlogged area in the lakeside area of NL. According to the comparison of scenarios ③ and  ⑥, when the NLB encounters rainfall events with a 5-year return period, the areas with  submerged depths over 0.1 m and 0.5 m increase by 22.27 km² and 26.14 km², respectively, under the condition of emergency water diversion.

Fig.  8 illustrates the relative increase in the waterlogging area and its change trend in the lakeside area of NL under three designed rainfall conditions with water diversion and non-water diversion. Graph (a) shows the contrast of the waterlogged area with an inundated water depth above 0.1 m under different rainfall events, and graph (b) shows the waterlogged area with an inundated water depth above 0.5 m. The black lines in graph (a) and (b) both show a downward trend, which indicates that The influence of emergency water diversion on waterlogging in the lakeside area decreases with increasing rainfall. Affected by the emergency water diversion, the relative increase in the waterlogging area with an inundated water depth above 0.1 m is between 1.5% and 2.8% (Fig. 7(a)),  and the relative increase in the waterlogging area with an inundated water depth above 0.5 m is between 8.4% and 43.1% (Fig. 7(b)) when  storm with 5-year, 10-year and 20-year return periods occur in the NLB. Under the same rainfall conditions, such as rainstorm events with 5-year return periods, the areal change in submerged water depth greater than 0.5 m (43.1%) in the lakeside area is obviously larger than that of submerged water depth greater than 0.1 m (2.8%). The calculated results indicate that emergency water diversion has more obvious effects on the waterlogging area with deeper (0.5 m) submerged water.

[Figure]

**Figure 78: Changes**  **in waterlogging areas in the lakeside area**  **under the influence**  **of water diversion (shaded bar), and no-diversion (black bar). The black line represents the trend of the relative increase in waterlogged area caused by water diversion.**

5     The emergency water diversion of the SNWDP alleviated the drought situation in the Shandong Peninsula area and  increased the degree of waterlogging in the NLB under abrupt drought-flood alternations. The sluices of the E are the only flood-discharge channels of the upper lake, and the increased water in the upper lake due to water diversion also  increased the task of flood discharge in those sluices. Fig.  9 shows the flood-discharge process of the E dam project under different design rainfall events. When the NLB encountered a storm with a 5-

10     year return period, the upper lake level did not reach the flood-discharge conditions without the influence of water diversion (Fig. 8 a).  Affected by water diversion, the sluices began to drain the flood water after  30 hours of rain, with a total discharge volume of 85 million m³ (Fig. 89). When the NLB encountered a rainstorm with a 10-year return period, under the condition of water diversion,  the sluices began to drain the flood water after 28 hours of rain, which was 36 hours ahead of the situation  with no

15     water diversion (at the 66th hour) and the The total flood volume that was discharged  by the ED project was 104 million m³  greater than that without the effect of water diversion (Fig. 89). When the NLB encountered a storm with a 20-year return period, under the condition of water diversion, the time  that the E dam project began to discharge the flood  occurred  26 hours after the rain started, which was 32 hours ahead of the situation with no water diversion (at the 58th hour) Compared with the scenario without water diversion,  the total discharge volume

20     increased 129 million m³  (Fig. 89).

[Figure]

**Figure 89: Changes in the flood discharge of the EDrji dam under water diversion with different storm levels. Flood discharge under rainfall events for different return periods: 5 years (a), 10 years (b), 20 years (c). The bar chart with different colours shows the three-day rainfall processes at each rainfall station. Black lines represent the flood-discharge process of the ED with and without the effect of water diversion.**

**6 Conclusions and policy implications**

The following selected conclusions are presented.

(1) In order tTo clarify the impacts that inter-basin water transfer has on the water-receiving areas, this study used theing MIKE model to simulate the flooding and waterlogging in the NLB. One- and two-dimensional coupled floodings and a waterlogging simulation models of the NLB were established to simulate the water diversion of the ER-SNWDP. The MIKE 11 model was applied to simulate the flow movement of water in the water diversion channel and the tributaries of NL, and the MIKE 21 model was applied to simulate the waterlogging in the lakeside area and the water flow in NL. The verification results show that the presented method can effectively simulate the flooding and waterlogging processes in the NLB under the effect of the ER-SNWDP.

(2) The ER-SNWDP emergency water transfer to NL increases the risk of waterlogging damage in the lakeside area if it occurs simultaneously with the DFAAabrupt drought-flood alternations. The increased water level caused by water diversion decreases the efficiency of waterlogging drainage, and as a result, the waterlogged area with an inundated water depth above 0.1 m increased by 0.99%, and that with an inundated water depth above 0.5 m increased by 13.32%. The flood-discharge time of the EDrji dam increased by 4 days, and the total discharge volume increased by 249 million m³ during the simulation.

(3) The ER-SNWDP emergency water transfer to Shandong Peninsula raised the water level of NL, which acted as a regulation and storage lake. Compared with the no water transfer situation, the waterlogging areas in the lakeside area increased when NL encountered a storms with 5-year, 10-year and 20-year return periods under water diversion. The calculation results show that water diversion has a more obvious effect on the waterlogging areas, with an inundated water

depth above 0.5 m,  and the area increasing by 8.4-43.1%. The total volume of flooding discharged by the E also increased. In addition, we found that with the increase in rainfall intensity, the influence of water diversion on the lakeside area in the NL inundated area gradually decreased and the water transfer had more serious effects  during a rainstorm with  lower return period.

5    Certain implications for the management of inter-basin water diversion and  lake basin along the route of the water diversion project are presented below.

(1) For a complicated flood control and drainage system that contains a number of hydraulic structures, such as pumping stations, sluices, and embankments, flood movement  behaviour regulations must be implemented and flood disaster losses must be reduced by establishing  one- and two-dimensional coupled hydrodynamic model to accurately simulate the

10   flow process and a clear movement direction of flood.

(2) For a long-distance inter-basin water transfer project, due to the large difference between high and low precipitation in  water- -supply and water-receiving areas (combined with global climate change and hydrological uncertainty), strengthening the analysis of the emergency water diversion influence on flood control and drainage is not only necessary for the scientific management of the inter-basin water transfer project but also conducive to realizing the expected benefits and

15   reducing the negative effects of the project.

(3) To reduce the inter-basin water transfer project effect on waterlogging in the water- -receiving area, we can take steps based on the following factors. First, additional emphasis should be placed on planning projects, increasing the number of waterlogging drainage pumping stations and enlarging the capacity of  flood discharge buildings in  water-receiving basin. Second,  hydrological forecasting and early warning ability should be improved and the accuracy and forecast

20   period of  rainfall event should be increased  to stop  water diversion or lower the water level of the lake before the rainstorm.

Regarding for future research, we can expand on the following  aspects: (1)  the vulnerability of hazard-affected bodies, populations, gross domestic product ( GDP) and other information should be considered to more accurately reflect waterlogging disasters in the research area;  (2) a case study analysis of

25   the balance between water transfer risk and water resource benefits should be conducted; and (3) in the future, the spatial distribution of roughness in NL should be considered to improve the accuracy of flood simulation in NL.

*Data availability.* All data except for the DEM of the lakeside and Nansihu Lake in 2013 were acquired by the authors. Data except for the DEM can be requested by email from the author wangzz77@163.com.

*Competing interests.* The authors declare that they have no conflict of interest.

30   *Author contribution.* Kun Wang prepared the manuscript with contributions from all co-authors. Kun Wang and Zongzhi Wang developed the model and Zongzhi Wang designed the scenario. Kelin Liu, Liang Cheng and Lihui Wang guided and participated in the modelling, Kelin Liu and Liang Cheng dealt with the boundary conditions of the model. Ailing Ye made the electric artworks and words processing.

*Acknowledgments.* This study was financially supported by the National Key Research and Development Program of China
35   (2017YFC0403504) and the National Science Foundation of China under grant No. 51479119 and 51579064.

**References**

An, W. C. and Li, X. M.: Phosphate adsorption characteristics at the sediment – water interface and phosphorus fractions in Nansi Lake, China, and its main inflow rivers, Environ. Monit. and Assess., 173–184, doi: 10.1007/s10661-007-0149-6, 2009.

Abbott, M.B.: Computational Hydraulics. Pitman, London, 1979.

de Andrade, J. G. P., Barbosa, P. S. F., Souza, L. C. A. and Makino, D. L.: Interbasin Water Transfers: The Brazilian Experience and International Case Comparisons, Water Resour. Manag., 25(8), 1915–1934, doi:10.1007/s11269-011-9781-6, 2011.

5   Arrighi, C., Brugioni, M., Castelli, F., Franceschini, S. and Mazzanti, B.: Urban micro-scale flood risk estimation with parsimonious hydraulic modelling and census data, Nat. Hazards Earth Syst. Sci., 13(5), 1375–1391, doi:10.5194/nhess-13-1375-2013, 2013.

Arrighi, C., Brugioni, M., Castelli, F., Franceschini, S. and Mazzanti, B.: Urban micro-scale flood risk estimation with parsimonious hydraulic modelling and census data, Nat. Hazards Earth Syst. Sci., 13(5), 1375–1391,
10      doi:10.5194/nhess-13-1375-2013, 2013.

Aron, G., White, E. L. and Coelen, S. P.: Feasibility of Interbasin Water Transfer, JAWRA J. Am. Water Resour. Assoc., 13(5), 1021–1034, doi:10.1111/j.1752-1688.1977.tb03867.x, 1977.

Bisht, D. S., Chatterjee, C., Kalakoti, S., Upadhyay, P., Sahoo, M. and Panda, A.: Modeling urban floods and drainage using SWMM and MIKE URBAN: a case study, Nat. Hazards, 84(2), 749–776, doi: 10.1007/s11069-016-2455-1, 2016.

15   Bureau of South to North Water Transfer of Planning, Designing and Management, Ministry of Water Resources. Brief Introduction of General Planning for South-to-North Water Transfer Project, China Water Resources, 2003(2):11-13+18. (In Chinese).

Cai, X. and Ringler, C.: Balancing agricultural and environmental water needs in China: Alternative scenarios and policy options, Water Policy, 9 (SUPPL. 1), 95–108, doi:10.2166/wp.2007.047, 2007.

20   Cole, D. S., Carver, W. B., Hall, B. S. and Slover, P. C.: Interbasin Transfers of Water, Proc. 2011 Georg. Water Resour. Conf., 2011.

Davies, B. R., Thoms, M. and Meador, M.: An assessment of the ecological impacts of inter–basin water transfers, and their threats to river basin integrity and conservation, Aquat. Conserv. Mar. Freshw. Ecosyst., 2(4), 325–349, doi:10.1002/aqc.3270020404, 1992.

25   Dixon, B. and Earls, J.: Effects of urbanization on streamflow using SWAT with real and simulated meteorological data, Appl. Geogr., 35(1–2), 174–190, 2012.

Dotto, C. B. S., Kleidorfer, M., Deletic, A., Rauch, W., Mccarthy, D. T. and Fletcher, T. D.: Performance and sensitivity analysis of stormwater models using a Bayesian approach and long-term high resolution data, Environ. Model. Softw., 26(10), 1225–1239, 2011.

30   Dutta, D., Herath, S. and Musiake, K.: Flood inundation simulation in a river basin using a physically based distributed hydrologic model, Hydrol. Process., 14(3), 497–519, 2015.

Emanuel, R. E., Buckley, J. J., Caldwell, P. V., McNulty, S. G. and Sun, G.: Influence of basin characteristics on the effectiveness and downstream reach of interbasin water transfers: Displacing a problem, Environ. Res. Lett., 10(12), doi:10.1038/s41598-017-06225-9, 2015.

35   Guo, Y., Liu S., Liang X Y. Study on Emergency Water Supply Scheme for North Extension of Eastern Route of South-to-North Water Transfer Project, Haihe Water Resources, 2018(3). (In Chinese).

Gupta, J. and Zaag, P. Van Der: Interbasin water transfers and integrated water resources management : Where engineering, science and politics interlock, Phys. Chem. Earth, 33, 28–40, doi:10.1016/j.pce.2007.04.003, 2008.

He, B., Huang, X., Ma, M., Chang, Q., Tu, Y., Li, Q., Zhang, K. and Hong, Y.: Analysis of flash flood disaster
40      characteristics in China from 2011 to 2015, Nat. Hazards, doi: 10.1007/s11069-017-3052-7, 2018a.

Hamel, P. and Fletcher, T. D.: Modelling the impact of stormwater source control infiltration techniques on catchment baseflow, Hydrol. Process., 5817–5831, doi:10.1002/hyp.10069, 2014.

Hsu, M. H., Chen, S. H. and Chang, T. J.: Inundation simulation for urban drainage basin with storm sewer system, J. Hydrol., 234(1–2), 21–37, doi: 10.1016/S0022-1694(00)00237-7, 2000.

Hu, S., Wang, Z., Wang, Y. and Zhang, L.: Total control-based unified allocation model for allowable basin water withdrawal and sewage discharge, Sci. China Technol. Sci., 53(5), 1387–1397, doi:10.1007/s11431-010-0155-8, 2010.

Hu, W., Zhai, S., Zhu, Z. and Han, H.: Impacts of the Yangtze River water transfer on the restoration of Lake Taihu, Ecol. Eng., 34(1), 30–49, 2008.

J V Boussinesq: Théorie des ondes et des remous qui se propagent le long d'un canal rectangulaire horizontal, encommuniquant au liquide contenu dans ce canal des vitesses sensiblement pareilles de la surface au fond, Mathématiques Pures et Appliquées 1872, 17(1): 55–108.

Karamouz, M., Mojahedi, S. A. and Ahmadi, A.: Interbasin water transfer: economic water quality-based model, J. Irrig. Drain. Eng., 136(2), 90–98, 2010.

Karim, F., Petheram, C., Marvanek, S., Ticehurst, C., Wallace, J. and Hasan, M.: Impact of climate change on floodplain inundation and hydrological connectivity between wetlands and rivers in a tropical river catchment, Hydrol. Process., 30(10), 1574–1593, 2016.

Khan, M. A., Vangani, N. S., Singh, N. and Singh, S.: Environmental Impact of Indira Gandhi Canal Project in Rawatsar Tehsil of Hanumangarh District, Rajasthan, Ann. Arid Zone, 38(2), 137–144, 1999.

Kundell, J. E.: INTERBASIN WATER TRANSFERS IN RIPARIAN STATES A CASE STUDY OF GEORGIA, Jawra J. Am. Water Resour. Assoc., 24(1), 87–94, 1988.

Larson K J, Başağaoğlu H, Mariño M A. Prediction of optimal safe ground water yield and land subsidence in the Los Banos-Kettleman City area, California, using a calibrated numerical simulation model. J. Hydrol., 242:79–102, 2001.

Lee, J. G. and Heaney, J. P.: Estimation of Urban Imperviousness and its Impacts on Storm Water Systems, J. Water Resour. Plan. Manag., 129(5), 419–426, 2003.

Li, W., Xu, B. and Wen, J.: Scenario-based community flood risk assessment: a case study of Taining county town, Fujian province, China, Nat. Hazards, 82(1), 193–208, doi: 10.1007/s11069-016-2187-2, 2016.

Liang, Y. S., Wang, W., Li, H. J., Shen, X. H., Xu, Y. L. and Dai, J. R.: The South-to-North Water Diversion Project: Effect of the water diversion pattern on transmission of Oncomelania hupensis, the intermediate host of Schistosoma japonicum in China, Parasites and Vectors, doi:10.1186/1756-3305-5-52, 2012.

Liu, C. and Zheng, H.: South-to-north water transfer schemes for China, Int. J. Water Resour. Dev., 18(3), 453–471, doi: 10.1080/0790062022000006934, 2002.

Liu, Q., Qin, Y., Zhang, Y. and Li, Z.: A coupled 1D–2D hydrodynamic model for flood simulation in flood detention basin, Nat. Hazards, 75(2), 1303–1325, 2015.

Liu, R., Liu, S. C., Cicerone, R. J., Shiu, C. J., Li, J., Wang, J. and Zhang, Y.: Trends of extreme precipitation in eastern China and their possible causes, Adv. Atmos. Sci., doi:10.1007/s00376-015-5002-1, 2015.

Ma, J., Hoekstra, A. Y., Wang, H., Chapagain, A. K. and Wang, D.: Virtual versus real water transfers within China, Philos. Trans. R. Soc. B Biol. Sci., doi:10.1098/rstb.2005.1644, 2006.

Matete, M. and Hassan, R.: Integrated ecological economics accounting approach to evaluation of inter-basin water transfers: An application to the Lesotho Highlands Water Project, Ecol. Econ., 60(1), 246–259, doi: 10.1016/j.ecolecon.2005.12.010, 2006.

Moel, H. De, Jongman, B., Kreibich, H., Merz, B., Penning-Rowsell, E. and Ward, P. J.: Flood risk assessments at different spatial scales, Mitig. Adapt. Strateg. Glob. Chang., 20(6), 865–890, 2015.

Quan, R. S.: Rainstorm waterlogging risk assessment in central urban area of Shanghai based on multiple scenario simulation, Nat. Hazards, 73(3), 1569–1585, doi: 10.1007/s11069-014-1156-x, 2014.

Rasmussen, P. W., Schrank, C. and Williams, M. C. W.: Trends of PCB concentrations in Lake Michigan coho and chinook salmon, 1975–2010, J. Great Lakes Res., 40(3), 748–754, 2014.

Sun, D. po, Xue, H., Wang, P. tao, Lu, R. li and Liao, X. long: 2-D Numerical Simulation of Flooding Effects Caused by South-to-North Water Transfer Project, J. Hydrodyn., 20(5), 662–667, doi:10.1016/S1001-6058(08)60110-9, 2008.

5 Tang, C., Yi, Y., Yang, Z. and Cheng, X.: Water pollution risk simulation and prediction in the main canal of the South-to-North Water Transfer Project, J. Hydrol., 519(PB), 2111–2120, doi:10.1016/j.jhydrol.2014.10.010, 2014.

Tian, C., Pei, H., Hu, W. and Xie, J.: Phytoplankton variation and its relationship with the environmental factors in Nansi Lake, China, Environ. Monit. Assess., 185(1), 295–310, doi:10.1007/s10661-012-2554-8, 2013.

Vrebos, D., Vansteenkiste, T., Staes, J., Willems, P. and Meire, P.: Water displacement by sewer infrastructure in the Grote
10 Nete catchment, Belgium, and its hydrological regime effects, Hydrol. Earth Syst. Sci., 18(3), 1119–1136, doi: 10.5194/hess-18-1119-2014, 2014.

Wang, H., Steyer, G. D., Couvillion, B. R., Rybczyk, J. M., Beck, H. J., Sleavin, W. J., Meselhe, E. A., Allison, M. A., Boustany, R. G. and Fischenich, C. J.: Forecasting landscape effects of Mississippi River diversions on elevation and accretion in Louisiana deltaic wetlands under future environmental uncertainty scenarios, Estuar. Coast. Shelf Sci.,
15 138(2), 57–68, 2014.

Wang, L., Gan, H., Wang, F., Sun, X. and Zhu, Q.: Characteristic analysis of plants for the removal of nutrients from a constructed wetland using reclaimed water, Clean - Soil, Air, Water, 38(1), 35–43, doi:10.1002/clen.200900162, 2010.

Wang, L., Yan, D., Wang, H., Yin, J. and Bai, Y.: Impact of the Yalong-Yellow River water transfer project on the eco-environment in Yalong River basin, Sci. China Technol. Sci., 56(4), 831–842, doi:10.1007/s11431-013-5155-z, 2013.

20 Wang, Z., Wu, J., Cheng, L., Liu, K. and Wei, Y.-M.: Regional flood risk assessment via coupled fuzzy c-means clustering methods: an empirical analysis from China's Huaihe River Basin, Nat. Hazards, 1–20, doi:10.1007/s11069-018-3325-9, 2018.

Wang, Y., Waterlogging disaster and its treatment in huai river basin. Science Press, in: Brief Introduction of Special Experiments on Typical Crops Submerged in Huaihe River Basin China, edited by: Peng S., Tian X., 94-141, 2015. (In
25 Chinese).

Webber, M., Crow-Miller, B. and Rogers, S.: The South–North Water Transfer Project: remaking the geography of China, Reg. Stud., 51(3), 370–382, doi:10.1080/00343404.2016.1265647, 2017.

Welch, E. B., Barbiero, R. P., Bouchard, D. and Jones, C. A.: Lake trophic state change and constant algal composition following dilution and diversion, Ecol. Eng., 1(3), 173–197, 1992.

30 Wu, Z., Zhang, J., Zhu, J., Ren, J. and Chen, S.: A monitoring project planning technique of the water quality spatial distribution in Nansi lake, Procedia Environ. Sci., 10(PART C), 2320–2328, doi:10.1016/j.proenv.2011.09.362, 2011.

Ye, A., Duan, Q., Chu, W., Xu, J. and Mao, Y.: The impact of the south-north water transfer project (CTP)'s central route on groundwater table in the Hai River basin, north China, Hydrol. Process., 28(23), 5755–5768, doi:10.1002/hyp.10081, 2014.

35 Zhai, S., Hu, W. and Zhu, Z.: Ecological impacts of water transfers on Lake Taihu from the Yangtze River, China, Ecol. Eng., 36(4), 406–420, 2010.

Zhang, L., Li, S., Loáciga, H. A., Zhuang, Y. and Du, Y.: Opportunities and challenges of interbasin water transfers: a literature review with bibliometric analysis, Scientometrics, 105(1), 279–294, doi: 10.1007/s11192-015-1656-9, 2015.

Zhang, Q.: The South-to-North Water Transfer Project of China: Environmental implications and monitoring strategy, J. Am.
40 Water Resour. Assoc., 45(5), 1238–1247, doi:10.1111/j.1752-1688.2009.00357.x, 2009.

Zhao, Z. Y., Zuo, J. and Zillante, G.: Transformation of water resource management: a case study of the South-to-North Water Diversion project, J. Clean. Prod., doi:10.1016/j.jclepro.2015.08.066, 2017.

Zhuang, W.: Eco-environmental impact of inter-basin water transfer projects: a review, Environ. Sci. Pollut. Res., doi: 10.1007/s11356-016-6854-3, 2016.

Zolghadr, M., Hashemi, M. R., Hosseinipour, E. Z. and Palmer, R. N. B. T.-W. E. and W. R. C.: Modeling of flood wave propagation through levee breach using MIKE21, a case study in Helleh River, Iran., or. 2683–2693., 2010.

**Appendix**

**Table 1 Statistical evaluation of model performance for water level simulations at selected gauging stations for 2007 and 2008 flood events.**

| Gauging station | NSE | |
| --- | --- | --- |
| | 2007 | 2008 |
| Nanyang | 0.72 | 0.65 |
| Makou | 0.69 | 0.76 |
| Erji Lake (downstream) | 0.67 | 0.98 |
| Weishan | 0.82 | 0.99 |

**Table 2 Computational scenario setting**

| Sr. no. | Initial water level of NL | Rainfall | Whether the ER-SNWDP works during the simulation |
| --- | --- | --- | --- |
| ① | 33.01 m in upper lake, 32.20 m in lower lake | Actual daily rainfall from August 22 to September 2, 2003 | NO |
| ② | 34.0 m in upper lake, 32.3 m in lower lake | | |
| ③ | | Designed storm with return periods of 5 years | |
| ④ | Actual water level on August 22, 2003 | 10 years | NO |
| ⑤ | | 20 years | |
| ⑥ | | 5 years | |
| ⑦ | 34.0 m in upper lake, 32.3 m in lower lake | 10 years | YES |
| ⑧ | | 20years | |

**Table 3 Results of waterlogging in the lakeside area of NL**

| Sr. no. | Water depth of NL | | Area with an inundated depth above 0.1 m | | Area with an inundated depth above 0.5 m | |
|---|---|---|---|---|---|---|
| | Average (m) | Max (m) | Total area (km$^2$) | Area ratio (%) | Total area (km$^2$) | Area ratio (%) |
| ① | 2.47 | 5.96 | 1126.59 | 32.51 | 383.68 | 11.07 |
| ② | 2.80 | 6.14 | 1160.85 | 33.50 | 434.77 | 12.55 |

**Table 4 Increment of the waterlogging area in the lakeside area of NL**

| Contrastive analysis | Area with an inundated depth above 0.1 m | | | Area with an inundated depth above 0.5 m | | |
|---|---|---|---|---|---|---|
| | Increment (km$^2$) | Relative increase (%) | Area ratio increase | Increment (km$^2$) | Relative increase (%) | Area ratio increase |
| Variation | 34.26 | 3.04 | 0.99 | 51.09 | 13.32 | 1.47 |

**Table 5 Results of waterlogging simulations of the lakeside area under different scenarios.**

| Sr. no. | Water depth of NL | | Area with an inundated depth above 0.1 m | | Area with an inundated depth above 0.5 m | |
|---|---|---|---|---|---|---|
| | Average (m) | Max (m) | Total area (km$^2$) | Area ratio (%) | Total area (km$^2$) | Area ratio (%) |
| ③ | 1.86 | 5.23 | 793.75 | 22.91 | 60.63 | 1.75 |
| ④ | 2.03 | 5.87 | 907.85 | 26.20 | 159.98 | 4.62 |
| ⑤ | 2.41 | 6.31 | 1002.05 | 28.92 | 240.54 | 6.94 |
| ⑥ | 2.10 | 5.87 | 816.02 | 23.55 | 86.77 | 2.50 |
| ⑦ | 2.26 | 6.10 | 926.40 | 26.73 | 181.73 | 5.24 |
| ⑧ | 2.17 | 6.08 | 1016.68 | 29.34 | 260.76 | 7.53 |

**SPRINGER NATURE | Author Services**

**Nature Research Editing Service Certification**

This is to certify that the manuscript titled Impacts of the emergency operation of the South-to-North Water Diversion Project's eastern route on flooding and drainage in the water-receiving area: An empirical case from China was edited for English language usage, grammar, spelling and punctuation by one or more native English-speaking editors at Nature Research Editing Service. The editors focused on correcting improper language and rephrasing awkward sentences, using their scientific training to point out passages that were confusing or vague. Every effort has been made to ensure that neither the research content nor the authors' intentions were altered in any way during the editing process.

Documents receiving this certification should be English-ready for publication; however, please note that the author has the ability to accept or reject our suggestions and changes. To verify the final edited version, please visit our verification page. If you have any questions or concerns over this edited document, please contact Nature Research Editing Service at support@as.springernature.com.

**Manuscript title:** Impacts of the emergency operation of the South-to-North Water Diversion Project's eastern route on flooding and drainage in the water-receiving area: An empirical case from China

**Authors:** zongzhi wang

**Key:** 1B19-EA74-31A7-694F-936P

This certificate may be verified at **secure.authorservices.springernature.com/certificate/verify**.

Nature Research Editing Service is a service from Springer Nature, one of the world's leading research, educational and professional publishers. We have been a reliable provider of high-quality editing since 2008.

Nature Research Editing Service comprises a network of more than 900 language editors with a range of academic backgrounds. All our language editors are native English speakers and must meet strict selection criteria. We require that each editor has completed or is completing a Masters, Ph.D. or M.D. qualification, is affiliated with a top US university or research institute, and has undergone substantial editing training. To ensure we can meet the needs of researchers in a broad range of fields, we continually recruit editors to represent growing and new disciplines.

Uploaded manuscripts are reviewed by an editor with a relevant academic background. Our senior editors also quality-assess each edited manuscript before it is returned to the author to ensure that our high standards are maintained.